# Characteristics, sources, and reactions of nitrous acid during winter at an urban site in the Central Plains Economic Region in China

Qi Hao[1,3], Nan Jiang[2,3]*, Ruiqin Zhang[2,3], Liuming Yang[1,3], and Shengli Li[2,3]

[1] College of Chemistry, Zhengzhou University, Zhengzhou 450001, China

[2] School of Ecology and Environment, Zhengzhou University, Zhengzhou 450001, China

[3] Research Institute of Environmental Science, Zhengzhou University, Zhengzhou 450001, China

## Abstract

Nitrous acid (HONO) in the core city of the Central Plains Economic Region was measured using an ambient ion monitor from January 9 to 31, 2019. Measurement time intervals were classified into the following periods in accordance with the daily mean values of $PM_{2.5}$: clean days (CD), polluted days (PD), and severely polluted days (SPD). The HONO concentrations during CD, PD, and SPD were 1.2, 2.3, and 3.7 ppbv, respectively. The contribution of the homogeneous reaction, heterogeneous conversion, and direct emission to HONO sources varied under different pollution levels. The mean values of the net HONO production of the homogeneous reaction ($P_{OH+NO}^{net}$) in CD, PD, and SPD periods were 0.13, 0.26, and 0.56 ppbv $h^{-1}$, respectively. The average conversions of $NO_2$ ($C_{HONO}$) in CD, PD, and SPD periods were $0.72 \times 10^{-2}$, $0.64 \times 10^{-2}$, and $1.54 \times 10^{-2}$ $h^{-1}$, respectively, indicating that the heterogeneous conversion of $NO_2$ was unimportant than the homogeneous reaction. Furthermore, the net production of the homogeneous reaction may have been the main factor for the increase in HONO under high-$NO_X$ conditions (i.e., when the concentration of NO was higher than that of $NO_2$) at nighttime. Daytime HONO budget analysis showed that the mean values of the unknown source ($P_{unknown}$) during CD, PD, and SPD periods were 0.26, 0.40, and 1.83 ppbv $h^{-1}$, respectively. The values of $P_{OH+NO}^{net}$, $C_{HONO}$, and $P_{unknown}$ in the SPD period were comparatively larger than those in other periods, indicating that HONO participated in many reactions. The proportions of nighttime HONO sources also changed during the entire sampling period. Direct emission and a

heterogeneous reaction controlled HONO production in the first half of the night and
provided a contribution larger than that of the homogeneous reaction. The proportion
of homogenization gradually increased in the second half of the night due to the steady
increase in NO concentration. The hourly level of HONO abatement pathways, except
for OH + HONO, was at least 0.22 ppbv $h^{-1}$ in the SPD period. The cumulative
frequency distribution of the $HONO_{emission}$/HONO ratio (less than 20%) was
approximately 77%, which suggested that direct emission was not important. The
heterogeneous HONO production increased when the relative humidity (RH) increased,
but it decreased when RH increased further. The average $HONO/NO_X$ ratio (4.9%) was
more than twice the assumed globally averaged value (2.0%).
**1. Introduction**

Nitrous acid (HONO) is important in the photochemical cycle and can provide

hydroxyl radicals (OH) (Harrison et al., 1996):
$HONO + hv \rightarrow \cdot OH + NO$ (300 nm < $\lambda$ < 405 nm)           (R1).
According to measurement and simulation studies (Alicke et al., 2002), the contribution
of HONO to $\cdot$OH concentration can reach 25−50%, especially when the concentration
of OH radicals produced by the photolysis of ozone, acetone, and formaldehyde is
relatively low (two to three hours after sunrise) (Czader et al., 2012). HONO photolysis
was the most important primary source of $\cdot$OH which contributed up to 46 % of the
total primary production rate of radicals for daytime conditions (Tan et al., 2018). $\cdot$OH
is an important oxidant in the atmosphere, and it can react with organic substances,
control the oxidation capacity of the atmosphere, and accelerate the formation of
secondary aerosols in the urban atmosphere (Sörgel et al., 2011). Therefore, the changes
in the contribution of the homogeneous reaction, heterogeneous conversion, and direct
emission during pollution can be observed by studying the formation mechanism of
HONO.

Several instruments have been used to determine ambient HONO concentrations,

and these include differential optical absorption spectrophotometer (DOAS)
(Elshorbany et al., 2012; Winer and Biermann, 1994), long path absorption photometer
(LOPAP) (Heland et al., 2001), wet chemical derivatization technique-HPLC/UV-Vis
detection (Michoud et al., 2014), stripping coil-UV/Vis absorption photometer (SC-AP)
(Pinto et al., 2014), IBBCEAS (Duan et al., 2018; Min et al., 2016), CIMS (Hirokawa
et al., 2009; Roberts et al., 2010), and ambient ion monitor (AIM) (VandenBoer et al.,
2014). A result comparison of different instruments showed that SC-AP is compatible
with two spectral measurement instruments, namely, LOPAP and DOAS (Pinto et al.,
2014). Compared with HONO measured by SC-AP deployed onsite, HONO measured
by AIM has a small error and is within the acceptable analytical uncertainty
(VandenBoer et al., 2014). Previous studies have reported that HONO concentrations
range from a few pptv in clean remote areas to several ppbv (0.1−2.1 ppbv) in air-
polluted urban areas (Hou et al., 2016; Michoud et al., 2014).

The sources of HONO are direct emission and homogeneous and heterogeneous

reactions (Acker et al., 2005; Grassian, 2001; Kurtenbach et al., 2001). HONO can be
directly discharged into the atmosphere during vehicle operation and biomass
combustion. Through a tunneling experiment, Kurtenbach et al. (2001) have discovered
that motor vehicles emit a small amount of HONO, and the HONO/$NO_X$ ratio of HONO
combustion sources (aside from $NO_X$ and other pollutants) is 0.1–0.8%. Another study
showed that the homogeneous reaction of NO and OH radicals is the major source of
HONO under increased NO concentrations (Spataro et al., 2013). Furthermore, HONO
can react with the ·OH (Alicke et al., 2003; Vogel et al., 2003). Tong et al. (2015) used
NO + OH and HONO + OH homogeneous reactions, to calculate the net generation rate
of HONO homogeneous reactions at night, which are expressed as:
$NO + \cdot OH \rightarrow HONO$ (R2);
$HONO + \cdot OH \rightarrow NO_2 + H_2O$ (R3).
Such calculations have been applied in studies on homogeneous reactions and daytime
budgets (Hou et al., 2016; Huang et al., 2017). These are studies of homogeneous
reactions, and some researchers have begun to explore the mechanism of $NO_2$
heterogeneous reactions. Finlayson-Pitts et al. (2003) studied the mechanism of
chemical adsorption of $NO_2$ and H ions on the adsorbed surface was revealed by using
isotope-labeled water:
$2NO_2 + H_2O \rightarrow HONO + HNO_3$ (R4).
In China, most studies for HONO have been focused on the Yangtze River Delta, Pearl
River Delta, and Jing-Jin-Ji region. For example, Hao et al. (2006) reported that field
measurement results, especially $HONO/NO_2$ and relative humidity (RH), have a
significant correlation and proved that heterogeneous reactions are an important source
of nighttime HONO. Although the specific chemical mechanisms of heterogeneous
reactions remain unknown, the intensity of HONO formation by $NO_2$ can be expressed
by the HONO conversion frequency (Alicke et al., 2002; Li et al., 2012). Su et al.
(2008a) revealed the importance of the $\cdot OH$ from HONO during daytime (9:00–15:00
local time) and found that many unknown sources which are closely related to the solar
radiation leading to HONO formation. The unknown sources of HONO may include
the $NO_2$ photolysis of sooty surface and adsorbed nitric acid and nitrate at UV
wavelengths (Kleffmann et al., 1999). The homogeneous nucleation of $NO_2$, $H_2O$, and
$NH_3$ is the HONO formation pathway (Zhang and Tao, 2010). In the meanwhile, HONO
can deposit and react with amines in forming nitrosamines (Li et al., 2012) for sinking.
The method of budget analysis needs to include the HONO sources and sinks. The
researchers suggested that the method of budget analysis is crucial for obtaining the
missing source. Spataro et al. (2013) measured the HONO level in Beijing's urban area
and discussed the spatiotemporal changes, meteorological effects, and contributions of
HONO from different sources. They used the measured HONO data to compare
pollution periods in Beijing's urban and suburban areas. Tong et al. (2015) discovered
that the pathway of the HONO formation mechanism, namely, direct emission,
heterogeneous formation, and homogeneous reaction is the same, but the pathway is
different in the two sites. A few studies (Cui et al., 2018; Hou et al., 2016) compared
the characteristics and sources of HONO during severe-pollution and clean periods.
Although the definitions of the two periods are different, both can be used to analyze
the diurnal variation, source, and daytime budget of HONO during the aggravation of
pollution.

There is no study of HONO in the Central Plains Economic Region (CPER), with

a total population of 0.18 billion by the end of 2011. CPER is the important region for
food production and modern agriculture published by the Chinese government
(http://www.gov.cn/zhengce/content/2011-10/07/content_8208.htm). The file
described the different factors which affect atmospheric pollution, including the level
of economic development, energy structure, industrial structure and geographical
location (solar radiation) with the Yangtze River Delta, Pearl River Delta, and Jing-Jin-
Ji region. As the core city of CPER, Zhengzhou characterized by severe PM (particulate
matters) pollution (Jiang et al., 2017; 2018d), is selected in the study. In recent years,
comprehensive PM research has been conducted on the chemical characteristics of PM
in Zhengzhou (Jiang et al., 2018b; Li et al., 2019), source apportionment (Jiang et al.,
2018c; 2018e; Liu et al., 2019), health risks (Jiang et al., 2019a; 2019b), and emission
source profiles (Dong et al., 2019; Jiang et al., 2018a). However, no study has been
performed on the sources and characteristics of HONO in Zhengzhou. Moreover, no
synthetic research on different pollution levels in the area is available. In the current
study, AIM was used to sample and analyze HONO concentrations. The interactions
between HONO and other factors, such as $PM_{2.5}$, during pollution, were assessed to
understand the formation and removal of HONO and the influence on different
pollution periods. The levels of $PM_{2.5}$ were divided into three periods to analyze the
HONO sources, sinks, and reactions in different periods. Many papers (Huang et al., 2017;
Tong et al., 2016) took $PM_{2.5}$ as the main control factor of HONO, and studied the differences
of HONO sources and characteristics between clean and polluted periods. No homogeneous
reaction, direct emission, heterogeneous reaction, and daytime budget analysis were conducted
during the period of worsening pollution (namely HD period in this paper). Total $NO_X$
emissions in cities with different leading factors of emissions have been declining year
by year due to Chinese government emission control measures, but some Chinese cities
are still in high-$NO_X$ areas (e.g. Beijing, Shanghai, Guangzhou and Zhengzhou.) (Kim
et al., 2015; Liu et al., 2017). Under high-$NO_X$ conditions, some papers (Cui et al., 2018;
Hou et al., 2016) suggested that heterogeneous reaction was the main source of HONO
and did not conduct a quantitative analysis of homogeneous reaction, especially in
winter. So, we explore relevant studies of homogeneous reactions. In addition, the
source contributions of HONO at night varied with the degree of pollution level were
not explained. RH was also analyzed to provid a detailed understanding of HONO
generation intensity under different RH conditions. Analysis of the sources of HONO
at night provides strong support for conducting HONO budget analysis during daytime.
To the best of the authors' knowledge, the formation characteristics of HONO at
continuous and high time resolutions and different pollution levels have not been
studied in Zhengzhou. This work can assist the governments of the CPER in
formulating policy to decrease the level of HONO precursors, i.e., NO and $NO_2$,
and HONO direct emission from the vehicle.

## 2. Experiment and methods

### 2.1. Sampling site and period

The sampling site is on the rooftop (sixth floor) of a building in Zhengzhou
University (34°48' N, 113°31' E), which is located in the northwestern part of
Zhengzhou, China. The observation height is about 20 m from the ground, and the
observation platform is relatively open without any tall buildings around. The site is
about 500 m from the western Fourth-Ring Expressway of Zhengzhou City and about
2 km from Lian Huo Expressway to the north. The measurement period was from
January 9 to 31, 2019. Daily data were divided into two periods, namely, daytime (7:00–
18:00 local time) and nighttime (19:00–6:00 the next day, LT).

### 2.2. Instruments

AIM (URG-9000D, Thermo, USA), an online ion chromatographic monitoring
system for particle and gas components in the atmosphere, was used to measure HONO
concentration continuously at a temporal resolution of 1 h. The atmospheric airflow
entered the $PM_{2.5}$ cyclone cutting head through the sample tube, and gas–solid
separation was performed with a parallel plate denuder with a new synthetic polyamide
membrane. The denuder had no moving parts and could be changed without stopping
the sampler. HONO was absorbed by the denuder with an absorption liquid (5.5 mol
$m^{-3}$ $H_2O_2$). The chemicals that could be oxidized were absorbed by $H_2O_2$ on the porous
membrane surface, but several gases (e.g., $O_2$ and $N_2$) were expelled by the air pump.
The abundance of other gaseous acids and bases affected the efficiency of HONO
collection by AIM due to the relation between Henry's law constant and pH. This
measurement method and its details have been successfully evaluated in many field
studies (Markovic et al., 2012; Wang et al., 2019; Yang et al., 2020), and shown in the
supplement. In addition, a QXZ1.0 automatic weather station (Yigu Technologies,
China) was used for synchronous observation of meteorological parameters, including
temperature (T), RH, wind direction (WD), and wind speed (WS). The temporal
resolution of the model analyzer (TE [used for measuring $O_3$], 48i [used for measuring
CO], 42i [used for measuring NO, $NO_X$, and $NO_2$], and TEOM 1405 $PM_{2.5}$ monitor
[used for measuring $PM_{2.5}$], Thermo Electron, USA) is 1 h. Detailed information can
be found in the work of (Wang et al., 2019). Measurement technique, detection limit,
and accuracy of measured species are shown in **Table S1**.
During the sampling period, all instruments were subject to strict quality control
to avoid possible contamination. The instrument accessories and sampling process were
periodically replaced and calibrated, respectively. The instrument parts and
consumables were changed before the observation process, and the sampling flow was
calibrated to reduce the negative effect of accessories. Before this measurement period,
the membrane of the denuder has been replaced and standard anion and cation solutions
have been prepared on Jan. 3rd. The standard curve should be drawn to ensure the
appropriateness of the correlation coefficient ($\geqslant$ 0.999) and the accuracy of the sample
retention time and response value. The minimum detection limit of AIM was 0.004
ppbv. Other detailed information can be found in the work of (Wang et al., 2019).

## 3. Results and Discussion

### 3.1. Temporal variations of meteorological parameters and pollutants

The daily changes in meteorological parameters and $PM_{2.5}$ are shown in **Fig. 1**. In
accordance with the daily average concentration level of $PM_{2.5}$, the analysis and
measurement process was divided into three periods (clean days [CD], polluted days
[PD], and severely polluted days [SPD]). The days wherein the daily averages of $PM_{2.5}$
were lower than the daily average of second grade in China National Ambient Air
Quality Standards (CNAAQS) (75 $\mu g\,m^{-3}$) represented CD (January 9, 16, 17, 21, 22,
23, 26, and 31), with RH ranging from 5 to 79% and WS ranging from 0 to 4.2 $m\,s^{-1}$.
The days wherein the daily averages of $PM_{2.5}$ were between 75 and 115 $\mu g\,m^{-3}$
represented PD (January 10, 15, 18, 20, 25, 27, and 28), with RH ranging from 17 to
86% and WS ranging from 0 to 4.6 $m\,s^{-1}$. The days wherein the daily averages of $PM_{2.5}$
were higher than 115 $\mu g\,m^{-3}$ represented SPD (January 11, 12, 13, 14, 19, 24, 29 and
30), with RH ranging from 30 to 96% and WS ranging from 0 to 3.5 $m\,s^{-1}$. Northwest
or east wind was observed in most of the observation periods, except for January 21–
22. WD was north, the maximum WS reached 4 m/s, the $PM_{2.5}$ concentration decreased
rapidly, and the effect of pollutant removal was evident. **Table 1** lists the data statistics
of HONO, $PM_{2.5}$, $NO_2$, NO, $NO_X$, $HONO/NO_2$, $HONO/NO_X$, $O_3$, CO, T, RH, WS, and
WD during the measurement period together with their mean value ± standard deviation.
The meteorological parameters in **Table 1** show that the average RH in CD, PD, and
SPD periods was 33, 49, and 68%, respectively. In SPD, RH was high and WD was low
(mean value of 0.4 $m\,s^{-1}$).
In accordance with the data on trace gases, the average HONO values in CD, PD,
and SPD were 1.1, 2.3, and 3.7 ppbv, respectively. The mean values of $NO_2$ were 25,
33, and 42 ppbv (46, 63, and 78 $\mu g\,m^{-3}$ lower than the first grade in CNAAQS [80 $\mu g$
$m^{-3}$]), respectively. The mean values of CO were 1, 1, and 2 ppmv (1, 2, and 2 $mg\,m^{-3}$
lower than the first grade in CNAAQS [4 $mg\,m^{-3}$]), respectively. **Fig. 2** shows the

concentration changes in HONO and gas species throughout the measurement period. The variations of the average HONO, $PM_{2.5}$, $NO_2$, and CO in the three periods were similar. The mean values of all pollutant concentrations except $O_3$ in the SPD period were the largest, and those in the CD period were the smallest. The highest mean value of $O_3$ occurred in the CD period, similar to previous observations (Hou et al., 2016; Huang et al., 2017; Zhang et al., 2019).

The HONO concentrations ranged from 0.2 to 14.8 ppbv and had an average of 2.5 ppbv, which is higher than the average values of 0.6 (Rappenglück et al., 2013), 1.5 (Hou et al., 2016), and 1.0 ppbv (Huang et al., 2017) in previous urban studies. The diurnal variations of HONO during the measurement were similar in the three periods, as shown in **Fig. 3** and **Fig. 4**. The diurnal variations of HONO, NO, $NO_2$, $O_3$, $HONO/NO_2$, and $HONO/NO_X$ are illustrated in **Fig. 4**. The error bars of **Fig. 4** were placed separately in the tables of the supplement (**Table S2**). After sunset, the HONO concentrations in CD, PD, and SPD began to accumulate due to the attenuation of solar radiation and the stabilization of the boundary layer (Cui et al., 2018). The maximum values of 1.7, 4.1, and 6.9 ppbv were reached in the morning (08:00–10:00 LT) in CD, PD, and SPD, respectively. After 10:00 LT, the HONO concentration decreased because of the increased solubility and rapid photolysis, remaining at a low level before sunset (14:00–16:00 LT). The NO concentration decreased rapidly in the forenoon, and remained low in the afternoon. After sunset, the concentrations of NO and $NO_2$ began to increase and remained at a higher level than the daytime. Furthermore, the diurnal variation of NO in the CD period was similar to that of $NO_2$. The peak was reached at around 09:00 LT due to vehicle emission in the morning rush hours, and the lowest value was observed at around 16:00 LT. After 18:00 LT, the boundary layer height decreased in the evening rush hours, resulting in an increase in NO and $NO_2$ concentrations (Hendrick et al., 2014). $O_3$ showed a diurnal cycle and had maximum values in CD, PD, and SPD periods in the afternoon. The $HONO/NO_2$ ratio is commonly used to estimate the formation of HONO in $NO_2$ transformation (Wang et

al., 2013). Compared with HONO formation, $NO_2$ transformation is less affected by the
migration of atmospheric airmass during atmospheric migration (Li et al., 2012). The
$HONO/NO_2$ ratio in the CD period began to increase after sunset and reached its peak
at night. Then, it decreased in the morning as a result of the enhancement of $NO_2$
emission and photolysis of HONO. However, the mean value of $HONO/NO_2$ in PD and
SPD periods gradually increased from nighttime and eventually reached the maximum
values of 14.3 and 18.9% at 09:00 and 10:00 LT, respectively. The average $HONO/NO_X$
ratio (4.9%) was more than twice the assumed globally averaged value (2.0%)
(Elshorbany et al., 2014). This result indicates that the strength of the heterogeneous
reaction increased slightly with the exacerbation of pollution. The $HONO/NO_2$ ratio
showed a diurnal cycle with a low level in the afternoon and a high level after sunset
due to the heterogeneous reaction of $NO_2$ on the ground and aerosol surface (Su et al.,
2008b). For comparison, the daytime and nighttime HONO, $HONO/NO_2$, and
$HONO/NO_X$ mean values in other cities around the world are listed in **Table 2**. The
values of HONO, $HONO/NO_2$, and $HONO/NO_X$ in Zhengzhou are relatively higher
than those in other parts of the world. The reason for this phenomenon is that
Zhengzhou is a high-$NO_X$ area which provides HONO with abundant precursors ($NO_2$
and NO) in winter (Kim et al., 2015).
**3.2. Nocturnal HONO sources and formation**
**3.2.1. Homogeneous reaction of NO and OH**

The homogeneous reaction of NO and OH (R2 and R3) is the main pathway of

HONO formation in the gas phase. Spataro et al. (2013) found that the formation
mechanism leads to an increase in HONO in high-pollution areas with an increase in
NO at night. $P_{OH+NO}^{net}$ can be understood as the net hourly HONO production amount
of homogeneous reaction and is calculated as

$P_{OH+NO}^{net} = k_{OH+NO} [OH][NO] - k_{OH+HONO} [OH][HONO]$          (1).

At T = 298 K and P = 101 kPa, the rate constants of $k_{OH+NO}$ and $k_{OH+HONO}$ are

$9.8 \times 10^{-12}$ and $6.0 \times 10^{-12}$ $cm^3$ $molecule^{-1}$ $s^{-1}$, respectively (Atkinson et al., 2004; Sander
et al., 2003). [OH] is the concentration of ·OH that was not measured during the
campaign. Tan et al. (2018) found that by the field measurement, the average
concentration of ·OH in Beijing at nighttime was about $2.5 \times 10^5$ molecule $cm^{-3}$.
Moreover, the same ·OH concentration was also used to calculate the homogeneous
reaction of HONO in the recent researches of Beijing (Zhang et al., 2019), Shanghai
(Cui et al., 2018), and Xi'an (Huang et al., 2017). And, nighttime OH concentration
increased as the latitude decreases ranged 3 to $6 \times 10^5$ molecule $cm^{-3}$ (Lelieveld et al.,
2016). Zhengzhou has a lower latitude than Beijing, so the concentration of OH used
in this study is $2.5 \times 10^5$ molecule $cm^{-3}$. $P_{OH+NO}^{net}$ primarily depends on the
concentrations of NO and HONO because the values of $k_{OH+NO}$ and $k_{OH+HONO}$ are close.
**Fig. 5** shows the nocturnal variations of $P_{OH+NO}^{net}$, NO, and HONO during CD, PD, and
SPD periods. The uncertainties of $P_{OH+NO}^{net}$, NO, and HONO in **Fig. 5** are shown in
**Table S3**. When the NO levels were high, the variations of $P_{OH+NO}^{net}$ followed those of
NO during the three periods (Atkinson et al., 2004). The mean value of $P_{OH+NO}^{net}$ was
0.33 ppbv $h^{-1}$, and the specific values in CD, PD, and SPD periods were 0.13, 0.26, and
0.56 ppbv $h^{-1}$, respectively. We assumed $\pm$ 50% ·OH values to estimate the
uncertainty of $P_{OH+NO}^{net}$. The ·OH values of $1.25 \times 10^5$ and $3.75 \times 10^5$ molecule $cm^{-3}$ were
calculated the $P_{OH+NO}^{net}$ values of 0.16 and 0.49 ppbv $h^{-1}$.
$P_{OH+NO}^{net}$ varied from 0.01 to 0.47 ppbv $h^{-1}$ during the CD period. The mean value
of $P_{OH+NO}^{net}$ increased before midnight, decreased after midnight, and increased slightly
at 3 am. In the PD period, $P_{OH+NO}^{net}$ ranged from 0.07 to 0.44 ppbv $h^{-1}$. The situation
was similar to that in the CD period, except that the value remained almost constant. In
addition, the contribution of HONO from homogeneous reaction during the SPD period
was larger than those in the CD and PD periods, and the level of $P_{OH+NO}^{net}$, with an
average value of 0.56 ppbv $h^{-1}$, was lower than the value in a previous study (2.18 ppbv
$h^{-1}$ in Beijing) (Tong et al., 2015). From 19:00 LT to 03:00 LT, the mean value of
$P_{OH+NO}^{net}$ increased from 0.15 to 0.9 ppbv $h^{-1}$. HONO increased from 2.84 to 4.59 ppbv
and subsequently decreased to 4.43 ppbv. By integrating $P_{OH+NO}^{net}$ during the eight
hours, the homogeneous reaction can provide an accumulated HONO formation of at
least 3.36 ppbv (i.e., $0.15 + 0.20 + 0.25 + 0.25 + 0.35 + 0.56 + 0.7 + 0.9$ ppbv). However,
the mean accumulation value of measured HONO in this nighttime period was merely
1.59 ppbv. With the increase in pollution level, the HONO accumulation period at
nighttime increased. This result indicates that first, the homogeneous reaction of OH +
NO is sufficient to augment HONO in the first half of the night, although $NO_2$
transformation and other sources may still exist. When the concentration of NO is
relatively high, the net production generated by OH + NO may be the leading factor for
the increase in HONO at night (Tong et al., 2015). Second, the hourly level of HONO
abatement pathways, except OH + HONO, should be at least 0.22 ppbv h$^{-1}$ (i.e., 3.36 –
1.59 ppbv)/8 h). This phenomenon may arise because the dry deposition on ground
surfaces can be the main HONO removal pathway at night, similar to a previous study
(Li et al., 2012).
**3.2.2. Direct emission**

At present, no HONO emission inventory or emission factor database for

Zhengzhou is available. As a result, estimating any HONO from direct emission is
difficult. In the current study, directly emitted HONO could have been generated by
vehicle exhaust and biomass combustion because the site is close to the western Fourth-
Ring Expressway of Zhengzhou City and about Lian Huo Expressway to the north.
Hence, only night data (17:00–06:00 LT) were considered to avoid the problem of
instant photolysis of directly emitted HONO. In a previous study, the HONO/$NO_X$ ratio
from tunnel measurement was set to 0.65% to estimate an upper limit of HONO emitted
by traffic near the site (Kurtenbach et al., 2001). The minimum value of HONO/$NO_X$
in the SPD period in the current work was 1.5%, which is slightly higher than the value
measured in the abovementioned study. Directly emitted HONO at night was not
transformed immediately. The HONO concentrations corrected by direct emissions are
given as

$[HONO]_{correct} = [HONO] – [HONO]_{emission} = [HONO] – 0.0065 \times [NO_X]$   (2),

where $[HONO]_{emission}$, $[NO_X]$, and 0.0065 are direct emission HONO concentration,
$NO_X$ concentration, and $HONO/NO_2$ direct emission ratio, respectively. The direct
emission contribution was estimated by comparing the direct emission HONO with the
observed HONO. The ranges of $HONO_{emission}/HONO$ in CD, PD, and SPD periods were
2–52%, 6–34%, and 2–41%, respectively, and the mean values were 17, 16, and 16%,
respectively. The frequency distribution of the $HONO_{emission}/HONO$ ratio at nighttime
is shown in **Fig. 6**. For this upper limit estimation, the frequency distribution of
$HONO_{emission}/HONO$ (less than 20%) was approximately 77%. Hence, direct emission
may not be the main reason for the high growth of HONO levels. Compared with the
direct emission of other sites, that of the measurement site accounted for a lower
proportion possibly because the site is relatively far from the highway on the campus.
**3.2.3. Heterogeneous conversion of $NO_2$ to HONO**
$NO_2$ is an important precursor for HONO formation. In addition, recent
field measurements in many urban locations have shown that a positive
correlation exists between HONO and $NO_2$ (Cui et al., 2018; Hao et al.,
2006; Huang et al., 2017; Zhang et al., 2019), suggesting they have a
common source. Moreover, Acker et al. (2005) reported that different
meteorological conditions may lead to significant differences in the
relationship between the source and receptor, and these differences lead to
various types of correlation. During the measurement period, the
$HONO/NO_2$ ratio varied between 1.3 and 59.0%, with an average of 7.6%,
which is slightly higher than the averaged value of 6.2% in a previous study
(Cui et al., 2018). The $HONO/NO_2$ ratio calculated in this work is much
larger than that calculated for direct emission (< 1%) (Kurtenbach et al., 2001),
suggesting that heterogeneous reactions may be a more important pathway
for HONO production than direct emissions. With regard to the
heterogeneous conversion of $NO_2$, several studies (An et al., 2012; Shen and
Zhang, 2013) have reported that the surface of soot particles is the medium
of $NO_2$ conversion. The contribution of soot surface to HONO production is
usually much lower than expected because the uptake efficiency of $NO_2$
decreases with the prolonged reaction time caused by surface deactivation.
The aerosol surface is an important medium for the heterogeneous
transformation from $NO_2$ to HONO (Liu et al., 2014). The mass
concentration of aerosols was used as an alternative to identify the influence
of aerosols in this study because the surface density of aerosols could not be
obtained.

The correlations between $PM_{2.5}$ and $HONO/NO_2$ ratio in CD, PD, and

SPD periods are shown in **Fig. 7**. With the exacerbation of the $PM_{2.5}$ level,
the average value of $HONO/NO_2$ gradually increased, indicating that the
aerosol surface occupied an important position in the heterogeneous
transformation. A comparison of $HONO/NO_2$ and HONO with $PM_{2.5}$ showed
that the correlation between $HONO/NO_2$ and $PM_{2.5}$ ($R^2 = 0.23$) was weaker
than that between HONO and $PM_{2.5}$ ($R^2 = 0.55$) in the entire period. The
main source of HONO could not have been the transformation of $NO_2$.
Notably, the HONO correlation in the PD period was significantly stronger
than that in the two other periods. This result proves that HONO-related
reactions occurred more frequently during this period. The fair correlation
between HONO and $PM_{2.5}$ may pinpoint the mainly anthropogenic origins of these two
pollutants with the high direct or indirect contribution of combustion sources. The
reason for the increased HONO during the heavy pollution period could be by the
comparatively high loading and large particle surface (Cui et al., 2018). Similar
phenomena have been observed in a correlation study on CO and HONO
wherein CO was used as a tracer for traffic-induced emissions and tested by
considering the correlation between HONO and CO over an identical time
interval (Qin et al., 2009). The correlation coefficient between HONO and
CO was relatively moderate ($R^2 = 0.43$), indicating that HONO and CO
could come from the same source of emissions. Generally speaking, CO and
NO are mainly related to combustion processes such as vehicle emissions,
fossil fuel and biomass combustion (Tong et al., 2016). Thus, fossil fuel and
biomass combustion may contribute to HONO production, but they can not
be measured directly.
The absorbed water influences the heterogeneous formation (Stutz et
al., 2004). The influence of RH on the heterogeneous conversion is shown
in **Fig. 7(d)**. When RH was less, the $HONO/NO_2$ ratio slowly increased.
When RH was increased, the $HONO/NO_2$ ratio began to increase rapidly
with RH. The $HONO/NO_2$ ratio decreased when RH reached a certain high
level. Similar variation patterns have been obtained in previous studies
(Huang et al., 2017; Qin et al., 2009; Tong et al., 2015). Surface adsorbed
water functions not only as sources but also as sinks of HONO by affecting
the hydrolysis of $NO_2$ and the sedimentation of HONO to generate HONO
(Ammann et al., 1998). When RH ranged at the middle level, the
heterogeneous conversion of $NO_2$ to HONO was more significant than that
of deposition. This phenomenon confirms that RH improved the conversion
efficiency (Stutz et al., 2004). However, the surface reached saturation when
RH reached a certain high level. The excess water restricted $NO_2$
transformation (Wojtal et al., 2011). The absorption and dissolution of
HONO by the saturated surface water layer caused $HONO/NO_2$ ratio to
decrease drastically.
The correlation between $HONO_{correct}$ and $NO_2$ at nighttime is shown in
**Fig. S1**. $HONO_{correct}$ was used in the calculation to exclude the influence of
direct emission on $NO_2$ conversion. The nocturnal variations of $HONO_{correct}$,
$NO_2$, and $HONO_{correct}/NO_2$ ratios in the CD, PD, and SPD periods are
presented in **Fig. 8**. The uncertainties of $HONO_{correct}$, $NO_2$, and
$HONO_{correct}/NO_2$ ratios in **Fig. 8** are shown in **Table S4**. In general, the
$HONO_{correct}/NO_2$ ratio reached its maximum at or before midnight but
decreased after midnight. In the PD and SPD periods, HONO was generated
by heterogeneous reaction (R4), and $NO_2$ decreased after midnight. The
production of HONO was equal to its loss (mainly night deposition), and
HONO concentration reached a relatively banlance. In the current study,
directly emitted HONO state (Stutz, 2002). The weak correlation between
nighttime $HONO/NO_2$ and $PM_{2.5}$ can be reasonably explained by the stable
$HONO_{correct}/NO_2$ ratio after midnight (Qin et al., 2009). A previous study (Xu
et al., 2015) found that a low $HONO_{correct}$ in the first half of the night (19:00–
00:00 LT) indicates an important contribution of automobile exhaust
emissions, and a low $HONO_{correct}$ in the second half of the night means
heterogeneous reactions dominate. Therefore, the heterogeneous reaction
conversion rate of HONO was calculated in the current study by using the
data of $HONO_{correct}$.
The conversion rate of HONO ($C_{HONO}$) is usually used as an indicator
to test the efficiency of $NO_2$ heterogeneous reactions. Total $HONO_{correct}$ was
assumed to be generated by the heterogeneous transformation of $NO_2$. The
formula for the conversion rate of $NO_2$ ($C_{HONO}$) is as follows (Su et al.,
2008a; Xu et al., 2015):
$$C_{HONO} = \frac{([HONO_{correct}]_{t2} - [HONO_{correct}]_{t1}}{(t2 - t1)\,[NO_2]} \qquad (3),$$
where $[NO_2]$ is the average concentration of $NO_2$ within the t2–t1 time
interval (1 h). In this study, the averaged conversion rate of $NO_2$ was
$1.02 \times 10^{-2}\,h^{-1}$. The mean values of $C_{HONO}$ in the CD, PD, and SPD periods
were $0.72 \times 10^{-2}$, $0.64 \times 10^{-2}$, and $1.54 \times 10^{-2}\,h^{-1}$, respectively. The averaged
conversion rates in this study were $0.58 \times 10^{-2}$ and $1.46 \times 10^{-2}\,h^{-1}$ higher than
those of Beijing I (polluted) and II (heavily polluted) periods, respectively.
The increase in the conversion rate demonstrates that $NO_2$ had high reaction
efficiency through the process from $NO_2$ to HONO in the aggravation of
pollution, which could have led to the high utilization efficiency of the
aerosol surface. The exact uptake coefficients of $NO_2$ on ground and aerosol surfaces
are variable and should be different (Harrison and Collins, 1998). The present analysis
simplified this process by treating the ground and aerosol surfaces the same. The uptake
coefficient is mainly dependent on the surface characteristics, e.g. surface type and
moisture (Lu et al., 2018).
**3.3. Daytime HONO budget**
The expression of $d\,HONO\,/\,d\,t$ represents the observed variations of hourly
HONO concentrations, for which we can use $\Delta\,HONO/\Delta\,t$ instead:
$d\,HONO\,/\,d\,t = sources - sinks$
$\qquad = (P_{unknown} + P_{OH+NO} + P_{emi} + P_{het}) - (L_{OH+HONO} + L_{photo})$ (4),
$P_{OH+NO} = k_{OH+NO}\,[OH]\,[NO]$ (5),
$L_{OH+HONO} = k_{OH+HONO}\,[OH]\,[HONO]$ (6).
The $d\,HONO\,/\,d\,t$ calculated from the measurements was small and evenly
distributed around zero (Li et al., 2012). $P_{unknown}$ is the production rate by an
unknown daytime HONO source. $P_{OH+NO}$ is the rate of reaction of NO and
OH. $P_{emi}$ represents the direct emission rate of HONO from combustion
processes. By studying the source and reduction, the daytime HONO budget was
analyzed with Eq. (4) (Su et al., 2008b). The heterogeneous transformation
mechanism was assumed to be the same for day and night. Therefore, the
daytime heterogeneous productivity ($P_{het} = C_{HONO} \times [NO_2]$) was calculated
with the nighttime mean values of $C_{HONO}$ in different periods. $L_{OH+HONO}$ is
the rate of the reaction between OH and HONO (R3). The calculation
formulas of $P_{OH+NO}$ and $L_{OH+HONO}$ have been provided in **Section 3.2.1**. Upon
sunlight irradiation, ·OH and NO were formed as R1. $L_{photo}$ represents the
photolysis loss rate of HONO ($L_{photo} = J_{HONO} \times [HONO]$). The photolysis
frequency and ·OH concentration could not be directly measured in this
study. Therefore, the tropospheric ultraviolet and visible (TUV) transfer

model of the National Center for Atmospheric Research (http://cprm.acom.ucar.edu/Models/TUV/ Interactive_TUV/) (Hou et al., 2016) was used to calculate the $J_{HONO}$ value. The $J_{HONO}$ values obtained this way were assumed in clear sky days without clouds. $O_3$ column and the surface albedo. $O_3$ column density measured by the Ozone Monitoring Instrument (OMI, data available at https://ozonewatch.gsfc.nasa.gov/data/omi/Y2019/). The $O_3$ column density ranges from 292 to 306 DU during the entire period. The experimental site being situated in an urban region, the surface albedo is considered as 0.13 (Sailor, 1995). The ground elevation and the measurement altitude are 168 and 188 m respectively. The concentration of OH radicals was calculated with the formulas of $NO_2$, $O_3$, and $J_{O^1D}$ in the supplement (Rohrer and Berresheim, 2006). Aerosol effects were considered by using aerosol optical thickness (AOD), single scattering albedo (SSA), and Angstrom exponent as inputs in the TUV model. Typical AOD, SSA, and Angstrom exponent values of 1.32, 0.9, and 1.3, respectively, were adopted for the PD and SPD periods. In the CD period, the respective values were 0.66, 0.89, and 1.07 (Che et al., 2015; Cui et al., 2018; Hou et al., 2016). We wanted to study that under the same output conditions from the TUV model in the PD and SPD periods, the impact of different pollution levels changed on the daytime budget. Hence, the average profiles of $J_{HONO}$ and $J_{O^1D}$ concentrations in the CD, PD, and SPD periods are shown in **Fig. 9**. The mean values of $J_{HONO}$ and $\cdot OH$ concentration at noon in the CD, PD, and SPD periods were $5.93 \times 10^{-4}$, $3.79 \times 10^{-4}$, and $3.79 \times 10^{-4}$ molecule $cm^{-3}$ and $4.10 \times 10^6$, $2.93 \times 10^6$, and $3.76 \times 10^6$ molecule $cm^{-3}$, respectively. The results of the calculated OH radicals ranged from $(0.58-11.49) \times 10^6$ molecule $cm^{-3}$, and the mean value was $3.57 \times 10^6$ molecule $cm^{-3}$ at noon in Zhengzhou.

Each production and loss rate of daytime HONO during CD, PD, and SPD periods is illustrated in **Fig. 9** together with dHONO/dt. $P_{unknown}$ was at

a high level before midday. $P_{unknown}$ approached 0 ppbv $h^{-1}$ after midday. In the CD, PD, and SPD periods, the mean values of $P_{unknown}$ were 0.26, 0.40, and 1.83 ppbv $h^{-1}$, respectively; the mean values of $P_{OH+NO}$ were 1.14, 2.07, and 4.03 ppbv $h^{-1}$, respectively; the mean values of $P_{emi}$ were 0.17, 0.30, and 0.43 ppbv $h^{-1}$, respectively; and the mean values of $P_{het}$ were 0.14, 0.18, and 0.55 ppbv $h^{-1}$, respectively. The midday time $P_{unknown}$ (1.83 ppbv $h^{-1}$) calculated in Zhengzhou during the winter haze pollution period was close to the result obtained from Beijing's urban area (Hou et al., 2016) (1.85 ppbv $h^{-1}$). The $P_{unknown}$ contribution to daytime HONO sources in CD, PD, and SPD periods accounted for 15, 14, and 28% of the HONO production rate ($P_{unknown}$ + $P_{OH+NO}$ + $P_{emi}$ + $P_{het}$), respectively. Previous studies (Spataro et al., 2013; Yang et al., 2014) have shown that meteorological conditions, such as solar radiation and WS, can affect unknown sources. The low $P_{unknown}$ contribution of daytime HONO concentration may be related to the low solar radiation and low wind speed during severe pollution. The concentration of NO has a great influence on $P_{OH+NO}$, so the homogeneous reaction is still an important pathway of HONO production during the daytime. In addition to the photolysis of HONO and the homogeneous reaction of HONO and OH, one or more important sinks might exist to control the variation between the sources and sinks of the daytime HONO during complex contamination. However, further research is needed to analyze the unknown sources of daytime HONO.

## 4. Conclusions

Ambient HONO measurement using AIM with other atmospheric pollutants and meteorological parameters was conducted in the CPER. The HONO concentrations during the entire measurement varied from 0.2 to 14.8 ppbv, with an average of 2.5 ppbv. The HONO concentrations in the CD, PD, and SPD periods were 1.1, 2.3, and 3.7 ppbv, respectively, and the HONO/$NO_2$ ratios were 4.7, 7.1, and 9.4%, respectively. HONO concentration was a combined action of direct emission and heterogeneous reaction, and the contributions of the two were higher than that of homogeneous

reaction in the first half of the night. However, the proportion of homogenization gradually increased in the second half of the night due to the steady increase in NO concentration. The hourly level of other HONO abatement pathways aside from OH + HONO should be at least 0.22 ppbv h$^{-1}$ in the SPD period. The sum of the frequency distributions of the HONO$_{emission}$/HONO ratio (less than 20%) was approximately 77%, indicating that the direct emission of HONO was not the main source of the observed HONO level at night. The mean values of HONO$_{emission}$/HONO in the CD, PD, and SPD periods were 17, 16, and 16%, respectively. This phenomenon means that the policy of restricting motor vehicles published by the local government in January 2019 had a good effect on decreasing HONO emissions. In addition, when RH increased at the middle level, the heterogeneous HONO production increased, but it decreased when RH increased further due to the effect of surface water. The contribution of the three sources varied with different pollution levels. The mean values of $C_{HONO}$ in the CD, PD, and SPD periods were $0.72\times10^{-2}$, $0.64\times10^{-2}$, and $1.54\times10^{-2}$ h$^{-1}$, respectively. At nighttime in the SPD period, the heterogeneous conversion of NO$_2$ appeared to be unimportant. Furthermore, the net production generated by homogeneous reaction may be the leading factor for the increase in HONO under high-NO$_X$ conditions (i.e., the concentration of NO was relatively higher than that of NO$_2$) at nighttime. The mean value of $P_{OH+NO}^{net}$ in the CD, PD, and SPD periods were 0.13, 0.26, and 0.56 ppbv h$^{-1}$, respectively. Daytime HONO budget analysis showed that the mean values of $P_{unknown}$ in the CD, PD, and SPD periods were 0.26, 0.40, and 1.83 ppbv h$^{-1}$, respectively. Although the values of $P_{OH+NO}$ had high uncertainty because of the variation of NO concentrations, $P_{OH+NO}$ contributed the most to HONO production during the daytime. After the analysis, $C_{HONO}$, $P_{OH+NO}^{net}$, and $P_{unknown}$ in the SPD period were larger than those in the other periods, indicating that HONO participated in many reactions.

## Data availability

All the data used in this paper are available from the corresponding author upon

request (jiangn@zzu.edu.cn).

## Author contributions

NJ, RZ, and, SL conceived and designed the study. QH analyzed the data and wrote
the paper. LY performed aerosol sampling and data analyses.

## Competing interests

The authors declare that they have no conflict of interest.

## Acknowledgments

The study was supported by the financial support from the National Natural
Science Foundation of China (51808510, 51778587), National Key Research and
Development Program of China (2017YFC0212400), Natural Science Foundation of
Henan Province of China (162300410255), National Research Program for Key Issues
in Air Pollution Control (DQGG0107).

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

**Figure Captions:**

Fig. 1. Temporal trends of hourly average T, RH, WD, WS, and $PM_{2.5}$ during the measurement. (The shaded areas: white for the CD period; gray for the PD period; red for the SPD period.)

Fig. 2. Temporal variations of hourly average HONO, NO, $NO_2$, $O_3$, and CO during the measurement. (The shaded areas: white for the CD period; gray for the PD period; red for the SPD period.)

Fig. 3. Diurnal variations of HONO during the measurement.

Fig. 4. Diurnal variations of HONO, NO, $NO_2$, $O_3$, $HONO/NO_2$, and $HONO/NO_X$. The blue points and lines represented the CD period; the black points and lines represented the PD period; the red points and lines represented the SPD period.

Fig. 5. Nocturnal variations of $P_{OH+NO}^{net}$, HONO and NO during CD, PD and SPD periods.

Fig. 6. Percentage distribution of the nighttime $HONO_{emission}/HONO$. (The dotted line represents the average of $HONO_{emission}/HONO$.)

Fig. 7. Nighttime correlation studies between $PM_{2.5}$ and $HONO/NO_2$, $PM_{2.5}$ and HONO, CO and HONO, RH and $HONO/NO_2$ during the entire measurement period, CD, PD, and SPD periods. The blue represented the full measurement period; the light blue represented CD period; the black represented PD period; the red represented SPD period.

Fig. 8. Nocturnal variations of $HONO_{correct}$, $NO_2$, and $HONO_{correct}/NO_2$ in CD, PD and SPD periods.

Fig. 9. The average profiles of $J_{HONO}$ and $J_O{}^1{}_D$ concentrations during the daytime, and production and loss rate of the daytime HONO in CD, PD and SPD periods.

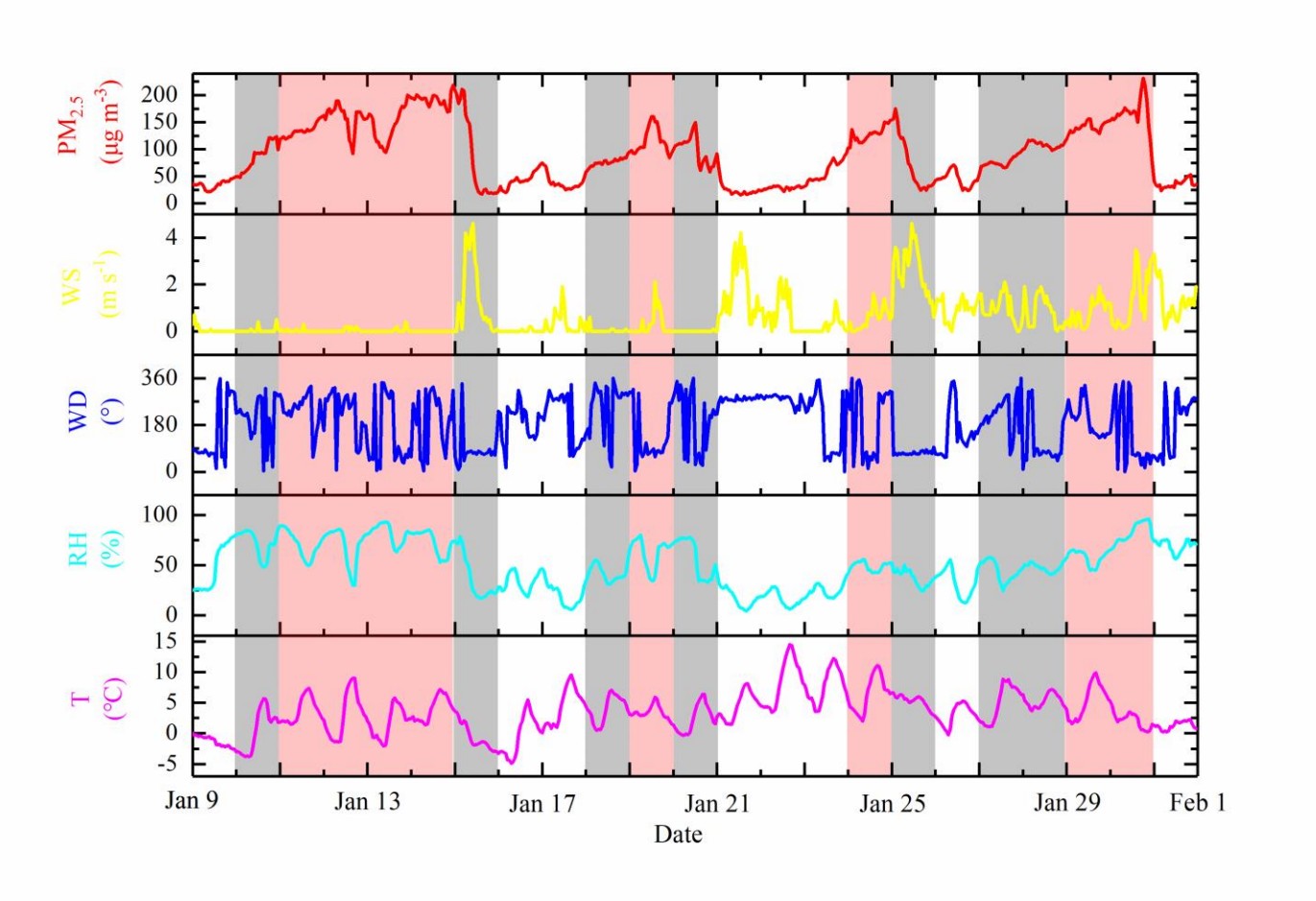

**Fig. 1.** Temporal trends of hourly average T, RH, WD, WS, and PM$_{2.5}$ during the measurement. (The shaded areas: white for the CD period; gray for the PD period; red for the SPD period.)

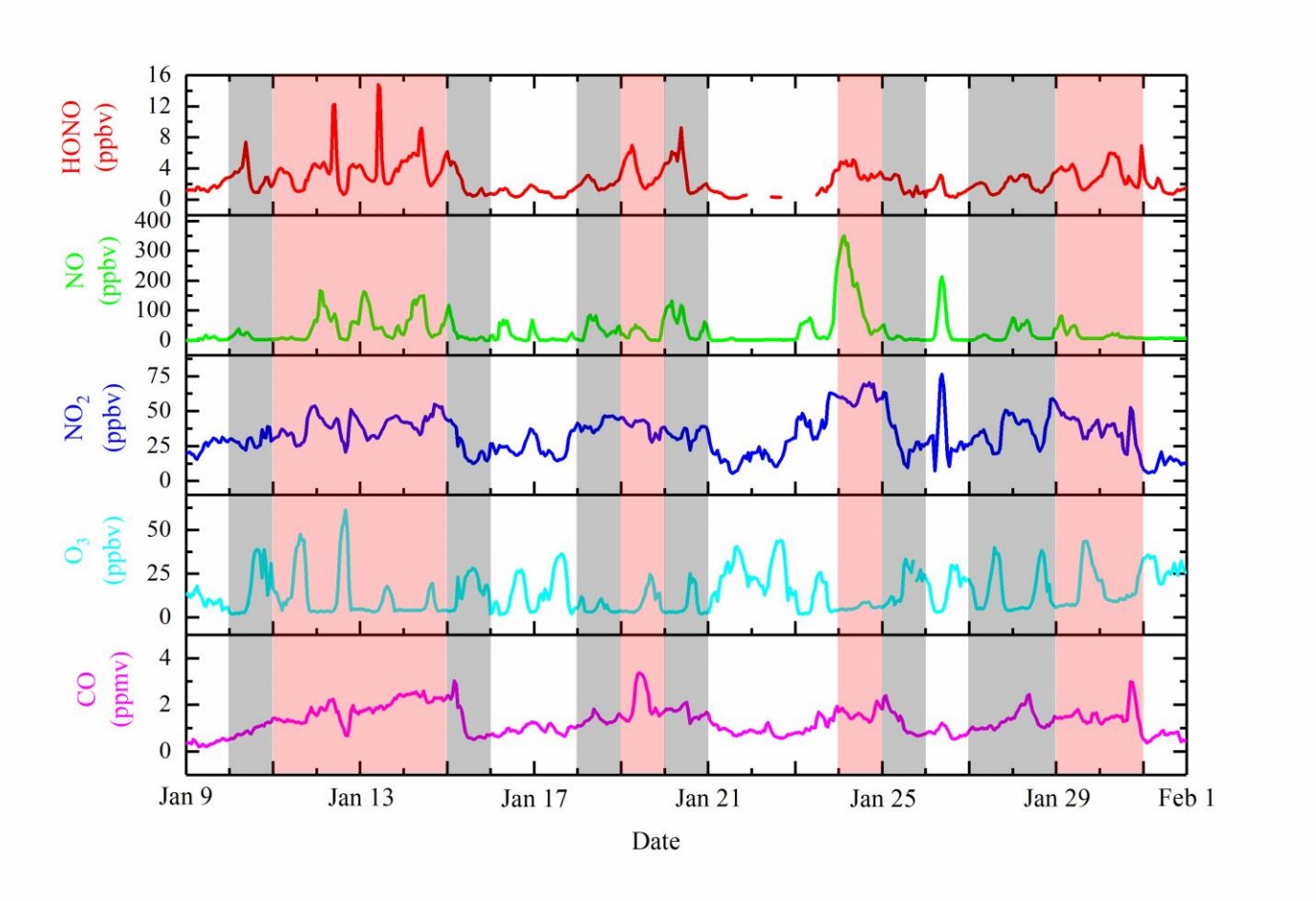

**Fig. 2.** Temporal variations of hourly average HONO, NO, NO₂, O₃, and CO during the measurement. (The shaded areas: white for the CD period; gray for the PD period; red for the SPD period.)

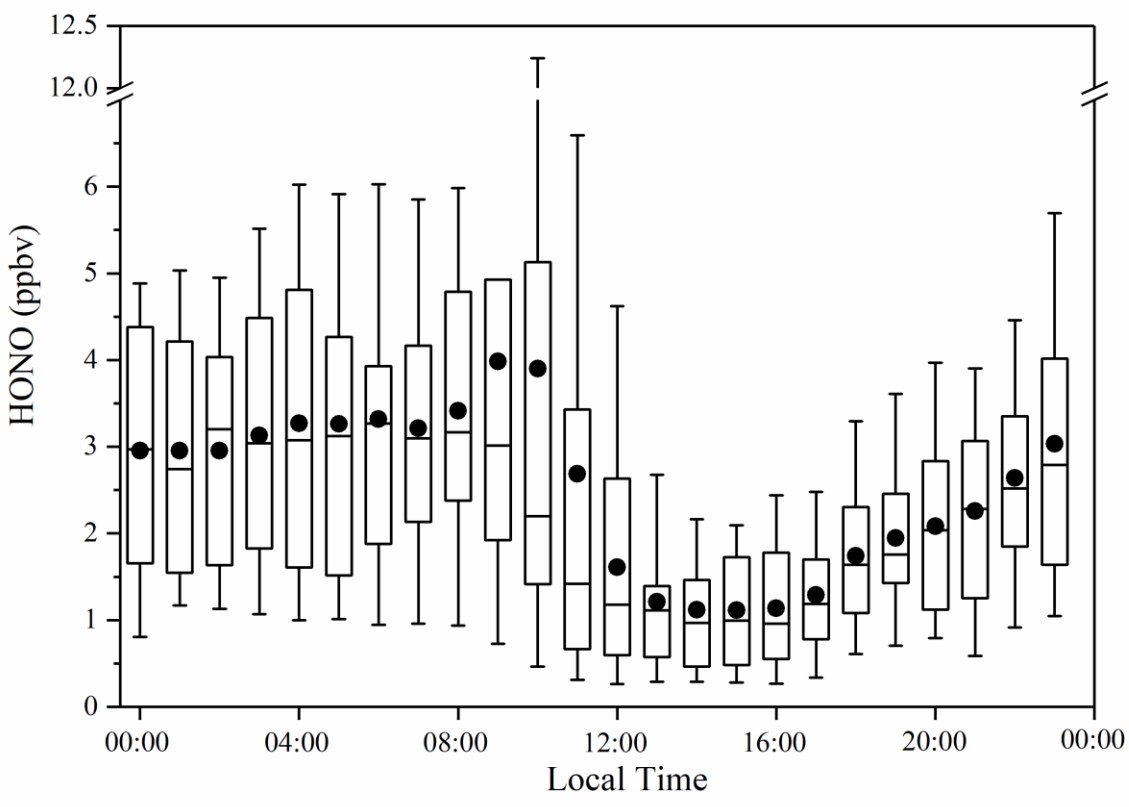

**Fig. 3**. Diurnal variations of HONO during the measurement.

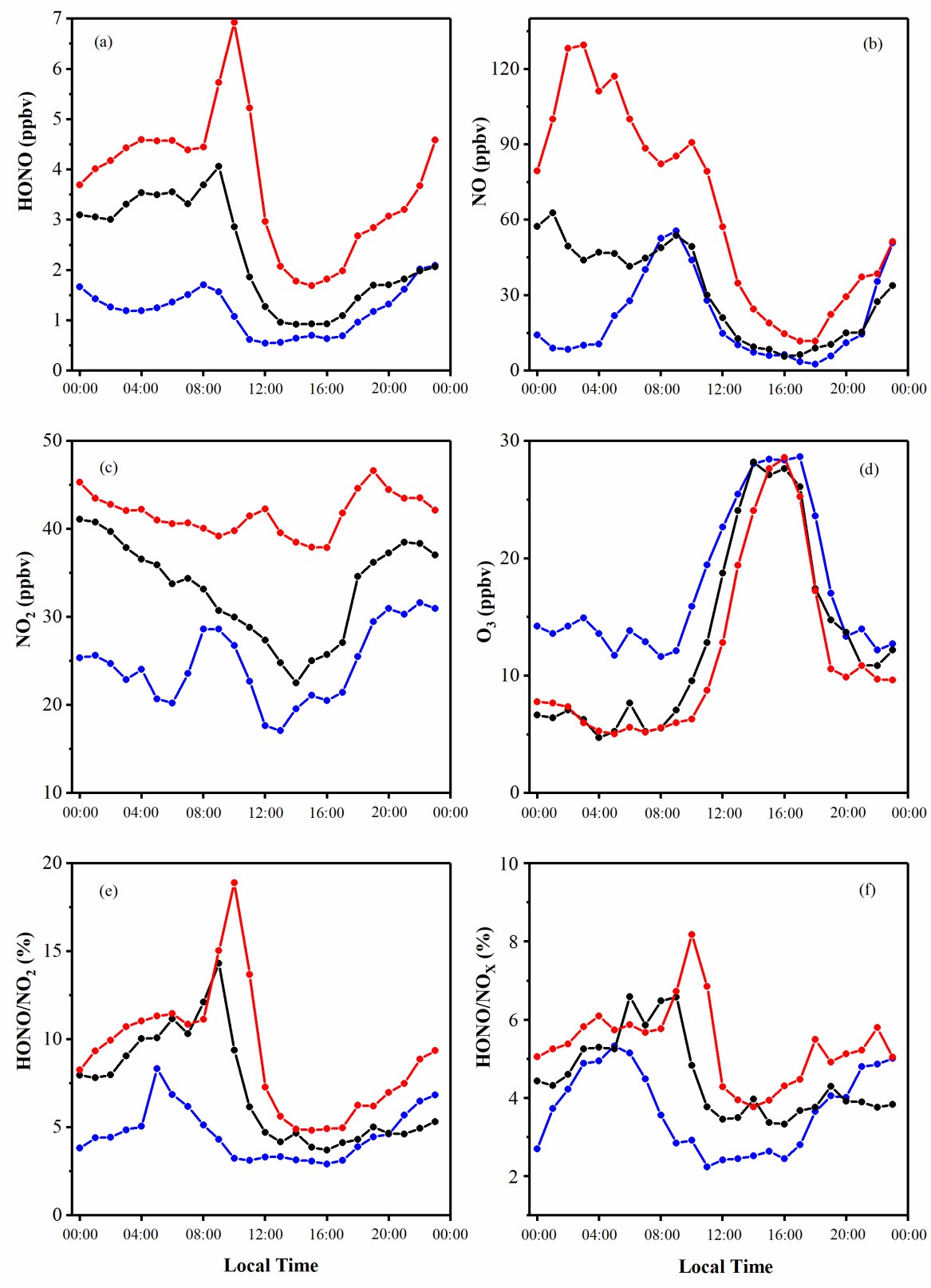

**Fig. 4.** Diurnal variations of HONO, NO, NO$_2$, O$_3$, HONO/NO$_2$, and HONO/NO$_X$. The blue points and lines represented the CD period; the black points and lines represented the PD period; the red points and lines represented the SPD period.

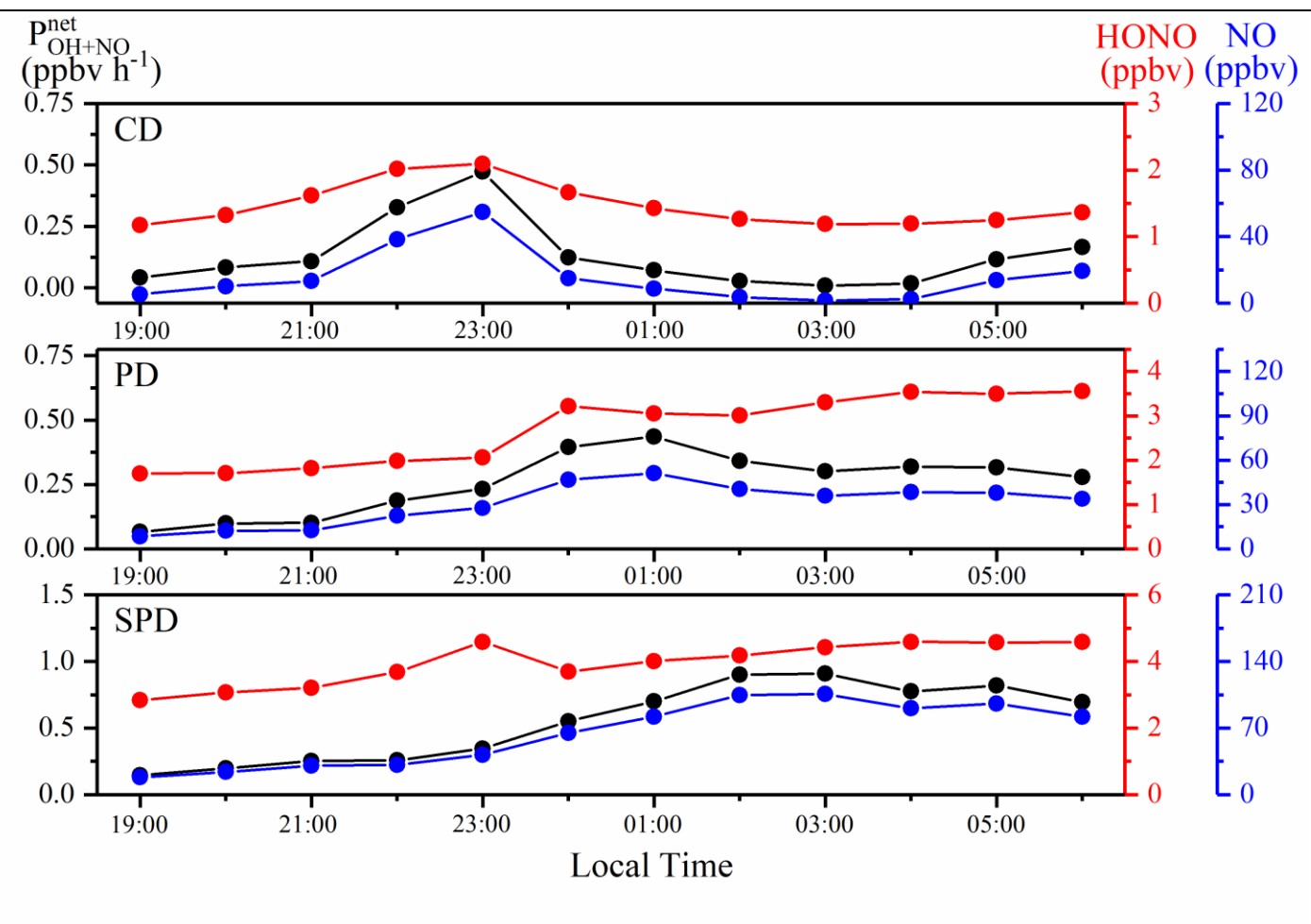

**Fig. 5.** Nocturnal variations of $P^{net}_{OH+NO}$, HONO and NO during CD, PD and SPD periods.

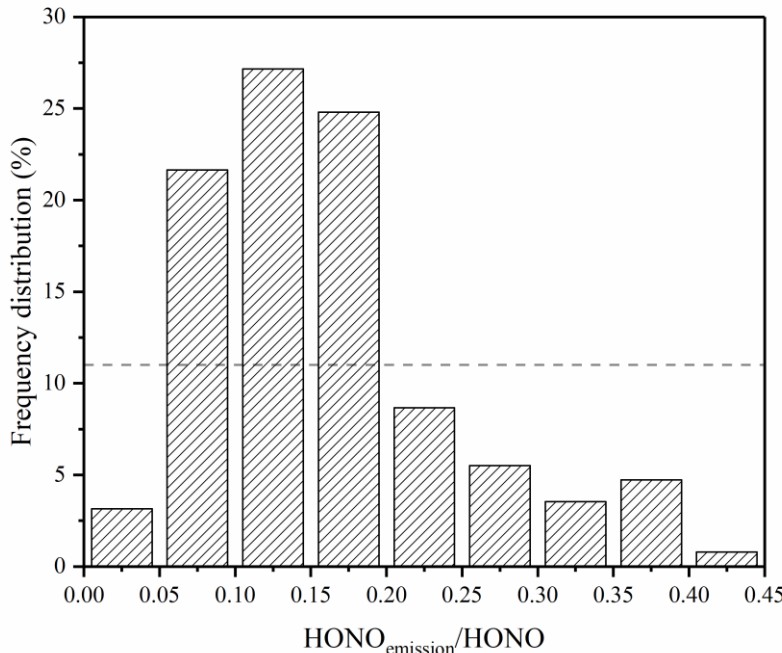

**Fig. 6.** Percentage distribution of the nighttime HONO$_{emission}$/HONO. (The dotted line represents the average of HONO$_{emission}$/HONO.)

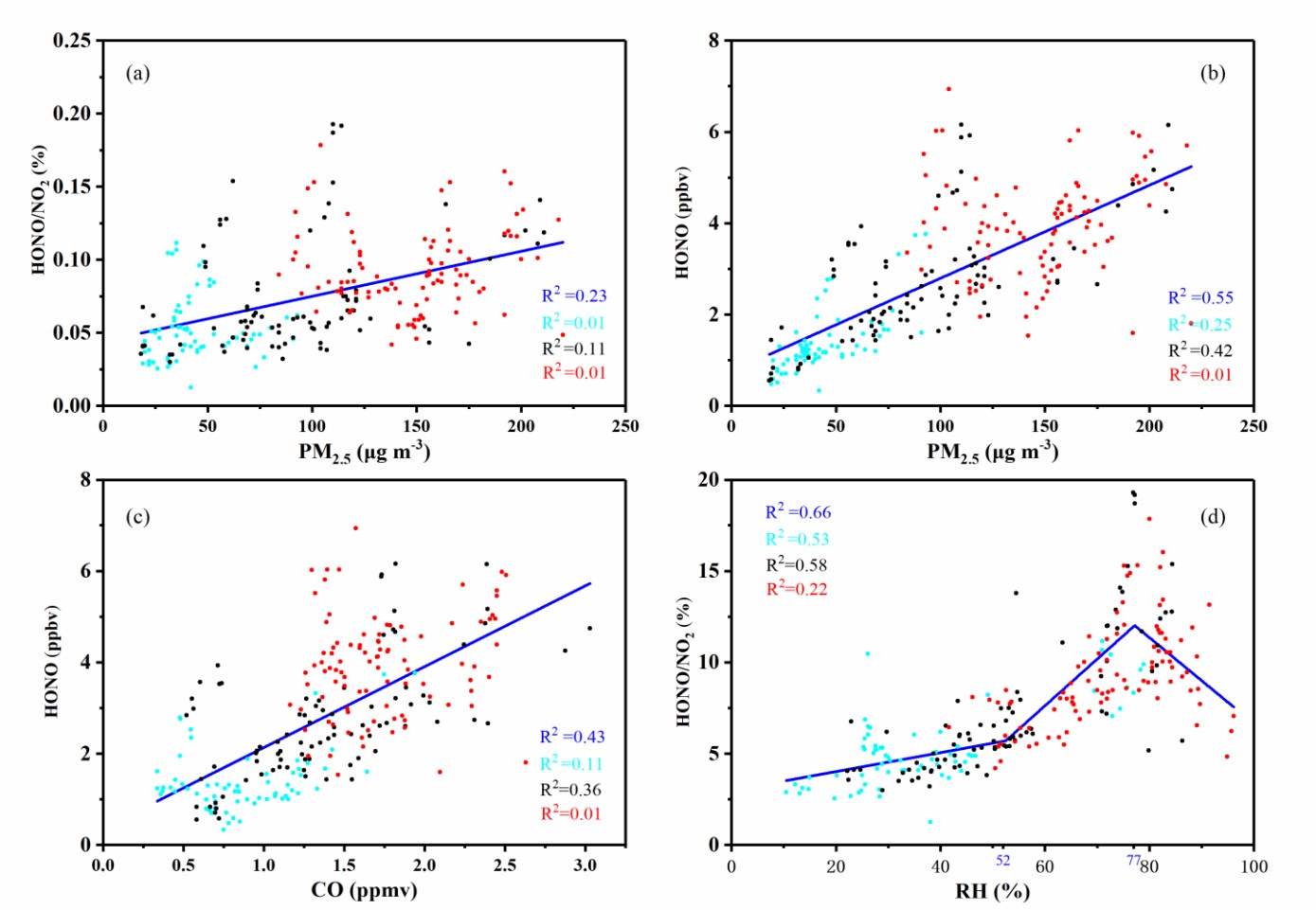

**Fig. 7.** Nighttime correlation studies between PM$_{2.5}$ and HONO/NO$_2$, PM$_{2.5}$ and HONO, CO and HONO, RH and HONO/NO$_2$ during the entire measurement period, CD, PD, and SPD periods. The blue represented the full measurement period; the light blue represented CD period; the black represented PD period; the red represented SPD period.

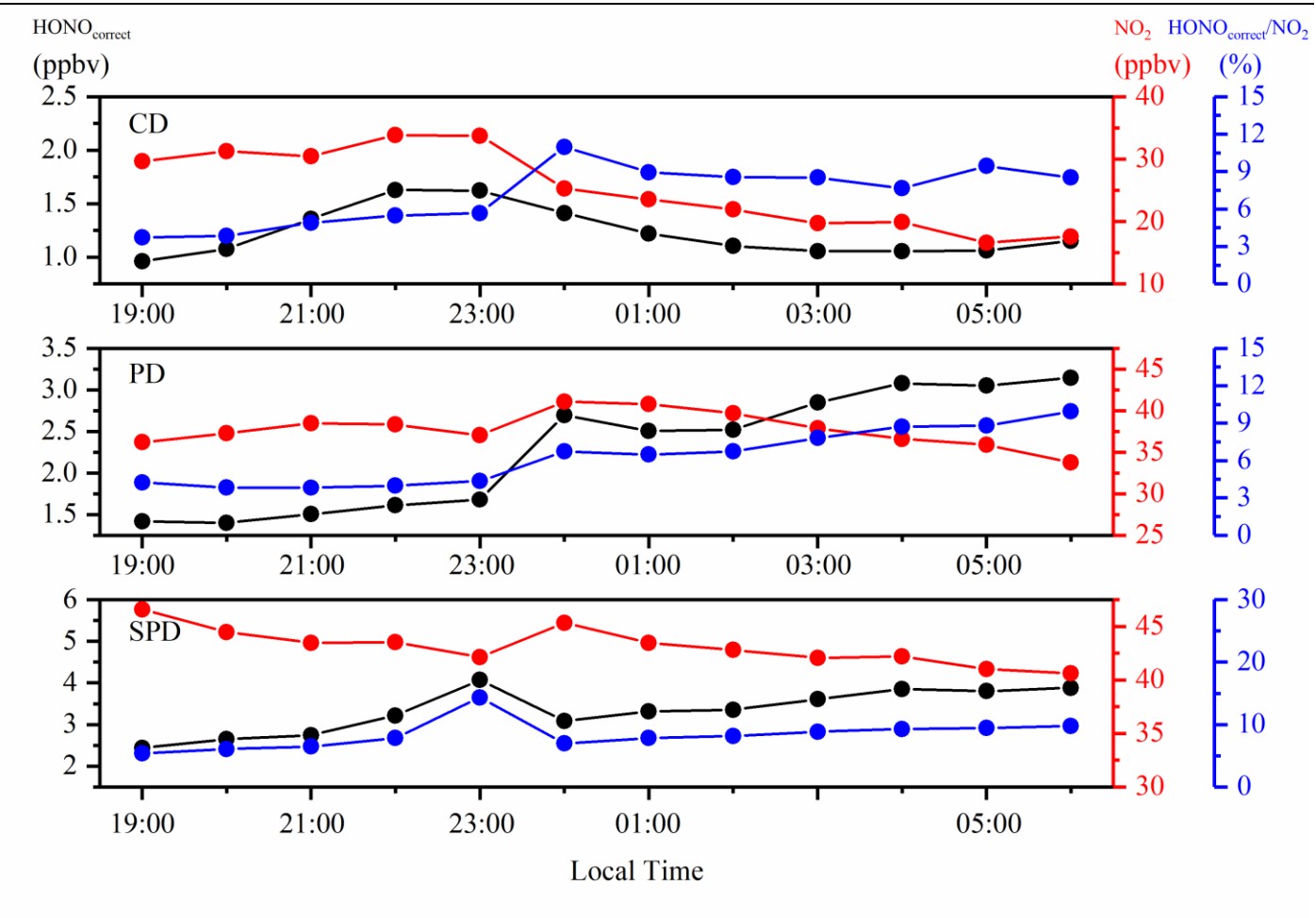

**Fig. 8.** Nocturnal variations of HONO$_{correct}$, NO$_2$, and HONO$_{correct}$/NO$_2$ in CD, PD and SPD periods.

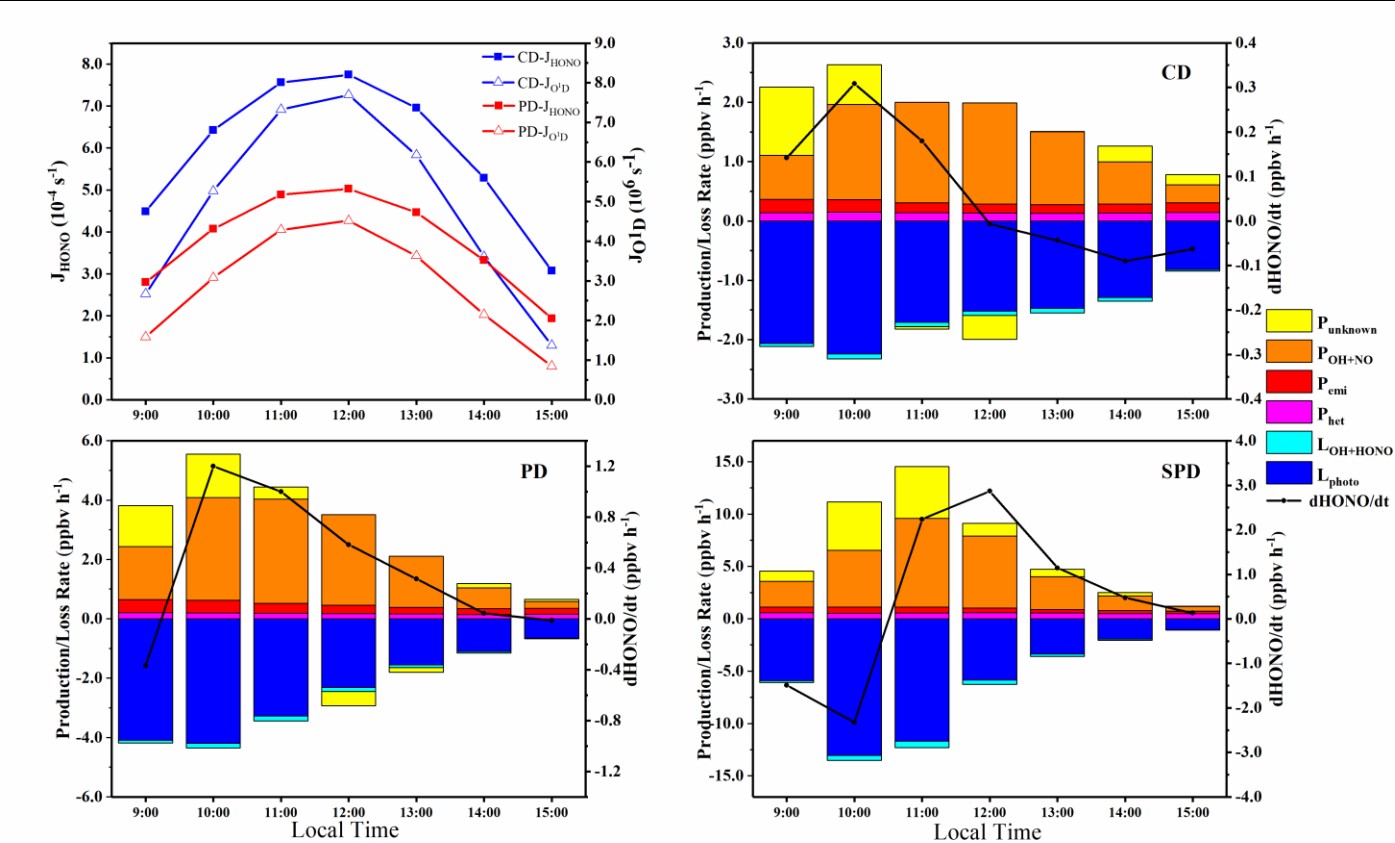

**Fig. 9**. The average profiles of $J_{HONO}$ and $J_O{}^1{}_D$ concentrations during the daytime, and production and loss rate of the daytime HONO in CD, PD and SPD periods.

**Table Captions:**

Table 1. Data statistics of HONO, $PM_{2.5}$, $NO_2$, NO, $NO_X$, $HONO/NO_2$, $HONO/NO_X$, $O_3$, CO, T, RH, and WS during the measurement period, mean value ± standard deviation.

Table 2. Comparisons of the daytime and nighttime HONO level, $HONO/NO_2$, and $HONO/NO_X$ mean values in Zhengzhou and other sites around the world.

**Table 1.**

Data statistics of HONO, PM$_{2.5}$, NO$_2$, NO, NO$_X$, HONO/NO$_2$, HONO/NO$_X$, O$_3$, CO, T, RH, and WS during the measurement period, mean value ± standard deviation.

| Trace gases | CD | | | PD | | | SPD | | | Total days |
|---|---|---|---|---|---|---|---|---|---|---|
| | Day | Night | All | Day | Night | All | Day | Night | All | |
| PM$_{2.5}$ (µg m$^{-3}$) | 37 ± 15 | 41 ± 17 | 39 ± 16 | 80 ± 32 | 93 ± 46 | 87 ± 40 | 148 ± 29 | 147 ± 33 | 147 ± 31 | 91 ± 54 |
| HONO (ppbv) | 0.9 ± 0.7 | 1.4 ± 0.7 | 1.1 ± 0.7 | 1.9 ± 1.7 | 2.7 ± 1.3 | 2.3 ± 1.5 | 3.5 ± 2.7 | 4.0 ± 1.1 | 3.7 ± 2.1 | 2.5 ± 1.9 |
| CO (ppmv) | 1 ± 0.3 | 1 ± 0.3 | 1 ± 0.3 | 1± 0.4 | 1 ± 0.6 | 1 ± 0.5 | 2 ± 0.6 | 2 ± 0.4 | 2 ± 0.5 | 1 ± 0.6 |
| NO (ppbv) | 18.4 ± 39.3 | 15 ± 34.3 | 16.7 ± 36.8 | 20.3 ± 26.2 | 30.7 ± 33.6 | 25.5 ± 30.4 | 40.8 ± 50.8 | 64.3 ± 82.1 | 52.5 ± 68.9 | 31.8 ± 51.4 |
| NO$_2$ (ppbv) | 23 ± 13 | 26 ± 13 | 25 ± 13 | 29 ± 9 | 38 ± 10 | 33 ± 11 | 40 ± 11 | 43 ± 10 | 42 ± 11 | 33 ± 14 |
| O$_3$ (ppbv) | 21.4 ± 11.5 | 13.8 ± 10.0 | 17.6 ± 11.4 | 17.4 ± 11.9 | 8.9 ± 8.1 | 13.1 ± 10.9 | 15.6 ± 14.2 | 7.9 ± 7.1 | 11.8 ± 11.8 | 14.2 ± 11.7 |
| HONO/NO$_2$ (%) | 4.2 ± 3.6 | 5.3 ± 2.2 | 4.7 ± 3.1 | 6.8 ± 5.8 | 7.4 ± 3.9 | 7.1 ± 4.9 | 9.0 ± 7.7 | 9.8 ± 5.8 | 9.4 ± 6.8 | 7.6 ± 6.4 |
| HONO/NO$_X$ (%) | 3.3 ± 2.7 | 6.0 ± 5.6 | 4.5 ± 4.5 | 4.4 ± 2.5 | 4.6 ± 1.7 | 4.5 ± 2.1 | 5.3 ± 3.4 | 5.8 ± 4.7 | 5.6 ± 4.1 | 4.9 ± 3.8 |
| RH (%) | 30 ± 21 | 36 ± 20 | 33 ± 21 | 44 ± 17 | 54 ± 18 | 49 ± 18 | 64 ± 18 | 73 ± 13 | 68 ± 16 | 50 ± 24 |
| WS (m s$^{-1}$) | 0.8 ± 1.0 | 0.5 ± 0.7 | 0.7 ± 0.9 | 1.1 ± 1.4 | 0.6 ± 0.9 | 0.9 ± 1.2 | 0.4 ± 0.7 | 0.3 ± 0.6 | 0.4 ± 0.7 | 0.6 ± 0.9 |
| T (℃) | 4.3 ± 4.6 | 2.7 ± 3.6 | 3.5 ± 4.2 | 3.7 ± 3.3 | 2.6 ± 3.1 | 3.1 ± 3.2 | 4.6 ± 3.2 | 2.9 ± 2.1 | 3.8 ± 2.8 | 3.5 ± 3.5 |

**Table 2.**

Comparisons of the daytime and nighttime HONO level, HONO/NO$_2$, and HONO/NO$_X$ mean values in Zhengzhou and other sites around the world.

| Date (Site) | Instrument | HONO (ppbv) | | | HONO/NO$_2$ (%) | | HONO/NO$_X$ (%) | | Reference |
|---|---|---|---|---|---|---|---|---|---|
| | | Day | Night | N/D | Day | Night | Day | Night | |
| Oct.–Nov. 2014 (Beijing, urban) | LOPAP (long path absorption photometer) | 0.9 | 1.8 | 2.0 | 2.6 | 4.6 | 1.7 | 2.2 | Tong et al., 2015 |
| Feb.–Mar. 2014 (Beijing, urban) | LOPAP | 1.8 | 2.1 | 1.2 | 3.8 (Severe haze) | 4.3 | 2.5 | 2.5 | Hou et al., 2016 |
| | | 0.5 | 0.9 | 1.8 | 7.8 (Clean) | 3.0 | 5.1 | 2.4 | |
| Jul. 2006 (Guangzhou, rural) | LOPAP | 0.2 | 0.9 | 4.5 | 1.0 | 2.5 | 4.3 | 4.5 | Li et al., 2012 |
| Jul. 2014–Aug. 2015 (Xi'an, urban) | LOPAP | 0.5 | 1.6 | 3.2 | 3.3 | 6.2 | | | Huang et al., 2017 |
| Aug. 2010–Jun. 2012 (Shanghai, urban) | Active DOAS | 0.8 | 1.1 | 1.4 | 4.2 | 4.5 | | | Wang et al., 2013 |
| Jul. 2009 (Paris, urban) | wet chemical derivatization technique-HPLC/UV-VIS detection | 0.1 | 0.2 | 2.0 | 3.3 | 2.5 | | | Michoud et al., 2014 |
| Jan. 2019 | AIM | 2.2 | 2.8 | 1.3 | 6.8 | 8.5 | 4.4 | 5.5 | This study |