# Peer review of "Characteristics, sources, and reactions of nitrous acid during"

_Atmospheric Chemistry and Physics, 2019_

## Referee Comment (RC1) · Anonymous Referee #1 · 24 Nov 2019

**Manuscript title:** Characteristics, sources and reactions of nitrous acid during winter in the core city of the Central Plains Economic Region in China via high-time-resolution online measurements

Authors: Hao et al.

https://doi.org/10.5194/acp-2019-916

The paper covers the monitoring results of HONO – classified into three different pollution levels (clean, polluted days, and severe polluted days). The net HONO production rates for both homogeneous and heterogenous reactions were determined. The observed HONO concentrations during daytime and nighttime were explained by the calculated HONO production rates. The cumulative frequency ratio distribution of HONO emission/HONO was used to elucidate the possible HONO source.

The Introduction needs a brief description of your tasks (what tasks?) to achieve your goals (specify goals). All I read is that you will monitor HONO levels along with RH, especially during the night. Rationale for your study is weak (see the following general comments),

Methodology is adequate, but you need to cover uncertainty in all measured pollutants so you have a feel about the correct use of significant figures. Results/discussion section needs rewritten to follow the logical sequence – the format is similar to that of Zhang et al., 2019.

My overall recommendations are major revision (did not read your conclusions).

General comments:

1. In general, the text can be followed. However, there are many awkward, unclear, redundant, unnecessary, ambiguous, and confusing phrases/statements (see the following tech/English comments). A professional expert must edit the text for clarity and for better flow before resubmission.

2. The rationale for your study is weak – need more elaboration. The fact that no study has been performed in in Zhengzhou (L 129) does not justify the novelty of your study. You should have covered the setbacks of previous studies and state those tasks (including tables/figures) currently evaluated have not been properly addressed in previous studies. Unfortunately, I have found none. Anyway, pls emphasize the uniqueness of your study.

3. I have a hard time figuring out that the results from one single sampling site located on the rooftop of a building in Zhengzhou university (L 134) can represent the air quality in general and HONO in particular in the entire

Zhengzhou city (180 million population, L 106). This is why there are so many monitoring stations (traffic, urban and background) within a given large city to reflect the air quality within the city. Pls modify your title as well those in the text.

4. Need to clearly cover HONO sources and sinks as well as homogeneous (NO + $^\bullet$OH) and heterogenous ($NO_2 + H_2O$) in the text (introduction and discussion). For example, the role of ground surface at night for HONO deposit (sink) and the reemission (source) from HONO reservoirs (e.g., soil nitrite), etc. The other sink source, albeit insignificant, is that HONO may react with others to form new compounds, as in the case of reactions with amines in forming nitrosamines. How about transport of HONO? – what is the lifetime in atmosphere (hours under in-door conditions).

5. Need to discuss the uncertainty in your results due to variations of many parameters (e.g., rate constants and OH radical values, L 250-253). This leads to the following comments about the use of significant figures.

6. Be careful about use of significant figures. Delete decimal points for RH (L 182, 185, 187, 194, Table 2, etc.), for $NO_2$ level (L 197-198; also, you compare with standard of 80), for $PM_{2.5}$ (Table 2), for level in µg m$^{-3}$ (L 198, etc.), for ratio (L 34, 304, etc.).

   Use 2 significant figures for rate constants (L 17, 19, 24, 33, 258-260, 266-268, 391-394, p. 16, etc.), considering the uncertainty of all parameters used and average value of OH selected.

   Use one digit after the decimal points for these values (L 271-273). Use 13.4 (L 271)

7. Be consistent with the format of unit. You use m s-1 (or µg m$^{-3}$), yet m/s and µg·m$^{-3}$ are in Table 2 (delete centered dot). Use ppbv throughout the text, but ppbV in Tables 1 and 2 and Fig. 8 (some ppbv and one uses ppbV).

   - Fig. 2: Unit for CO is wrong – should be ppmv.
   - L 428-431: The unit for OH concentration is wrong.
   - L 441: The unit for $P_{unkown}$ is wrong

8. Data need show the variation; use Box plots or error bars in figures and add standard deviation in tables.

9. Need proper citations for equations and rate constants, e.g., in L 247, 250

10. The comparison with others (Table 1) may not be useful and must be made with care since the studied year is different (some in 2012 – may not have adequate end-of-the pipe treatment), nature of sampling sites is different (some in urban, suburban and even rural sites) and atmospheric dynamics in these regions are far different.

11. Why no discussion of OH production rate as a function of $O_3$ levels?? In other words, is any HONO information related to $O_3$ pollution level?? It is relevant since ·OH radical generated from HONO is in turn used for $O_3$ production.

1. Title is misleading.
2. L 15: All of a sudden, the phrase "three sources" pops up. Need to clearly state what they are.
3. Why not place centered dot symbol for OH radical? or ·OH
4. Use a proper term in lieu of rate which refers to time
5. L 45: Why only these two? What about others including HONO itself and acetone?
6. L 49: what reaction? Should be more than one reaction.
7. Citation:
   a. Provide adequate spacing between citations,
      or Hou et al., 2016; Michoud et…. (need a spacing before the 2nd citation)
   b. Make sure all cited references are listed and vice versa. For example, citations in L 55 (L 60) and Table 1 (Elshorbany et al., 2012) are not listed.
   c. Delete redundant citations.  No need to cite twice in the beginning and at the end of the sentence. Pls change throughout the text (L 75, 82, 88, 93, 156, 157, 246, etc.) by delete the second citation.
   d. L 95; et al. (2013); delete the extra comma
   e. Avid excessive self-citations (one is enough, e.g., L 115, 116, etc.)
   f. For the same last name, use the format of Jiang et al., 2018c, 2018e; Liu…. (L 115). Pls change throughout the text.
   g. L 140: why cite this? Need year for this reference.
   h. L 181: why cite this one? Delete
   i. L 313: Acker et al. (2005) reported...
8. L 60: what is 1:1??
9.  L 79: There is no connection between these two sentences. Need a sentence (such as revised R3 reaction) leading to the following sentence.
10. L 123: These two measurements ($PM_{2.5}$ and HONO) cannot clarify the sources, sinks and reactions. Pls reword.

11. L 173: what is uncertainty of AIM? How about MDL for other gases??
12. L 190: Table 1 must come before Table 2. Rearrange the table number.
13. Interesting. You cover all these parameters (L 199-204) shown in Fig. 2, yet Fig. 2 is mentioned several sentences later (L205). The same illogical sequence is found in L 403 which mentions $P_{unknown}$ and $P_{emi,}$ but the equation for these is shown much later (L 436). Also need to cover these rates for estimating daytime HONO budget.
14. L 240: why??
15. L 252: why in reference to Beijing?
16. L 253: Can you calculate OH radical concentration from those discussed later in L 418?
17. P. 11: You are talking about night and mentioned no pollution source near the site. Why all these calculations related to traffic?? Unclear.
18. Pls explain the contradictory statements: important pathway (L 321) and unimportant pathway (L 20) for heterogeneous reaction for HONO formation.
19. Use the stack format for equations in a separate line (L 389); use the solidus format for those within the line (L 402).
20. L 431: Why the same values for both PD and SPD? That means you treat PD and SPD the same.
21. Fig. 1: How could one tell the wind direction?? There is no shade area of black and red color (caption says so). Change in Fig. 2 too.
22. Fig. 6d and RH effect: Was the phenomenon also observed by others?? Who is to say that 77% is the inflection point? Just say "reach a certain high level, HONO…."
23. References:
    - Be consistent!
      Use lowercases for journals (e.g., L 502, 515, etc.)
    - Need periods after all journal abbreviations (e.g., L 502, 515, etc.)
    - Adequate subscript/superscript (e.g., L 535)
    - Use correct journal, or Phy. Chem. Chem Phy., (L 536)
    - Use lowercases for articles (e.g., L 505)
    - L 546 ref is not cited.

Tech/English comments:

1. L 13, 178, 179: Use polluted (not pollution)
2. L 33: Use was (not should be)
3. Be consistent about the use of verb tense; some use past tense in the first part of the sentence with present tense used in the latter part.
4. L 32, 279: Delete the redundant abatement (first one)
5. L 44: delete rate
6. L 54: UV/Vis
7. L 56: result comparison
8. L 67: have discovered

9. L 68-69: How does ratio "account for"? Use "is" 0.1-0.8%
10. Reword awkward L 81, word "radiation" (L 93), confusing (L 108-110).
11. L 99: Use pathway of…: mechanism is the same, but pathway is different.
12. L 107: "Food Production and Modern Agriculture" specified by (not published)
13. L 112: delete "is" and insert "is" ahead of selected
14. L 114: Not Zhengzhou chemical characteristics. Should have written as: chemical characteristics of PM in Zhengzhou.
15. L 121: $PM_{2.5}$ is not chemical
16. L 124: How systematic?
17. L 137: western Fourth-Ring Expressway
18. L 151: Use chemicals (in lieu of substances)
19. L 153: $O_2$ and $N_2$ (not O and N)
20. L 160-161: Clearly state which instrument is for which compound, e.g., 48i for CO measurement.
21. L 167-168: The manual uses the term "should be". But in your statement, you should use "was changed" …."was calibrated".
24. L 171: A space after "≥" sign
25. L 181: specify that it is daily average
26. Delete the first unit (L 182, 185, 188, 206, 301, 317, 360, etc.), first two units (L 301, 449, etc.)
27. L 189: .. north with high WS … Also, how high is high? > 3 m/s, or > 4 m/s? Be specific!
28. L 190: effect of pollutant removal
29. L 202: mean values of what? of all pollutant concentrations?
30. L 209: Is your sampling site in urban area?  You mentioned on the university campus. If so, the comparison is not valid.
31. L 214: No logic about sunrise – previous max HONO values (8-10 am) are way past sunrise. Should have written after 10 am…..
32. L 215: reword the incomplete phrase.
33. L 219: delete again
34. L 220: and the concentration remained the same (Is it true??) until sunrise.
35. L 222: you meant $NO_2$ diffusion?
36. L 228: How is atmospheric migration? Should be "migration of atmospheric airmass".
37. L 248: should be Eq. (1); Eq. (2) is in L 296.
38. L 249, 337, 338, 349, 389, etc.: A space b/a the = sign
39. L 254: Adequate subscript in $k_{OH+bar}$
40. L 274: reword
41. L 279: Use rate (not level)
42. L 280: delete the extra spacing b/a the "/" sign
43. L 307: on the campus
44. L 324: medium of what??
45. L 349-350: wording? cannot see "indicating ...of what most cases"? Why?

46. L 361: sedimentation of what? Do you mean deposit?
47. L 367: Delete "study of the"
48. L 374: decreases after…
49. L 383: in the (not then)
50. L 396: How is rate improvement? Use increase
51. L 402: The expression of …. represents
52. L 411: delete an
53. Table 1: A space before (%), or HONO/NOX (%); before year, or Jun. 2012, Also add SD
54. Fig. 3: Need errors bars. Use the present tense, "represent". Show one example in Box plots so one has an idea about the magnitude of variation.
55. Fig. 4: Need error bars
56. Fig. 6: A space after HONO

---

## Referee Comment (RC2) · Anonymous Referee #2 · 29 Nov 2019

**Manuscript title**: Characteristics, sources and reactions of nitrous acid during winter in the core city of the Central Plains Economic Region in China via high-time-resolution online measurements

Authors: Hao et al.

https://doi.org/10.5194/acp-2019-916

Hao et al. present measurements of nitrous acid using ambient ion monitor as well as HONO budget during January 2019 in Zhengzhou (China). After a very brief description of the experiment and of the measurements of HONO and other compounds (see 1$^{st}$ main comment), the authors classified the campaign into three periods regarding the level of pollution using PM$_{2.5}$ measurements as a parameter for the classification: clean days (CD), pollution days (PD) and severe pollution days (SPD). They then discuss the level of HONO as well as the nocturnal and daytime budget of HONO for these different time period. This reveals high concentrations of HONO for the whole campaign with increasing concentrations while the level of pollution increases. This study also highlights the low contribution of heterogeneous NO$_2$ conversion to the nocturnal HONO budget while the net homogeneous production of HONO at night may have been the main factor for the increase in HONO under high NO$_x$ conditions, even if this conclusion may be revised due to wrong level of OH used in the calculation (see 2$^{nd}$ main comment). In addition, the authors estimate the unknown source of HONO during the three periods revealing important level of missing sources for HONO which increase with the level of pollution. The authors, therefore, point out that further research is needed to identify these missing processes but do not try to give some insight in the identification of these processes (see 4$^{th}$ main comment). In conclusion, this manuscript is within the scope of ACP and will be of interest for the atmospheric community. I therefore recommend publication in ACP but after major revision.

**Main comments**:

1) The description of HONO measurements is too brief and insufficient while all the study rely on it. A detailed and self-sufficient description of the measurement technique for HONO is therefore needed even if it has been described in another study. Estimation of instrumental uncertainties are also lacking.

Furthermore, description of the measurement techniques used for ancillary species should also be given (at least the measurement principle and not only the model and brand of the analyzers).

On the contrary Fig. S1 and S2 does not bring valuable information and should be completed to describe more precisely the measurement principle or should be removed.

2) P10, line 253: $1.0 \times 10^6$ molecules cm$^{-3}$ is very high for nighttime concentrations of OH especially in January. Lelieveld et al. (2016) report nocturnal concentrations of OH between $1.5 \times 10^4$ and $3 \times 10^4$ molecules cm$^{-3}$ for January in the region concerned by the present study

and not $1.0 \times 10^6$ molecules cm$^{-3}$ as stated by the authors. Tan et al. (2018) also found nighttime OH concentrations below $1 \times 10^5$ in Beijing during winter (February). The calculation of $P^{net}_{OH+NO}$ should therefore be corrected using a more realistic OH concentrations. This may change the quantitative and relative contribution of homogeneous reaction to accumulated HONO formation at night. In this case, discussion and conclusion of the article on this point should also be revised consequently.

3) A restructuration of section 3.3 is needed. Indeed all the paragraphs between the beginning of this section and the introductive paragraph for equations 4 to 6 (i.e. from P15, line 402 to P16, line 432) should be moved after these equations (i.e. eq. 4 to 6). Indeed, these paragraphs described the different terms used in the equation 4, 5 and 6 while they do not have been presented yet and this make the reading of this section very confusing.

4) P15, lines 403-404: "$P_{unknown}$ is the production rate by an unknown daytime HONO source". Please explain how $P_{unknown}$ is calculated. Do you assume that dHONO/dt is equal to zero to do so? If it is the case, it should be indicated somewhere.

P17, lines 459-460: "However, further research is needed to analyze the unknown sources of daytime HONO".

Why didn't you do it in this study? A deeper analysis of the processes that may be responsible for the observed unknown HONO production would have been valuable in this study. This further analysis is missing to strengthen the interest of this study for publication.

**Minor comments**:

-P1, line 22: Change "(i.e., the concentration of NO…" for "(i.e., when the concentration of NO…".

-P2, line 32: Change "The hourly abatement level of HONO abatement" for "The hourly level of HONO abatement".

-P2, line 46: Change "OH radical is also an important oxidant" for "OH radical is an important oxidant".

-P2, lines 49-50: "Therefore, reaction changes during pollution can be observed by studying the formation mechanism of HONO".

This sentence is not clear to me. Please clarify it or remove it.

-P2, lines 53-54: "Nitro-Mac" is the name of the instrument but it does not described the technique of measurement. Please replace it by "wet chemical derivatization technique-HPLC/UV-VIS detection".

-P3, line 55: The description of instruments existing for HONO measurements is not exhaustive. Important techniques such as IBBCEAS (e.g. Min et al., 2016; Duan et al., 2018) or CIMS (e.g. Hirokawa et al., 2009 ; Roberts et al., 2010) are missing. Please add them to your list.

-P3, line 72: Change "be absorbed by" for "react with".

-P5, lines 137-138: "The site is close to the West Fourth Ring Road".

How far is it? Please be more precise.

-P6, line 142: "High-Time-resolution instrument".

A temporal resolution of 1h is not what is usually called high time resolution. Please change the title of this section.

-P6, line 153: Change "(e.g., O and N)" for "(e.g., $O_2$ and $N_2$)"

-P7, lines 166-168: "The instrument parts and consumables should be changed regularly during the observation process, and the sampling flow should be calibrated to reduce the negative effect of accessories on sampling".

Could you be more specific? How often these maintenances have been made during the measurement period? What consumables exactly have been changed?

-P7, lines 169-170: "the denuder was replaced every six weeks. Standard anion and cation solutions were prepared every two months".

The measurement period lasted only three weeks from 9 to 31 January 2019. How is it compatible with the frequency of replacement given here and the frequency of calibration? Please clarify.

-P7, line 192: Wind direction is not presented in table 2. Please remove it from the list of parameters presented in table 2.

-P8, line 217: Change "Fig. S3" for "Fig. 3".

The comparison of diurnal variation of HONO during the three period is given in Fig. 3 and not in Fig. S3. Fig. S3 concerns the whole measurement period. Once the modification will be made, there will be no reference in the article to Fig. S3. So please comment this figure in the text or remove it from the supplement.

-P8, lines 217-218: "The NO and $NO_2$ concentration increased in the morning rush hours, decreased rapidly afterward, and remained low in the afternoon."

This statement is not true for $NO_2$ and only right for NO during the CD period but not for the PD and SPD period. Please modify this statement consequently.

-P10, line 251: Change "that cannot be obtained in the measurement" for "that was not measured during the campaign".

-P10, line 253: Wrong unit: please change "$cm^3$ molecule$^{-1}$" for "molecule $cm^{-3}$".

-P11, line 279: Change "the hourly abatement level of HONO abatement" for "the hourly level of HONO abatement".

-P11, lines 278-282: "Second, the hourly abatement level of HONO abatement pathways, except OH + HONO, should be at least 1.47 ppbv h$^{-1}$ (i.e., 13.41 – 1.59 ppbv) / 8 h). The

contributions of other HONO abatement pathways in the current work even exceeded the formation of heterogeneous reactions, similar to a previous study (Spataro et al., 2013)."

If this statement is maintained after the recalculation of $P^{net}_{OH+NO}$ using a more realistic nocturnal OH concentrations, authors should comment on which other losses of HONO can be significant at night (e.g. deposition, heterogeneous losses…). At least, a raw estimation of loss by deposition could be performed to estimate whether it can explain the lacking abatement processes.

-P13, lines 342-344: "The increased HONO in ambient air during the pollution period could have been caused by the comparatively high loading and large particle surface".

The fair correlation between HONO concentrations and $PM_{2.5}$ mass concentrations may also just pinpoint the mainly anthropogenic origins of these two pollutants with high direct or indirect contribution of combustion sources for both of them and not the importance of HONO heterogeneous formation pathways on aerosol surfaces. A correlation between the calculated unknown source of HONO and the $PM_{2.5}$ mass concentrations (as a proxy for aerosol surface even if it is not perfect) would have been more convincing. Authors can probably use the $P_{unknown}$ calculated in section 3.3 to perform this correlation.

-P14, line 383: Change "in then current study" for "in the current study".

-P15, line 393: Change "the conversion rates" for "the averaged conversion rates".

-P15, lines 395-396: Change "The improvement" for "the increase".

-P15, lines 398-399: "the high utilization efficiency of the aerosol surface due to good particle surface properties".

I do not understand this statement. Please clarify and rephrase.

-P15-16, lines 415-418: "the tropospheric ultraviolet and visible (TUV) transfer model of the National Center for Atmospheric Research (http://cprm.acom.ucar.edu/Models/TUV/Interactive_TUV/) (Hou et al.,2016) was used to calculate the $J_{HONO}$ value".

It should be addressed that the $J_{HONO}$ values obtained this way are only suitable for clear sky days without clouds, unless the presence of clouds have been taken into account. If so, the method used should be described. Furthermore, the values for $O_3$ column as well as for the surface albedo used in TUV model should be indicated and justification about the choice of these values should be given.

-P16, lines 418-419: "The concentration of OH radicals was calculated with the formulas of $NO_2$, $O_3$, and $JO^1D$".

Please specify the equation used for OH calculation.

-P16, line 427: "The mean values of JHONO and OH radical concentration".

 Is it daily mean or mean values at noon? Please specify this.

-P17, lines 454-455: "Although the values of $P_{OH+NO}$ had high uncertainty because of the NO concentrations".

How NO concentrations can affect largely the uncertainties of $P_{OH+NO}$ calculations? Does NO measurements suffer from high uncertainties? Why? If this is the case this point should be also addressed in the section 2.2. Please clarify this statement.

-Fig. 8: Please modify the legend of the figure to be consistent with the title and the manuscript (use PD and SPD instead of HD and SHD). Furthermore, $J_{HONO}$ and $J_O{}^1{}_D$ are shown only for two periods and not for all three. Why? Please include the values for the third period (SPD) or explain why it is not shown.

-Table 2: Please remove WD from the title of the table since no data of wind direction is shown in it.

Reference:

Duan, J., Qin, M., Ouyang, B., Fang, W., Li, X., Lu, K., Tang, K., Liang, S., Meng, F., Hu, Z., Xie, P., Liu, W., and Häsler, R.: Development of an incoherent broadband cavity-enhanced absorption spectrometer for in situ measurements of HONO and NO2, Atmos. Meas. Tech., 11, 4531–4543, https://doi.org/10.5194/amt-11-4531-2018, 2018.

Hirokawa, J., Kato, T., and Mafune, F.: In situ measurements of atmospheric nitrous acid by chemical ionization mass spectrometry using chloride ion transfer reactions, Anal. Chem., 81, 8380– 8386, 2009.

Hou, S., Tong, S., Ge, M., and An, J.: Comparison of atmospheric nitrous acid during severe haze and clean periods in Beijing, China, Atmos. Environ., 124, 199–206, https://doi.org/10.1016/j.atmosenv.2015.06.023, 2016.

Lelieveld, J., Gromov, S., Pozzer, A., and Taraborrelli, D.: Global tropospheric hydroxyl distribution, budget and reactivity, Atmos. Chem. Phys., 16, 12477–12493, https://doi.org/10.5194/acp-16-12477-2016, 2016.

Min, K.-E., Washenfelder, R. A., Dubé, W. P., Langford, A. O., Edwards, P. M., Zarzana, K. J., Stutz, J., Lu, K., Rohrer, F., Zhang, Y., and Brown, S. S.: A broadband cavity enhanced absorption spectrometer for aircraft measurements of glyoxal, methylglyoxal, nitrous acid, nitrogen dioxide, and water vapor, Atmos. Meas. Tech., 9, 423–440, https://doi.org/10.5194/amt-9-423-2016, 2016.

Roberts, J. M., Veres, P., Warneke, C., Neuman, J. A., Washenfelder, R. A., Brown, S. S., Baasandorj, M., Burkholder, J. B., Burling, I. R., Johnson, T. J., Yokelson, R. J., and de Gouw, J.: Measurement of HONO, HNCO, and other inorganic acids by negative-ion proton-transfer chemical-ionization mass spectrometry (NI-PT-CIMS): application to biomass burning emissions, Atmos. Meas. Tech., 3, 981–990, https://doi.org/10.5194/amt-3-981-2010, 2010.

Tan, Z., Rohrer, F., Lu, K., Ma, X., Bohn, B., Broch, S., Dong, H., Fuchs, H., Gkatzelis, G. I., Hofzumahaus, A., Holland, F., Li, X., Liu, Y., Liu, Y., Novelli, A., Shao, M., Wang, H., Wu, Y., Zeng, L., Hu, M., Kiendler-Scharr, A., Wahner, A., and Zhang, Y.: Wintertime photochemistry in Beijing: observations of ROx radical concentrations in the North China Plain during the BEST-ONE campaign, Atmos. Chem. Phys., 18, 12391–12411, https://doi.org/10.5194/acp-18-12391-2018, 2018.

---

## Author Comment (AC1) · 29 Jan 2020

**Itemized Response to Anonymous Referee #1's Comments**

**Ms. Ref. No.**: acp-2019-916

**Title:** Characteristics, sources, and reactions of nitrous acid during winter at an urban site in the Central Plains Economic Region in China

**Response to Anonymous Referee #1:**

We have carefully addressed your comments on our manuscript and made necessary revisions of the previous manuscript. We sincerely thank you for valuable and constructive inputs. We believe that we have adequately addressed all of your comments and thus the current version has been greatly improved with those valuable comments and further English editing. The revised phrases/sentences/paragraphs are shown in the line number of the revised text.

The followings are our itemized replies to your comments.

**General comments:**

1. In general, the text can be followed. However, there are many awkward, unclear, redundant, unnecessary, ambiguous, and confusing phrases/statements (see the following tech/English comments). A professional expert must edit the text for clarity and for better flow before resubmission.

> Response: We sent the manuscript to a professional expert to enhance the readability of the manuscript. The revised portion has been highlighted with yellow color in the revised version (see the response of the following tech/English comments).

2. The rationale for your study is weak – need more elaboration. The fact that no study has been performed in Zhengzhou (L 129) does not justify the novelty of your study. You should have covered the setbacks of previous studies and state those tasks (including tables/figures) currently evaluated have not been properly addressed in previous studies. Unfortunately, I have found none. Anyway, pls emphasize the

uniqueness of your study.

Response: For your comment, we add the sentences in the revised text.

L 134-147: Many papers (Huang et al., 2017; Tong et al., 2016) took $PM_{2.5}$ as the main control factor of HONO, and studied the differences of HONO sources and characteristics between clean and polluted periods. Homogeneous reaction, direct emission, heterogeneous reaction, and daytime budget analysis were conducted during the period of worsening pollution (namely HD period in this paper). Total $NO_X$ emissions in cities with different leading factors of emissions have been declining year by year due to Chinese government emission control measures, but some Chinese cities are still in high-$NO_X$ areas (e.g. Beijing, Shanghai, Guangzhou, and Zhengzhou.) (Kim et al., 2015; Liu et al., 2017). Under high-$NO_X$ conditions, some papers (Cui et al., 2018; Hou et al., 2016) suggested that heterogeneous reaction was the main source of HONO and did not conduct a quantitative analysis of homogeneous reaction, especially in winter. So, we explore relevant studies of homogeneous reactions. In addition, the source contributions of HONO at night varied with the degree of pollution level were not explained.

3. I have a hard time figuring out that the results from one single sampling site located on the rooftop of a building in Zhengzhou university (L 134) can represent the air quality in general and HONO in particular in the entire Zhengzhou city (180 million population, L 106). This is why there are so many monitoring stations (traffic, urban and background) within a given large city to reflect the air quality within the city. Pls modify your title as well those in the text.

Response: Thank you for your comment. As you say, one single sampling site cannot represent the air quality in general and HONO in the entire Zhengzhou city. We have revised the title, "Characteristics, sources, and reactions of nitrous acid during winter at an urban site in the Central Plains Economic Region in China".

4. Need to clearly cover HONO sources and sinks as well as homogeneous (NO+ •OH) and heterogenous ($NO_2$ + $H_2O$) in the text (introduction and discussion). For example, the role of ground surface at night for HONO deposit (sink) and the reemission (source) from HONO reservoirs (e.g., soil nitrite), etc. The other sink source, albeit insignificant, is that HONO may react with others to form new compounds, as in the case of reactions with amines in forming nitrosamines. How about transport of HONO? – what is the lifetime in atmosphere (hours under in-door conditions).

Response: OK. We added the sentence in the introduction in the revised text.

L 51-54: Therefore, the changes in the contribution of the homogeneous reaction, heterogeneous conversion, and direct emission during pollution can be observed by studying the formation mechanism of HONO.

In the discussion, we thought that the contribution of soot surface to HONO production is usually much lower than expected because the uptake efficiency of $NO_2$ decreases with the prolonged reaction time caused by surface deactivation. The aerosol surface is an important medium for the heterogeneous transformation from $NO_2$ to HONO (Liu et al., 2014). So, we added the sentences in the revised text.

L 364-368: the contribution of soot surface to HONO production is usually much lower than expected because the uptake efficiency of $NO_2$ decreases with the prolonged reaction time caused by surface deactivation. The aerosol surface is an important medium for the heterogeneous transformation from $NO_2$ to HONO (Liu et al., 2014).

We have clearly known that other HONO sources are not the main HONO sources and sinks: 1. HONO is formed by $NO_2$ through the photolysis of sooty surface and adsorbed nitric acid and nitrate at UV wavelengths (Kleffmann et al., 1999). 2. The homogeneous nucleation of $NO_2$, $H_2O$, and $NH_3$ is the HONO formation pathway (Zhang and Tao, 2010). 3. HONO can deposit and react with amines in forming nitrosamines (Li et al., 2012). So, we added the sentences in the revised

text.

L 98-102: The unknown sources of HONO may include the $NO_2$ photolysis of sooty surface and adsorbed nitric acid and nitrate at UV wavelengths (Kleffmann et al., 1999). The homogeneous nucleation of $NO_2$, $H_2O$, and $NH_3$ is the HONO formation pathway (Zhang and Tao, 2010). In the meanwhile, HONO can deposit and react with amines in forming nitrosamines (Li et al., 2012) for sinking.

We knew that the lifetime of HONO was 10–20 min at daytime (Lu et al., 2018) and the estimated lifetime was about 3.3 h in the nighttime (Nie et al., 2015).

5. Need to discuss the uncertainty in your results due to variations of many parameters (e.g., rate constants and OH radical values, L 250-253). This leads to the following comments about the use of significant figures.

Response: Measured species and performance of the instruments are counted in **Table S1**. The rate constants are learned from the study (Atkinson et al., 2004). We don't know the uncertainty of rate constants.

**Table S1.**
Measured species and performance of the instruments.

| Species | Measurement technique | Detection limit | Accuracy |
|---------|----------------------|-----------------|----------|
| $PM_{2.5}$ | Tapered Element Oscillating Microbalance | 1.5 μg m$^{-3}$ | ± 5% |
| HONO | Ion Chromatography | 4 pptv | ± 20% |
| CO | Absorbs Infrared Radiation | 40 ppbv | ± 5% |
| NO | Chemiluminescence | 60 pptv | ± 20% |
| $NO_2$ | Chemiluminescence | 300 pptv | ± 20% |
| $O_3$ | UV Photometry | 0.5 ppbv | ± 5% |

The results came from instrument manufacturers.

Hence, we assumed ± 50% ·OH values to estimate the uncertainty of $P^{net}_{OH+NO}$. The ·OH values of $1.25 \times 10^5$ and $3.75 \times 10^5$ molecule cm$^{-3}$ were calculated the $P^{net}_{OH+NO}$ values of 0.16 and 0.49 ppbv h$^{-1}$.

L 295-297: We assumed $\pm 50\%$ ·OH values to estimate the uncertainty of $P^{net}_{OH+NO}$. The ·OH values of $1.25\times10^5$ and $3.75\times10^5$ molecule cm$^{-3}$ were calculated the $P^{net}_{OH+NO}$ values of 0.16 and 0.49 ppbv h$^{-1}$.

6. Be careful about use of significant figures. Delete decimal points for RH (L 182, 185, 187, 194, Table 2, etc.), for NO2 level (L 197-198; also, you compare with standard of 80), for PM2.5 (Table 2), for level in µg m-3 (L 198, etc.), for ratio (L 34, 304, etc.).

Response: Thank you for the comment. We modified the problem in **Table 1** and revised the text. We have learned how to use significant figures.

**Table 1.**
Data statistics of HONO, PM$_{2.5}$, NO$_2$, NO, NO$_X$, HONO/NO$_2$, HONO/NO$_X$, O$_3$, CO, T, RH, and WS during the measurement period, mean value $\pm$ standard deviation.

| Trace gases | CD | | | PD | | | SPD | | | Total days |
| --- | --- | --- | --- | --- | --- | --- | --- | --- | --- | --- |
| | Day | Night | All | Day | Night | All | Day | Night | All | |
| PM$_{2.5}$ (µg·m$^{-3}$) | 37±15 | 41±17 | 39±16 | 80±32 | 93±46 | 87±40 | 148±29 | 147±33 | 147±31 | 91±54 |
| HONO (ppbv) | 0.9±0.7 | 1.4±0.7 | 1.1±0.7 | 1.9±1.7 | 2.7±1.3 | 2.3±1.5 | 3.5±2.7 | 4.0±1.1 | 3.7±2.1 | 2.5±1.9 |
| CO (ppmv) | 1±0.3 | 1±0.3 | 1±0.3 | 1±0.4 | 1±0.6 | 1±0.5 | 2±0.6 | 2±0.4 | 2±0.5 | 1±0.6 |
| NO (ppbv) | 18.4±39.3 | 15±34.3 | 16.7±36.8 | 20.3±26.2 | 30.7±33.6 | 25.5±30.4 | 40.8±50.8 | 64.3±82.1 | 52.5±68.9 | 31.8±51.4 |
| NO$_2$ (ppbv) | 23±13 | 26±13 | 25±13 | 29±9 | 38±10 | 33±11 | 40±11 | 43±10 | 42±11 | 33±14 |
| O$_3$ (ppbv) | 21.4±11.5 | 13.8±10.0 | 17.6±11.4 | 17.4±11.9 | 8.9±8.1 | 13.1±10.9 | 15.6±14.2 | 7.9±7.1 | 11.8±11.8 | 14.2±11.7 |
| HONO/NO$_2$ (%) | 4.2±3.6 | 5.3±2.2 | 4.7±3.1 | 6.8±5.8 | 7.4±3.9 | 7.1±4.9 | 9.0±7.7 | 9.8±5.8 | 9.4±6.8 | 7.6±6.4 |
| HONO/NO$_X$ (%) | 3.3±2.7 | 6.0±5.6 | 4.5±4.5 | 4.4±2.5 | 4.6±1.7 | 4.5±2.1 | 5.3±3.4 | 5.8±4.7 | 5.6±4.1 | 4.9±3.8 |
| RH (%) | 30±21 | 36±20 | 33±21 | 44±17 | 54±18 | 49±18 | 64±18 | 73±13 | 68±16 | 50±24 |
| WS (m·s$^{-1}$) | 0.8±1.0 | 0.5±0.7 | 0.7±0.9 | 1.1±1.4 | 0.6±0.9 | 0.9±1.2 | 0.4±0.7 | 0.3±0.6 | 0.4±0.7 | 0.6±0.9 |
| T (°C) | 4.3±4.6 | 2.7±3.6 | 3.5±4.2 | 3.7±3.3 | 2.6±3.1 | 3.1±3.2 | 4.6±3.2 | 2.9±2.1 | 3.8±2.8 | 3.5±3.5 |

Delete decimal points for RH (L182, 185, 187, 194, Table 2, etc.),

Response:

L 205: …with RH ranging from 5 to 79%...

L 207: …RH ranging from 17 to 86%...

L 210: …RH ranging from 30 to 96%...

L 217: …RH in CD, PD, and SPD periods was 33, 49, and 68%...

level in µg m-3 (L 198,etc.),

Response:

L 221: …values of $NO_2$ were 25, 33, and 42 ppbv (46, 63, and 78 $\mu g\ m^{-3}$ lower than…

for ratio (L 34, 304, etc.).

Response:

L 34: …HONO ratio (less than 20%) was approximately 77%...

L 343: …$HONO_{emission}$/HONO ratio less than 20% was approximately 77%...

7. Be consistent with the format of unit. You use m s-1 (or µg m-3), yet m/s and µg·m-3 are in Table 2 (delete centered dot). Use ppbv throughout the text, but ppbV in Tables 1 and 2 and Fig. 8 (some ppbv and one uses ppbV).

Response: Sorry for the careless. The problems have been revised. And we checked the full text to avoid the problems.

**Table 1.**

Data statistics of HONO, PM$_{2.5}$, NO$_2$, NO, NO$_X$, HONO/NO$_2$, HONO/NO$_X$, O$_3$, CO, T, RH, and WS during the measurement period, mean value ± standard deviation.

| Trace gases | CD | | | PD | | | SPD | | | Total days |
|---|---|---|---|---|---|---|---|---|---|---|
| | Day | Night | All | Day | Night | All | Day | Night | All | |
| PM$_{2.5}$ (µg·m$^{-3}$) | 37±15 | 41±17 | 39±16 | 80±32 | 93±46 | 87±40 | 148±29 | 147±33 | 147±31 | 91±54 |
| HONO (ppbv) | 0.9±0.7 | 1.4±0.7 | 1.1±0.7 | 1.9±1.7 | 2.7±1.3 | 2.3±1.5 | 3.5±2.7 | 4.0±1.1 | 3.7±2.1 | 2.5±1.9 |
| CO (ppmv) | 1±0.3 | 1±0.3 | 1±0.3 | 1±0.4 | 1±0.6 | 1±0.5 | 2±0.6 | 2±0.4 | 2±0.5 | 1±0.6 |
| NO (ppbv) | 18.4±39.3 | 15±34.3 | 16.7±36.8 | 20.3±26.2 | 30.7±33.6 | 25.5±30.4 | 40.8±50.8 | 64.3±82.1 | 52.5±68.9 | 31.8±51.4 |
| NO$_2$ (ppbv) | 23±13 | 26±13 | 25±13 | 29±9 | 38±10 | 33±11 | 40±11 | 43±10 | 42±11 | 33±14 |
| O$_3$ (ppbv) | 21.4±11.5 | 13.8±10.0 | 17.6±11.4 | 17.4±11.9 | 8.9±8.1 | 13.1±10.9 | 15.6±14.2 | 7.9±7.1 | 11.8±11.8 | 14.2±11.7 |
| HONO/NO$_2$ (%) | 4.2±3.6 | 5.3±2.2 | 4.7±3.1 | 6.8±5.8 | 7.4±3.9 | 7.1±4.9 | 9.0±7.7 | 9.8±5.8 | 9.4±6.8 | 7.6±6.4 |
| HONO/NO$_X$ (%) | 3.3±2.7 | 6.0±5.6 | 4.5±4.5 | 4.4±2.5 | 4.6±1.7 | 4.5±2.1 | 5.3±3.4 | 5.8±4.7 | 5.6±4.1 | 4.9±3.8 |
| RH (%) | 30±21 | 36±20 | 33±21 | 44±17 | 54±18 | 49±18 | 64±18 | 73±13 | 68±16 | 50±24 |
| WS (m·s$^{-1}$) | 0.8±1.0 | 0.5±0.7 | 0.7±0.9 | 1.1±1.4 | 0.6±0.9 | 0.9±1.2 | 0.4±0.7 | 0.3±0.6 | 0.4±0.7 | 0.6±0.9 |
| T (°C) | 4.3±4.6 | 2.7±3.6 | 3.5±4.2 | 3.7±3.3 | 2.6±3.1 | 3.1±3.2 | 4.6±3.2 | 2.9±2.1 | 3.8±2.8 | 3.5±3.5 |

[Figure]

**Fig. 9**. The average profiles of $J_{HONO}$ and $J_O{}^1{}_D$ concentrations during the daytime, and production and loss rate of the daytime HONO in CD, PD and SPD periods.

8. Data need show the variation; use Box plots or error bars in figures and add standard deviation in tables.

Response: OK. With too many error bars in figures, it will make the figures look unclear. So the error bars of **Fig. 4**, **Fig. 5**, and **Fig. 8** were placed separately in the tables of the supplement (**Table S2**, **Table S3**, and **Table S4**).

9. Need proper citations for equations and rate constants, e.g., in L 247, 250

Response: Sorry, it was our error of citations. Proper citations have been added.
L 278: At T = 298 K and P = 101 kPa… respectively (Atkinson et al., 2004; Sander et al., 2003).

10. The comparison with others (Table 1) may not be useful and must be made with care since the studied year is different (some in 2012 – may not have adequate end-of-the pipe treatment), nature of sampling sites is different (some in urban, suburban and even rural sites) and atmospheric dynamics in these regions are far different.

Response: You're right. The comparison with others must be made with care since the studied year is different. Therefore, only the observation data of HONO in the last ten years were used for comparative analysis of HONO concentration changes in urban. We removed the HONO level (Jun.-Jul. 2005 in Germany) in the revised text (**Table 2**). We analyzed the effect of the site on HONO concentration in urban. So, we added sampling sites in **Table 2**.

**Table 2.**

Comparisons of the daytime and nighttime HONO level, HONO/NO$_2$, and HONO/NO$_X$ mean values in Zhengzhou and other sites around the world.

| Date (Site) | Instrument | HONO (ppbv) | | | HONO/NO$_2$ (%) | | HONO/NO$_X$ (%) | | Reference |
|---|---|---|---|---|---|---|---|---|---|
| | | Day | Night | N/D | Day | Night | Day | Night | |
| Oct.–Nov. 2014 (Beijing, urban) | LOPAP (long path absorption photometer) | 0.9 | 1.8 | 2.0 | 2.6 | 4.6 | 1.7 | 2.2 | Tong et al., 2015 |
| Feb.–Mar. 2014 (Beijing, urban) | LOPAP | 1.8 | 2.1 | 1.2 | 3.8 (Severe haze) | 4.3 | 2.5 | 2.5 | Hou et al., 2016 |
| | | 0.5 | 0.9 | 1.8 | 7.8 (Clean) | 3.0 | 5.1 | 2.4 | |
| Jul. 2006 (Guangzhou, rural) | LOPAP | 0.2 | 0.9 | 4.5 | 1.0 | 2.5 | 4.3 | 4.5 | Li et al., 2012 |
| Jul. 2014–Aug. 2015 (Xi'an, urban) | LOPAP | 0.5 | 1.6 | 3.2 | 3.3 | 6.2 | | | Huang et al., 2017 |
| Aug. 2010–Jun. 2012 (Shanghai, urban) | Active DOAS | 0.8 | 1.1 | 1.4 | 4.2 | 4.5 | | | Wang et al., 2013 |
| Jul. 2009 (Paris, urban) | wet chemical derivatization technique-HPLC/UV-VIS detection | 0.1 | 0.2 | 2.0 | 3.3 | 2.5 | | | Michoud et al., 2014 |
| Jan. 2019 | AIM | 2.2 | 2.8 | 1.3 | 6.8 | 8.5 | 4.4 | 5.5 | This study |

Jul. 2006 (Guangzhou, urban): This paper is a relatively early study on the HONO concentration level in China. We thought this paper still had a certain level of high

quality, so we put it in the table for comparison.

For avoiding this problem of atmospheric dynamics, we used observed values of HONO in Chinese cities mostly.

11. Why no discussion of OH production rate as a function of $O_3$ levels?? In other words, is any HONO information related to $O_3$ pollution level?? It is relevant since $\cdot$OH radical generated from HONO is in turn used for $O_3$ production.

Response: Thank you for the comment. We hold the opinion that the discussion of OH production rate as a function of $O_3$ levels can be written as an article on account of complexity. At the same time, we found that the negative correlation between HONO and $O_3$ was lower in the entire period, and it was known from **Fig 4d** that the heavier the pollution level, the lower the $O_3$ concentration.

[Figure]

**Figure** The correlation between HONO and $O_3$.

[Figure]

**Fig. 4d.** Diurnal variations of $O_3$. The blue points and lines represented the CD period; the black points and lines represented the PD period; the red points and lines represented the SPD period.

**Specific comments:**

1. "Title is misleading"

Response: OK. We have added "at an urban site" in the title. The title is "Characteristics, sources, and reactions of nitrous acid during winter at an urban site in the Central Plains Economic Region in China."

2. L 15: All of a sudden, the phrase "three sources" pops up. Need to clearly state what they are.

Response: According to your comment, the sentence was modified in the revised text.

L 14-15: The contribution of the homogeneous reaction, heterogeneous conversion, and direct emission to HONO sources varied under different pollution levels.

3. Why not place centered dot symbol for OH radical? or •OH

Response: OK. We have changed ·OH for ·OH in the revised text.

L 44: …HONO to ·OH concentration…

L 47: …most important primary source of ·OH…

L 77: …react with the ·OH…

L 96: …the importance of the ·OH…

L 280: …concentration of ·OH…

L 282: …the average concentration of ·OH…

L 283: …the same ·OH…

L 471: …·OH and NO were formed…

L 495: …values of $J_{HONO}$ and ·OH concentration…

4. Use a proper term in lieu of rate which refers to time

Response: Sorry for the careless. We have removed the word "rate".

L 44: …the contribution of HONO to ·OH concentration can reach 25−50%…"

5. L 45: Why only these two? What about others including HONO itself and acetone?

Response: We have modified the sentence in the revised text.

L 43-46: …the contribution of HONO to ·OH concentration can reach 25%−50%, especially when the concentration of OH radicals produced by the photolysis of ozone, acetone, and formaldehyde…

6. L 49: what reaction? Should be more than one reaction.

Response: Sorry for my confusion. We have changed "reaction" for "the changes in the contribution of the homogeneous reaction, heterogeneous conversion, and direct emission" in the revised text.

L 51-54: …the changes in the contribution of the homogeneous reaction, heterogeneous conversion, and direct emission during pollution can be observed…

7. Citation:

Response: Sorry for my carelessness. All modifications have been made, and we have checked the full text.

a. Provide adequate spacing between citations, or Hou et al., 2016; Michoud et…. (need a spacing before the 2nd citation).

L 57: …(Elshorbany et al., 2012; Winer and Biermann, 1994)…

L 60: …(Duan et al., 2018; Min et al., 2016)…

L 61: …(Hirokawa et al., 2009; Roberts et al., 2010).

L 68: …(Hou et al., 2016; Michoud et al., 2014).

L 70: …(Acker et al., 2005; Grassian, 2001; Kurtenbach et al., 2001).

b. Make sure all cited references are listed and vice versa. For example, citations in L 55 (L 60) and Table 1 (Elshorbany et al., 2012) are not listed.

L 785: VandenBoer, T. C., Markovic…

L 606: Elshorbany, Y. F., Steil, B., Brühl, C…

c. Delete redundant citations. No need to cite twice in the beginning and at the end of the sentence. Pls change throughout the text (L 75, 82, 88, 93, 156, 157, 246, etc.) by delete the second citation.

L 72: Kurtenbach et al. (2001)…

L 77: Tong et al. (2015)…

L 90: Hao et al. (2006)…

L 96: Su et al. (2008a)…

L 105: Spataro et al. (2013)…

L 108: Tong et al. (2015)…

d. L 95; et al. (2013); delete the extra comma.

L 105: Spataro et al. (2013)…

e. Avid excessive self-citations (one is enough, e.g., L 115, 116, etc.)

f. For the same last name, use the format of Jiang et al., 2018c, 2018e; Liu…. (L 115). Pls change throughout the text.

L 123-127: As the core city of CPER, Zhengzhou characterized by severe PM (particulate matters) pollution (Jiang et al., 2018b), is selected in the study. In recent years, comprehensive PM research has been conducted on the chemical characteristics of PM in Zhengzhou. (Li et al., 2019), source apportionment (Liu et al., 2019), health risks (Jiang et al., 2019), and emission source profiles (Jiang et al., 2018a).

g. L 140: why cite this? Need year for this reference.

L 163-164: …(7:00–18:00 local time)…

h. L 181: why cite this one? Delete

L 204: …Air Quality Standards (CNAAQS) (75 $\mu g\,m^{-3}$)…

i. L 313: Acker et al. (2005) reported...

L 352: …Acker et al. (2005)…

8. L 60: what is 1:1??

Response: Sorry for my confusion. Through the comparison of measurement results, the correlation between SC-AP deployed onsite and AIM tended to 1. For avoiding the confusion, we have removed the words "; the results exhibited a consistency of nearly 1:1" in the revised text.

L 64-66: Compared with HONO measured by SC-AP deployed onsite, HONO measured by AIM has a small error and is within the acceptable analytical uncertainty (VandenBoer et al., 2014).

9. L 79: There is no connection between these two sentences. Need a sentence(such as revised R3 reaction) leading to the following sentence.

Response: OK. We have added the sentence between these two sentences in the revised text.

L 82-87: Such calculations have been applied in studies on homogeneous reactions and daytime budgets (Hou et al., 2016; Huang et al., 2017). These are studies of homogeneous reactions, and some researchers have begun to explore the mechanism of $NO_2$ heterogeneous reactions. Finlayson-Pitts et al. (2003) studied the mechanism of chemical adsorption of $NO_2$ and H ions on the adsorbed surface…

10. L 123: These two measurements ($PM_{2.5}$ and HONO) cannot clarify the sources, sinks and reactions. Pls reword.

Response: Sorry, we have modified the sentence in the revised text.

L 133-134: The levels of $PM_{2.5}$ were divided into three periods to analyze the HONO sources, sinks, and reactions in different periods.

11. L 173: what is uncertainty of AIM? How about MDL for other gases??

Response: Thank you for your comment. Measurement technique, detection limit, and accuracy of measured species are shown in Table S1 in the supplement.

**Table S1.**
Measured species and performance of the instruments.

| Species | Measurement technique | Detection limit | Accuracy |
|---|---|---|---|
| $PM_{2.5}$ | Tapered Element Oscillating Microbalance | 1.5 μg m$^{-3}$ | ± 5% |
| HONO | Ion Chromatography | 4 pptv | ± 20% |
| CO | Absorbs Infrared Radiation | 40 ppbv | ± 5% |
| NO | Chemiluminescence | 60 pptv | ± 20% |
| $NO_2$ | Chemiluminescence | 300 pptv | ± 20% |
| $O_3$ | UV Photometry | 0.5 ppbv | ± 5% |

The results came from instrument manufacturers.

12. L 190: Table 1 must come before Table 2. Rearrange the table number.

Response: OK. We have rearranged the table number.

L 213: **Table 1** lists the data statistics…

L 265: …listed in **Table 2.**

13. Interesting. You cover all these parameters (L 199-204) shown in Fig. 2, yet Fig. 2 is mentioned several sentences later (L205). The same illogical sequence is found in L 403 which mentions $P_{unknown}$ and $P_{emi,}$ but the equation for these is shown much later (L 436). Also need to cover these rates for estimating daytime HONO budget.

Response: Sorry for my confusion. The illogical sequence had been changed. We have modified the sentences in the revised text.

L 223-226: **Fig. 2** shows the concentration changes in HONO…The variations of the average HONO, $PM_{2.5}$, $NO_2$…

L 456-465: …d HONO / d t = sources − sinks

$$= (P_{unknown} + P_{OH+NO} + P_{emi} + P_{het}) - (L_{OH+HONO} + L_{photo}) \qquad (4),$$

$$P_{OH+NO} = k_{OH+NO} [OH] [NO] \qquad (5),$$

$$L_{OH+HONO} = k_{OH+HONO} [OH] [HONO] \tag{6}.$$

The d HONO / d t calculated from…

14. L 240: why??

Response: Sorry for my carelessness. Although observations of HONO levels in Zhengzhou were different from other cities because of periods, seasons, and meteorological conditions, a higher concentration of HONO was found in Zhengzhou. This situation attracted our attention. We thought the environmental impact of the increase in pollutant emissions and the number of vehicles exceeds the efforts of Zhengzhou to protect the atmospheric environment. Total $NO_X$ emissions in cities have been declining year by year due to Chinese government emission control measures, but some Chinese cities are still in high-$NO_X$ areas (e.g. Beijing, Shanghai, Guangzhou, and Zhengzhou.) (Kim et al., 2015; Liu et al., 2017). Therefore, we have added the sentence in the full text.

L 267-269: The reason for this phenomenon is that Zhengzhou is a high-$NO_X$ area which provides HONO with abundant precursors ($NO_2$ and NO) in winter (Kim et al., 2015).

15. L 252: why in reference to Beijing?

Response: We revisit and determine the OH concentration. The OH concentration previously used is $1 \times 10^6$ molecule $cm^{-3}$ is obtained by the simulation (Lelieveld et al., 2016). So, we have modified the sentences in the revised text.

L 280-285: [OH] is the concentration of ·OH that was not measured during the campaign. Therefore, Tan et al. (2018) found that by the field measurement, the average concentration of ·OH in Beijing at nighttime was about $2.5 \times 10^5$ molecule $cm^{-3}$. Moreover, the same ·OH concentration was also used to calculate the

homogeneous reaction of HONO in the recent researches of Beijing (Zhang et al., 2019), Shanghai (Cui et al., 2018), and Xi'an (Huang et al., 2017).

Finally, we chose the average concentration of ·OH in Beijing was about $2.5 \times 10^5$ molecule $cm^{-3}$ as the nighttime ·OH concentration in Zhengzhou.

16. L 253: Can you calculate OH radical concentration from those discussed later in L 418?

Response: OK. The concentration of ·OH during the daytime was calculated by the TUV model. But there is no sunlight at night, so it cannot be counted.

17. L 287: You are talking about night and mentioned no pollution source near the site. Why all these calculations related to traffic?? Unclear.

Response: Sorry for your comment. The sentence is not clear enough. So, we have changed "because no pollution source was near the measurement site" for "because the site is close to the western Fourth-Ring Expressway of Zhengzhou City and about Lian Huo Expressway to the north." in the revised text.

L 324-326: In the current study, directly emitted HONO could have been generated by vehicle exhaust and biomass combustion because the site is close to the western Fourth-Ring Expressway of Zhengzhou City and about Lian Huo Expressway to the north.

18. Pls explain the contradictory statements: important pathway (L 321) and unimportant pathway (L 20) for heterogeneous reaction for HONO formation.

Response: Sorry for my carelessness. With respect to direct emissions, heterogeneous reactions may be a more important pathway for HONO production.

The HONO/$NO_2$ ratio calculated in this work is much larger than that calculated for direct emission (< 1%) (Kurtenbach et al., 2001), suggesting that heterogeneous reactions may be a more important pathway for HONO production than direct emissions.

However, the average conversions of $NO_2$ ($C_{HONO}$) in CD, PD, and SPD periods were $0.72\times10^{-2}$, $0.64\times10^{-2}$, and $1.54\times10^{-2}$ h$^{-1}$, respectively, indicating that the heterogeneous conversion of $NO_2$ was unimportant than the homogeneous reaction. So, in order to prevent confusion, we have modified the sentence in the revised text.

L 17-20: The average conversions of $NO_2$ ($C_{HONO}$) in CD, PD, and SPD periods were $0.72\times10^{-2}$, $0.64\times10^{-2}$, and $1.54\times10^{-2}$ h$^{-1}$, respectively, indicating that the heterogeneous conversion of $NO_2$ was unimportant than the homogeneous reaction.

19. Use the stack format for equations in a separate line (L 389); use the solidus format for those within the line (L 402).

Response: Thank you. The equations had been changed.

L 438: $C_{HONO} = \dfrac{([HONO_{correct}]_{t2} - [HONO_{correct}]_{t1}}{(t2 - t1)\,[NO_2]}$

L 456: d HONO / d t = sources − sinks

20. L 431: Why the same values for both PD and SPD? That means you treat PD and SPD the same.

Response: OK. These are the averages per hour of $J_{HONO}$ and $J_{O^1D}$ for PD and SPD. We treated PD and SPD the same. The reason is that the main input parameters of TUV cannot be obtained directly, so we quoted the input parameters in the literature. However, the input parameters of PD and SPD are not

distinguished in the papers. We wanted to study that under the same output conditions from the TUV model, the impact of different pollution levels changed on the daytime budget. We have added the sentence in the revised text.

L 491-493: We wanted to study that under the same output conditions from the TUV model in the PD and SPD periods, the impact of different pollution levels changed on the daytime budget.

21. Fig. 1: How could one tell the wind direction?? There is no shade area of black and red color (caption says so). Change in Fig. 2 too.

Response: Sorry. We have removed the word, "WD" in **Fig. 1** in the revised text. The colors of the shaded area were redefined in **Fig. 1** and **Fig. 2**. The shaded areas: white for the CD period; gray for the PD period; red for the SPD period.

22. Fig. 6d and RH effect: Was the phenomenon also observed by others?? Who is to say that 77% is the inflection point? Just say "reach a certain high level,HONO…."

Response: Thank you for your comment. The phenomenon also observed by other studies (Cui et al., 2018; Huang et al., 2017; Tong et al., 2015). "52% and 77%" was removed in the revised text.

L 400-402: When RH was increased, the $HONO/NO_2$ ratio began to increase rapidly with RH. The $HONO/NO_2$ ratio decreased when RH reached a certain high level.

L 409-410: However, the surface reached saturation when RH reached a certain high level.

- Adequate subscript/superscript (e.g., L 535)

    L 582: …gas phase reactions of $O_x$, $HO_x$, $NO_x$, and $SO_x$…

- Use correct journal, or Phy. Chem. Chem Phy., (L 536)

    L 583: …Atmos. Chem. Phys.,...

- Use lowercases for articles (e.g., L 505)

    L 563: …Concentrations of nitrous acid, nitric acid, nitrite and nitrate in the gas
    and aerosol…

- L 546 ref is not cited.

    The reference has been removed in the revised text.

**Tech/English comments:**

    Response: Thank you for your carefulness. The problems have been modified and
    modified issues have been marked in yellow throughout the revised text. The parts
    that need explanation have been listed.

1. L 13, 178, 179: Use polluted (not pollution)

L 12: …polluted days (PD), and severely polluted days…

L 200-201: …clean days [CD], polluted days [PD], and severely polluted…

2. L 33: Use was (not should be)

L 33: …OH + HONO, was at least 0.22 ppbv h$^{-1}$…

3. Be consistent about the use of verb tense; some use past tense in the first part of the sentence with present tense used in the latter part.

L 34-35: …(less than 20%) was approximately 77%, which suggested…

4. L 32, 279: Delete the redundant abatement (first one)

L 32-33: …hourly level of HONO abatement pathways…

5. L 44: delete rate

L 43-44: …the contribution of HONO to ·OH…

6. L 54: UV/Vis

L 58-59: …stripping coil-UV/Vis…

7. L 56: result comparison

L 62: A result comparison of different instruments…

8. L 67: have discovered

L 72: …have discovered that motor vehicles…

9. L 68-69: How does ratio "account for"? Use "is" 0.1-0.8%

L 74: …(aside from $NO_X$ and other pollutants) is 0.1–0.8%.

10. Reword awkward L 81, word "radiation" (L 93), confusing (L 108-110).

L 85-87: Finlayson-Pitts et al. (2003) studied the mechanism of chemical adsorption of $NO_2$ and H ions on the adsorbed surface was revealed by using isotope-labeled water.

L 95-98: Su et al. (2008) revealed the importance of the ·OH from HONO during daytime (9:00–15:00 local time) and found that many unknown sources which are closely related to the solar radiation leading to HONO formation.

L 119-123: The file described the different factors which affect atmospheric pollution, including the level of economic development, energy structure, industrial structure and geographical location (solar radiation) with the Yangtze River Delta, Pearl River Delta, and Jing-Jin-Ji region.

11. L 99: Use pathway of…: mechanism is the same, but pathway is different.

L 109: …the pathway of HONO formation mechanism, namely…

12. L 107: "Food Production and Modern Agriculture" specified by (not published)

This website can not be found because of the update. We have changed

"http://www.ndrc.gov.cn/zcfb/zcfbtz/201212/P020121203614181974825.pdf" for "http://www.gov.cn/zhengce/content/2011-10/07/content_8208.htm", in the revised text (L 119).

13. L 112: delete "is" and insert "is" ahead of selected

L 124: …is selected in the study.

14. L 114: Not Zhengzhou chemical characteristics. Should have written as: chemical characteristics of PM in Zhengzhou.

L 125-126: …on the chemical characteristics of PM in Zhengzhou…

15. L 121: $PM_{2.5}$ is not chemical

We have changed "chemical" for "factors" in the revised text (L 124).
L 131: …between HONO and other factors, such as $PM_{2.5}$…

16. L 124: How systematic?

We have removed "This investigation of $PM_{2.5}$ and HONO is expected to clarify the sources, sinks, and reactions in fine PM pollution and the importance of systematic research." We have added the sentences in the revised text.
L 133-147: The levels of $PM_{2.5}$ were divided into three periods to analyze the HONO sources, sinks, and reactions in different periods. Many papers (Huang et al., 2017; Tong et al., 2016) took $PM_{2.5}$ as the main control factor of HONO, and studied the differences of HONO sources and characteristics between clean and polluted periods. No homogeneous reaction, direct emission, heterogeneous

reaction, and daytime budget analysis were conducted during the period of worsening pollution (namely HD period in this paper). Total $NO_X$ emissions in cities with different leading factors of emissions have been declining year by year due to Chinese government emission control measures, but some Chinese cities are still in high-$NO_X$ areas (e.g. Beijing, Shanghai, Guangzhou and Zhengzhou.) (Kim et al., 2015; Liu et al., 2017). Under high-$NO_X$ conditions, some papers (Cui et al., 2018; Hou et al., 2016) suggested that heterogeneous reaction was the main source of HONO and did not conduct a quantitative analysis of homogeneous reaction, especially in winter. So, we explore relevant studies of homogeneous reactions. In addition, the source contributions of HONO at night varied with the degree of pollution level were not explained.

17. L 137: western Fourth-Ring Expressway

L 161: …from the western Fourth-Ring Expressway of Zhengzhou City…

18. L 151: Use chemicals (in lieu of substances)

L 173: The chemicals that could be oxidized…

19. L 153: $O_2$ and $N_2$ (not O and N)

L 174: …several gases (e.g., $O_2$ and $N_2$)…

20. L 160-161: Clearly state which instrument is for which compound, e.g., 48i for CO measurement.

We have modified the sentence in the revised text.

L 181-184: A temporal resolution of the model analyzer (TE [used for measuring O$_3$], 48i [used for measuring CO], 42i [used for measuring NO, NO$_X$, and NO$_2$], and TEOM 1405 PM$_{2.5}$ monitor [used for measuring PM$_{2.5}$], Thermo Electron, USA) is 1 h.

21. L 167-168: The manual uses the term "should be". But in your statement, you should use "was changed" …."was calibrated".

L 193: The standard curve should be…

L 190-191: …were changed before the observation process, and the sampling flow was calibrated…

There are no serial **Numbers 22 and 23**. In order to prevent the confusion, the serial number used is consistent with the serial number in your comment.

24. L 171: A space after "≥" sign

L 194: …the correlation coefficient ($\geq 0.999$)…

25. L 181: specify that it is daily average

L 203: …the daily average of second grade…

26. Delete the first unit (L 182, 185, 188, 206, 301, 317, 360, etc.), first two units (L 301, 449, etc.)

L 205: …0 to 4.2 m s$^{-1}$…

L 208: …0 to 4.6 m s$^{-1}$…

L 210: …0 to 3.5 m s$^{-1}$…

L 230: …0.2 to 14.8 ppbv…

L 340: …2–52%, 6–34%, and 2–41%…

L 356: …1.3 and 59.0%…

L 401: …when RH reached a certain high level.

L 511: …15, 14, and 28%...

27. L 189: .. north with high WS … Also, how high is high? > 3 m/s, or > 4 m/s? Be specific!

L 212: …the maximum WS reached 4 m/s…

28. L 190: effect of pollutant removal

L 213: …the effect of pollutant removal…

29. L 202: mean values of what? of all pollutant concentrations?

L 226: The mean values of all pollutant concentrations except $O_3$…

30. L 209: Is your sampling site in urban area? You mentioned on the university campus. If so, the comparison is not valid.

OK. We have added the sampling site in Table 2. The university campus is urban in Zhengzhou. So, we compared the difference of the daytime and nighttime HONO level, HONO/$NO_2$, and HONO/$NO_X$ mean values in urban in other cities.

**Table 2.**

Comparisons of the daytime and nighttime HONO level, HONO/NO$_2$, and HONO/NO$_X$ mean values in Zhengzhou and other sites around the world.

| Date (Site) | Instrument | HONO (ppbv) | | | HONO/NO$_2$ (%) | | HONO/NO$_X$ (%) | | Reference |
|---|---|---|---|---|---|---|---|---|---|
| | | Day | Night | N/D | Day | Night | Day | Night | |
| Oct.–Nov. 2014 (Beijing, urban) | LOPAP (long path absorption photometer) | 0.9 | 1.8 | 2.0 | 2.6 | 4.6 | 1.7 | 2.2 | Tong et al., 2015 |
| Feb.–Mar. 2014 (Beijing, urban) | LOPAP | 1.8 | 2.1 | 1.2 | 3.8 (Severe haze) | 4.3 | 2.5 | 2.5 | Hou et al., 2016 |
| | | 0.5 | 0.9 | 1.8 | 7.8 (Clean) | 3.0 | 5.1 | 2.4 | |
| Jul. 2006 (Guangzhou, rural) | LOPAP | 0.2 | 0.9 | 4.5 | 1.0 | 2.5 | 4.3 | 4.5 | Li et al., 2012 |
| Jul. 2014–Aug. 2015 (Xi'an, urban) | LOPAP | 0.5 | 1.6 | 3.2 | 3.3 | 6.2 | | | Huang et al., 2017 |
| Aug. 2010–Jun. 2012 (Shanghai, urban) | Active DOAS | 0.8 | 1.1 | 1.4 | 4.2 | 4.5 | | | Wang et al., 2013 |
| Jul. 2009 (Paris, urban) | wet chemical derivatization technique-HPLC/UV-VIS detection | 0.1 | 0.2 | 2.0 | 3.3 | 2.5 | | | Michoud et al., 2014 |
| Jan. 2019 | AIM | 2.2 | 2.8 | 1.3 | 6.8 | 8.5 | 4.4 | 5.5 | This study |

31. L 214: No logic about sunrise – previous max HONO values (8-10 am) are way past sunrise. Should have written after 10 am…..

L 240: After 10:00 LT, the HONO concentration…

32. L 215: reword the incomplete phrase

We have reworded the sentence in the revised text.

L 240-242: After 10:00 LT, the HONO concentration decreased because of the increased solubility and rapid photolysis, remaining at a low level before sunset (14:00–16:00 LT).

33. L 219: delete again

L 243-244: After sunset, the concentrations of NO and NO$_2$ began to increase and remained at a higher level than the daytime.

34. L 220: and the concentration remained the same (Is it true??) until sunrise

It was not the same. The concentrations of NO and NO$_2$ began to increase and remained at a higher level than the daytime. We have modified the sentence in the revised text.

L 243-244: After sunset, the concentrations of NO and NO$_2$ began to increase and remained at a higher level than the daytime.

35. L 222: you meant NO$_2$ diffusion?

Sorry for my confusion. The concentrations of NO and NO$_2$ decreased after the peak values. On the one hand, NO and NO$_2$ can be involved in the reactions. On the other hand, NO and NO$_2$ diffused because of the boundary layer height increased in the daytime.

36. L 228: How is atmospheric migration? Should be "migration of atmospheric airmass".

L 253: …less affected by the migration of atmospheric airmass…

37. L 248: should be Eq. (1); Eq. (2) is in L 296.

L 277: …$P^{net}_{OH+NO}$ = k$_{OH+NO}$ [OH][NO] – k$_{OH+HONO}$ [OH][HONO]    (1).

38. L 249, 337, 338, 349, 389, etc.: A space b/a the = sign

L 278: At T = 298 K and P = 101 kPa…

L 377-378: …PM$_{2.5}$ (R$^2$ = 0.23) was weaker than that between HONO and PM$_{2.5}$ (R$^2$ = 0.55)…

L 390-392: …CO was relatively moderate (R$^2$ = 0.43)…

L 438: …$C_{HONO} = \dfrac{([HONO_{correct}]t_2 - [HONO_{correct}]t_1}{(t_2 - t_1)\,[NO_2]}$ (3)…

39. L 254: Adequate subscript in k$_{OH+bar}$

L 289: … the reaction rates of k$_{OH+NO}$…

40. L 274: reword

L 311-312: With the increase in pollution level, the HONO accumulation period at nighttime increased.

41. L 279: Use rate (not level)

L 316: …the hourly rate of HONO…

42. L 280: delete the extra spacing b/a the "/" sign

L 317-318: …3.36 – 1.59 ppbv)/8 h).

43. L 307: on the campus

L 346: …from the highway on the campus…

44. L 324: medium of what??

Sorry for my confusion. We have modified the sentence in the revised text.

L 361-364: With regard to the heterogeneous conversion of $NO_2$, several studies (An et al., 2012; Shen and Zhang, 2013) have reported that the surface of soot particles is the medium of $NO_2$ conversion.

45. L 349-350: wording? cannot see "indicating ...of what most cases"? Why?

Sorry for my confusion. The correlation coefficient between HONO and CO was relatively moderate ($R^2 = 0.43$), indicating that HONO and CO could come from the same source of emissions. Generally speaking, CO and NO are mainly related to combustion processes such as vehicle emissions, fossil fuel and biomass combustion (Tong et al., 2016). We have modified the sentence in the revised text.

L 390-396: The correlation coefficient between HONO and CO was relatively moderate ($R^2 = 0.43$), indicating that HONO and CO could come from the same source of emissions. Generally speaking, CO and NO are mainly related to combustion processes such as vehicle emissions, fossil fuel and biomass combustion (Tong et al., 2016). Thus, fossil fuel and biomass combustion may contribute to HONO production, but they can not be measured directly.

46. L 361: sedimentation of what? Do you mean deposit?

Right. We have modified the sentence in the revised text.

L 406-408: When RH ranged at the middle level, the heterogeneous

conversion of $NO_2$ to HONO was more significant than that of deposition.

47. L 367: Delete "study of the"

L 414: The correlation between $HONO_{correct}$ and $NO_2$…

48. L 374: decreases after…

L 421: …heterogeneous reaction (R4), and $NO_2$ decreased after midnight.

49. L 383: in the (not then)

L 432: …calculated in the current study…

50. L 396: How is rate improvement? Use increase

OK.

L 445: The increase in the conversion rate demonstrates…

51. L 402: The expression of …. represents

L 454: The expression of d HONO / d t represents…

52. L 411: delete an

L 471: …·OH and NO were formed as R1.

53. Table 1: A space before (%), or HONO/NOX (%); before year, or Jun. 2012, Also add SD

**Table 2.**

Comparisons of the daytime and nighttime HONO level, HONO/NO$_2$, and HONO/NO$_X$ mean values in Zhengzhou and other sites around the world.

| Date (Site) | Instrument | HONO (ppbv) | | | HONO/NO$_2$ (%) | | HONO/NO$_X$ (%) | | Reference |
|---|---|---|---|---|---|---|---|---|---|
| | | Day | Night | N/D | Day | Night | Day | Night | |
| Oct.–Nov. 2014 (Beijing, urban) | LOPAP (long path absorption photometer) | 0.9 | 1.8 | 2.0 | 2.6 | 4.6 | 1.7 | 2.2 | Tong et al., 2015 |
| Feb.–Mar. 2014 (Beijing, urban) | LOPAP | 1.8 | 2.1 | 1.2 | 3.8 (Severe haze) | 4.3 | 2.5 | 2.5 | Hou et al., 2016 |
| | | 0.5 | 0.9 | 1.8 | 7.8 (Clean) | 3.0 | 5.1 | 2.4 | |
| Jul. 2006 (Guangzhou, rural) | LOPAP | 0.2 | 0.9 | 4.5 | 1.0 | 2.5 | 4.3 | 4.5 | Li et al., 2012 |
| Jul. 2014–Aug. 2015 (Xi'an, urban) | LOPAP | 0.5 | 1.6 | 3.2 | 3.3 | 6.2 | | | Huang et al., 2017 |
| Aug. 2010–Jun. 2012 (Shanghai, urban) | Active DOAS | 0.8 | 1.1 | 1.4 | 4.2 | 4.5 | | | Wang et al., 2013 |
| Jul. 2009 (Paris, urban) | wet chemical derivatization technique-HPLC/UV-VIS detection | 0.1 | 0.2 | 2.0 | 3.3 | 2.5 | | | Michoud et al., 2014 |
| Jan. 2019 | AIM | 2.2 | 2.8 | 1.3 | 6.8 | 8.5 | 4.4 | 5.5 | This study |

The values of SD were shown in the references.

54. Fig. 3: Need errors bars. Use the present tense, "represent". Show one example in Box plots so one has an idea about the magnitude of variation.

55. Fig. 4: Need error bars

With too many error bars in figures, it will make the figures look unclear. So the error bars of **Fig. 4**, **Fig. 5**, and **Fig. 8** were placed separately in the tables of the supplement (**Table S2**, **Table S3**, and **Table S4**).

56. Fig. 6: A space after HONO

We have modified the figure in the revised text **Fig. 7**.

**Fig. 7.** Nighttime correlation studies between $PM_{2.5}$ and $HONO/NO_2$, $PM_{2.5}$ and HONO, CO and HONO, RH and $HONO/NO_2$ during the entire measurement period, CD, PD, and SPD periods. The blue represented the full measurement period; the light blue represented CD period; the black represented PD period; the red represented SPD period.

[Figure]

**Characteristics, sources, and reactions of nitrous acid during winter at an urban site in the Central Plains Economic Region in China**

Qi Hao, Nan Jiang*, Ruiqin Zhang, Liuming Yang, and Shengli Li

Key Laboratory of Environmental Chemistry and Low Carbon Technologies of Henan Province, Research Institute of Environmental Science, College of Chemistry, School of Ecology and Environment, Zhengzhou University, Zhengzhou 450001, China

**Supplement:**

**1. This AIM method and its details.**

HONO was hygroscopically grown in the parallel plate denuder and collected as an aqueous solution in a cyclone assembly. The aqueous sample aliquots from both channels were transported to the ion chromatographic systems housed inside a ground container for hourly semicontinuous online analysis of HONO. The ion chromatographic system was calibrated for $NO_2^-$ using mixed anion standard solutions of $NO_2^-$.

**2. The concentration of OH radicals was calculated with the formulas of $NO_2$, $O_3$, and $J_O{}^1{}_D$.**

$$[OH] = \frac{k_{HO_2+NO}\tau_{HC}[NO_2]F_J}{k_{NO+O_3}} \times \sqrt{\frac{\alpha}{k_{HO_2+HO_2}[O_3]}} \times J(O^1D),$$

where [OH] represents the concentration of OH radicals, $k_{HO_2+NO} = 8.56 \times 10^{-12}$ cm$^3$ s$^{-1}$, $\tau_{HC} = 0.3$ s, [$NO_2$] represents the $NO_2$ concentration, $F_J = 2$ s$^{-0.5}$, $k_{NO+O_3} = 1.82 \times 10^{-14}$ cm$^3$ s$^{-1}$, $\alpha = 0.075$, $k_{HO_2+HO_2} = 8.56 \times 10^{-12}$ cm$^3$ s$^{-1}$, [$O_3$] represents the $O_3$ concentration, and $J(O^1D)$ represents the $O^1D$ efficiency of photolysis.

**Figure Captions:**

Fig. S1. The correlation study between $HONO_{correct}$ and $NO_2$ in the nighttime.

[Figure]

**Fig. S1**. The correlation study between HONO$_{correct}$ and NO$_2$ in the nighttime.

**Table Captions:**

Table S1. Measured species and performance of the instruments.

Table S2 The error bars of Fig. 4. (The units of all species except $HONO/NO_2$ and $HONO/NO_x$ are ppbv. The units of $HONO/NO_2$ and $HONO/NO_x$ are %.)

Table S3 The error bars of Fig. 5. (The units of all species except $P^{net}_{OH+NO}$ are ppbv. The unit of $P^{net}_{OH+NO}$ is ppbv/h.)

Table S4 The error bars of Fig. 8. (The units of all species except $HONO_{correct}/NO_2$ are ppbv. The unit of $HONO_{correct}/NO_2$ is %.)

**Table S1.** Measured species and performance of the instruments.

| Species | Measurement technique | Detection limit | Accuracy |
|---|---|---|---|
| $PM_{2.5}$ | Tapered Element Oscillating Microbalance | $1.5~\mu g~m^{-3}$ | ± 5% |
| HONO | Ion Chromatography | 4 pptv | ± 20% |
| CO | Absorbs Infrared Radiation | 40 ppbv | ± 5% |
| NO | Chemiluminescence | 60 pptv | ± 20% |
| $NO_2$ | Chemiluminescence | 300 pptv | ± 20% |
| $O_3$ | UV Photometry | 0.5 ppbv | ± 5% |

The results came from instrument manufacturers.

**Table S2**-1 The error bars of Fig. 4. (The units of all species except HONO/NO$_2$ and HONO/NO$_x$ are ppbv. The units of HONO/NO$_2$ and HONO/NO$_x$ are %.)

| Species-period | \multicolumn Local Time (hh:mm) | | | | | | | | | |
|---|---|---|---|---|---|---|---|---|---|---|
| | 00:00 | 01:00 | 02:00 | 03:00 | 04:00 | 05:00 | 06:00 | 07:00 | 08:00 | 09:00 |
| HONO-CD | 1.7 ± 1.3 | 1.4 ± 0.6 | 1.3 ± 0.4 | 1.2 ± 0.3 | 1.2 ± 0.2 | 1.2 ± 0.2 | 1.4 ± 0.3 | 1.5 ± 0.6 | 1.7 ± 0.9 | 1.6 ± 0.9 |
| HONO-PD | 3.2 ± 1.5 | 3.1 ± 1.3 | 3 ± 1.1 | 3.3 ± 1.2 | 3.5 ± 1.3 | 3.5 ± 1.2 | 3.6 ± 1.1 | 3.3 ± 0.9 | 3.7 ± 1.6 | 4.1 ± 2.8 |
| HONO-SPD | 3.7 ± 0.9 | 4 ± 0.8 | 4.2 ± 0.6 | 4.4 ± 0.8 | 4.6 ± 1 | 4.6 ± 1.2 | 4.6 ± 1.5 | 4.4 ± 1.3 | 4.4 ± 1.1 | 5.7 ± 3 |
| NO-CD | 14.3 ± 17 | 9 ± 9.7 | 8.5 ± 12.7 | 10.1 ± 22.4 | 10.6 ± 21.1 | 21.9 ± 29 | 27.8 ± 33 | 40.1 ± 51 | 52.6 ± 79 | 55.5 ± 84 |
| NO-PD | 57.3 ± 48 | 62.7 ± 55.9 | 49.6 ± 49 | 44 ± 47.8 | 47 ± 48.7 | 46.6 ± 30 | 41.4 ± 34 | 44.7 ± 33 | 48.9 ± 35 | 53.7 ± 44 |
| NO-SPD | 79.4 ± 103 | 100.1 ± 118 | 128.3 ±133 | 129 ± 134 | 111 ± 119 | 117 ± 95 | 100 ± 94 | 88.4 ± 85 | 82.3 ± 70 | 85.4 ± 71 |
| NO$_2$-CD | 25.4 ± 8.2 | 25.6 ± 9.9 | 24.7 ± 10.5 | 22.9 ± 10.4 | 24 ± 11.4 | 20.7 ± 11 | 20.2 ± 9 | 23.6 ± 11 | 28.6 ± 18 | 28.6 ± 18 |
| NO$_2$-PD | 41.1 ± 10 | 40.8 ± 11.2 | 39.7 ± 10.7 | 37.9 ± 7.1 | 36.6 ± 5.4 | 35.9 ± 5 | 33.8 ± 6 | 34.4 ± 6 | 33.2 ± 5 | 30.7 ± 6 |
| NO$_2$-SPD | 45.3 ± 9.5 | 43.5 ± 9.2 | 42.8 ± 8.8 | 42.1 ± 8.2 | 42.2 ± 8.1 | 41 ± 7.1 | 40.6 ± 6.9 | 40.7 ± 6 | 40.1 ± 6 | 39.2 ± 7 |
| O$_3$-CD | 14.2 ± 10 | 13.6 ± 10.4 | 14.2 ± 10.1 | 14.9 ± 9.4 | 13.6 ± 9.1 | 11.7 ± 10 | 13.8 ± 10 | 12.9 ± 9 | 11.6 ± 8 | 12.1 ± 7 |
| O$_3$-PD | 6.6 ± 6.1 | 6.4 ± 5.2 | 7.1 ± 5.2 | 6.3 ± 3.3 | 4.7 ± 2.2 | 5.3 ± 3 | 7.7 ± 6.9 | 5.3 ± 2.8 | 5.5 ± 3 | 7.1 ± 4 |
| O$_3$-SPD | 7.8 ± 6.4 | 7.7 ± 6.2 | 7.3 ± 5 | 6 ± 2.9 | 5.3 ± 2.3 | 5 ± 2.1 | 5.6 ± 2.5 | 5.2 ± 2.2 | 5.6 ± 2.6 | 6 ± 2.6 |
| HONO/NO$_2$-CD | 3.8 ± 1.5 | 4.4 ± 1 | 4.4 ± 1.1 | 4.9 ± 1 | 5.1 ± 0.8 | 8.3 ± 6 | 6.9 ± 2.1 | 6.2 ± 1.4 | 5.1 ± 0.8 | 4.3 ± 1.1 |
| HONO/NO$_2$-PD | 8 ± 3.6 | 7.8 ± 3.4 | 8 ± 3.3 | 9 ± 3.7 | 10 ± 4.5 | 10.1 ± 4 | 11.2 ± 4.6 | 10.3 ± 4 | 12.1 ± 7 | 14.3 ± 11 |
| HONO/NO$_2$-SPD | 8.3 ± 1.9 | 9.3 ± 1.4 | 10 ± 1.5 | 10.7 ± 1.9 | 11 ± 2.2 | 11.3 ± 3 | 11.5 ± 3.9 | 10.9 ± 3 | 11.1 ± 2 | 15 ± 8.3 |
| HONO/NO$_x$-CD | 2.7 ± 1.4 | 3.7 ± 1.5 | 4.2 ± 1.4 | 4.9 ± 1.1 | 4.9 ± 1 | 5.3 ± 2.5 | 5.1 ± 2.9 | 4.5 ± 2.4 | 3.6 ± 1.5 | 2.8 ± 1.4 |
| HONO/NO$_x$-PD | 4.4 ± 1.4 | 4.3 ± 1.7 | 4.6 ± 1.5 | 5.3 ± 1.3 | 5.3 ± 1 | 5.3 ± 1.1 | 6.6 ± 2.7 | 5.9 ± 2.3 | 6.5 ± 3.8 | 6.6 ± 4.3 |
| HONO/NO$_x$-SPD | 5.1 ± 2 | 5.3 ± 2.4 | 5.4 ± 3.4 | 5.8 ± 3.9 | 6.1 ± 3.9 | 5.7 ± 3.7 | 5.9 ± 3.6 | 5.7 ± 3 | 5.8 ± 2.9 | 6.7 ± 3.1 |

**Table S2**-2 The error bars of Fig. 4. (The units of all species except HONO/NO$_2$ and HONO/NO$_x$ are ppbv. The units of HONO/NO$_2$ and HONO/NO$_x$ are %.)

| Species-period | Local Time (hh:mm) | | | | | | | | | |
|---|---|---|---|---|---|---|---|---|---|---|
| | 10:00 | 11:00 | 12:00 | 13:00 | 14:00 | 15:00 | 16:00 | 17:00 | 18:00 | 19:00 |
| HONO-CD | 1.1 ± 0.6 | 0.6 ± 0.3 | 0.5 ± 0.3 | 0.6 ± 0.4 | 0.6 ± 0.5 | 0.7 ± 0.5 | 0.6 ± 0.5 | 0.7 ± 0.4 | 1 ± 0.5 | 1.2 ± 0.5 |
| HONO-PD | 2.9 ± 1.9 | 1.9 ± 1.3 | 1.3 ± 0.7 | 1 ± 0.3 | 0.9 ± 0.3 | 0.9 ± 0.3 | 0.9 ± 0.3 | 1.1 ± 0.4 | 1.4 ± 0.3 | 1.7 ± 0.3 |
| HONO-SPD | 6.9 ± 4.3 | 5.2 ± 3.8 | 3 ± 1.3 | 2.1 ± 0.7 | 1.8 ± 0.7 | 1.7 ± 0.6 | 1.8 ± 0.7 | 2 ± 0.5 | 2.7 ± 0.7 | 2.8 ± 0.8 |
| NO-CD | 43.9 ± 69.8 | 27.9 ± 40.8 | 14.9 ± 17.1 | 10.3 ± 7.8 | 7.3 ± 3 | 6 ± 4.5 | 6.4 ± 5.6 | 3.6 ± 3.4 | 2.6 ± 3.2 | 5.9 ± 7.7 |
| NO-PD | 49.3 ± 45.2 | 30 ± 26.2 | 21 ± 20.7 | 12.7 ± 14.7 | 9.4 ± 12.3 | 8.4 ± 9.5 | 5.7 ± 4.7 | 6.3 ± 6.8 | 9 ± 9 | 10 ± 10.3 |
| NO-SPD | 90.8 ± 73.4 | 79.3 ± 69.3 | 57.1 ± 52.3 | 34.8 ± 36.4 | 24.5 ± 28.7 | 19 ± 24.7 | 15 ± 18.8 | 11.8 ± 11 | 11.8 ± 7.9 | 22.4 ± 21 |
| NO$_2$-CD | 26.8 ± 15.7 | 22.7 ± 9.2 | 17.6 ± 7.1 | 17.1 ± 9 | 19.6 ± 9.6 | 21 ± 10.7 | 20.5 ± 9 | 21.4 ± 9 | 26 ± 12.5 | 30 ± 13.7 |
| NO$_2$-PD | 30 ± 6.9 | 28.8 ± 7.7 | 27.4 ± 9.6 | 24.8 ± 9.4 | 22.5 ± 10.6 | 25 ± 9.9 | 25.7 ± 9.3 | 27.1 ± 9 | 35 ± 8.7 | 36.2 ± 9.2 |
| NO$_2$-SPD | 39.8 ± 7.8 | 41.5 ± 8.3 | 42.3 ± 10.1 | 39.5 ± 12.6 | 38.5 ± 14.3 | 38 ± 14.7 | 38 ± 13.9 | 42 ± 15.4 | 45 ± 11.5 | 47 ± 10.8 |
| O$_3$-CD | 15.9 ± 8.8 | 19.5 ± 9.7 | 22.6 ± 8.3 | 25.5 ± 8.5 | 28.1 ± 9.1 | 29 ± 10.8 | 28 ± 10.8 | 29 ± 10.2 | 23.6 ± 10 | 17 ± 8.9 |
| O$_3$-PD | 9.6 ± 6.1 | 12.8 ± 6.2 | 18.7 ± 8.3 | 24.1 ± 8.4 | 28.2 ± 9.7 | 27 ± 10.8 | 28 ± 10.4 | 26 ± 10.5 | 17.4 ± 8.6 | 15 ± 11.6 |
| O$_3$-SPD | 6.3 ± 2.4 | 8.7 ± 4.5 | 12.8 ± 8.5 | 19.4 ± 12.9 | 24.1 ± 14.7 | 28 ± 16.6 | 29 ± 17.6 | 25 ± 16.1 | 17 ± 11.1 | 10.6 ± 9.7 |
| HONO/NO$_2$-CD | 4.1 ± 2.3 | 3.1 ± 1.9 | 3.3 ± 1.9 | 3.3 ± 1.3 | 3.1 ± 1.3 | 3.1 ± 1.3 | 2.9 ± 1.4 | 3.1 ± 1.4 | 3.9 ± 1.4 | 4.5 ± 2.2 |
| HONO/NO$_2$-PD | 9.4 ± 5.6 | 6.2 ± 3 | 4.7 ± 1.5 | 4.2 ± 1.2 | 4.7 ± 2.2 | 3.9 ± 0.7 | 3.7 ± 0.4 | 4.1 ± 1.2 | 4.3 ± 0.9 | 5 ± 1.5 |
| HONO/NO$_2$-SPD | 18.9 ± 13.7 | 13.7 ± 12 | 7.3 ± 3.5 | 5.6 ± 2.6 | 4.9 ± 2.1 | 4.8 ± 2.4 | 4.9 ± 1.6 | 5 ± 1 | 6.3 ± 1.8 | 6.2 ± 1.5 |
| HONO/NO$_x$-CD | 2.9 ± 2.1 | 2.2 ± 1.5 | 2.4 ± 1.5 | 2.5 ± 1.1 | 2.5 ± 1 | 2.6 ± 0.9 | 2.5 ± 0.9 | 2.8 ± 1 | 3.7 ± 1.1 | 4.1 ± 1.9 |
| HONO/NO$_x$-PD | 4.8 ± 2.4 | 3.8 ± 1.3 | 3.5 ± 1.2 | 3.5 ± 1.5 | 4 ± 2.1 | 3.4 ± 0.9 | 3.3 ± 0.5 | 3.7 ± 1.2 | 3.8 ± 0.7 | 4.3 ± 1.5 |
| HONO/NO$_x$-SPD | 8.2 ± 5.8 | 6.9 ± 5.7 | 4.3 ± 2 | 4 ± 2 | 3.8 ± 1.6 | 3.9 ± 1.9 | 4.3 ± 1.6 | 4.5 ± 1.2 | 5.5 ± 1.5 | 4.9 ± 1.3 |

**Table S2**-3 The error bars of Fig. 4. (The units of all species except $HONO/NO_2$ and $HONO/NO_x$ are ppbv. The units of $HONO/NO_2$ and $HONO/NO_x$ are %.)

| Species-period | Local Time (hh:mm) | | | |
|---|---|---|---|---|
| | 20:00 | 21:00 | 22:00 | 23:00 |
| HONO-CD | $1.3 \pm 0.6$ | $1.6 \pm 0.9$ | $2 \pm 0.9$ | $2.1 \pm 0.9$ |
| HONO-PD | $1.7 \pm 0.7$ | $1.8 \pm 0.8$ | $2 \pm 0.9$ | $2.1 \pm 0.9$ |
| HONO-SPD | $3.1 \pm 0.9$ | $3.2 \pm 0.9$ | $3.7 \pm 0.8$ | $4.6 \pm 1.2$ |
| NO-CD | $11.1 \pm 16.9$ | $14.5 \pm 22.5$ | $35.5 \pm 68.9$ | $50.8 \pm 99.2$ |
| NO-PD | $15 \pm 14.1$ | $15.3 \pm 14.7$ | $27.4 \pm 28.5$ | $33.9 \pm 28.9$ |
| NO-SPD | $29.4 \pm 24.2$ | $37.3 \pm 26.6$ | $38.5 \pm 23.1$ | $51.4 \pm 31.4$ |
| $NO_2$-CD | $31 \pm 13.8$ | $30.3 \pm 14.5$ | $31.6 \pm 13.6$ | $31 \pm 14.3$ |
| $NO_2$-PD | $37.3 \pm 10.5$ | $38.5 \pm 13.9$ | $38.3 \pm 13.5$ | $37.1 \pm 13.2$ |
| $NO_2$-SPD | $44.5 \pm 11$ | $43.5 \pm 11.5$ | $43.5 \pm 11.1$ | $42.1 \pm 13.1$ |
| $O_3$-CD | $13.3 \pm 10.1$ | $14 \pm 11$ | $12.2 \pm 8.7$ | $12.7 \pm 8.8$ |
| $O_3$-PD | $13.7 \pm 10.3$ | $10.9 \pm 8.5$ | $10.9 \pm 7.7$ | $12.2 \pm 10.4$ |
| $O_3$-SPD | $9.9 \pm 8.6$ | $10.8 \pm 9.2$ | $9.7 \pm 8.7$ | $9.6 \pm 9.6$ |
| $HONO/NO_2$-CD | $4.6 \pm 2.2$ | $5.7 \pm 2.6$ | $6.5 \pm 2.6$ | $6.8 \pm 2.7$ |
| $HONO/NO_2$-PD | $4.7 \pm 1.9$ | $4.6 \pm 1.2$ | $4.9 \pm 0.8$ | $5.3 \pm 0.8$ |
| $HONO/NO_2$-SPD | $7 \pm 1.5$ | $7.5 \pm 1.4$ | $8.9 \pm 2.3$ | $9.4 \pm 2.4$ |
| $HONO/NO_x$-CD | $4 \pm 1.9$ | $4.8 \pm 2.2$ | $4.9 \pm 2.8$ | $5 \pm 3$ |
| $HONO/NO_x$-PD | $3.9 \pm 2.1$ | $3.9 \pm 1.3$ | $3.8 \pm 1$ | $3.8 \pm 0.9$ |
| $HONO/NO_x$-SPD | $5.1 \pm 1.5$ | $5.2 \pm 2$ | $5.8 \pm 2$ | $5 \pm 1.4$ |

**Table S3**-1 The error bars of Fig. 5. (The units of all species except $P_{OH+NO}^{net}$ are ppbv. The unit of $P_{OH+NO}^{net}$ is ppbv/h.)

| | Local Time (hh:mm) | | | | | | | | | |
|---|---|---|---|---|---|---|---|---|---|---|
| Species-period | 19:00 | 20:00 | 21:00 | 22:00 | 23:00 | 00:00 | 01:00 | 02:00 | 03:00 | 04:00 |
| $P_{OH+NO}^{net}$-CD | 0.04 ± 0.06 | 0.08 ± 0.12 | 0.11 ± 0.17 | 0.33 ± 0.54 | 0.47 ± 0.79 | 0.12 ± 0.13 | 0.07 ± 0.08 | 0.03 ± 0.03 | 0.01 ± 0.1 | 0.02 ± 0.1 |
| HONO-CD | 1.18 ± 0.48 | 1.32 ± 0.62 | 1.62 ± 0.9 | 2.02 ± 0.94 | 2.09 ± 0.9 | 1.67 ± 1.34 | 1.43 ± 0.63 | 1.26 ± 0.44 | 1.2 ± 0.3 | 1.2 ± 0.22 |
| NO-CD | 5.4 ± 6.5 | 10.2 ± 14.4 | 13.3 ± 19.2 | 38.2 ± 62.2 | 54.9 ± 89.7 | 15 ± 14.8 | 8.8 ± 8.6 | 3.7 ± 4.2 | 1.5 ± 2.3 | 2.5 ± 2.6 |
| $P_{OH+NO}^{net}$-HD | 0.07 ± 0.07 | 0.1 ± 0.1 | 0.1 ± 0.1 | 0.19 ± 0.2 | 0.23 ± 0.2 | 0.4 ± 0.34 | 0.44 ± 0.4 | 0.34 ± 0.35 | 0.3 ± 0.34 | 0.3 ± 0.34 |
| HONO-HD | 1.7 ± 0.27 | 1.71 ± 0.68 | 1.82 ± 0.78 | 1.98 ± 0.89 | 2.06 ± 0.93 | 3.21 ± 1.54 | 3.05 ± 1.27 | 3.01 ± 1.08 | 3.3 ± 1.17 | 3.5 ± 1.34 |
| NO-HD | 8.5 ± 8.4 | 12.2 ± 11.5 | 12.5 ± 12 | 22.4 ± 23.3 | 27.7 ± 23.6 | 46.8 ± 39.5 | 51.2 ± 45.6 | 40.5 ± 40 | 35.9 ± 39 | 38 ± 39.7 |
| $P_{OH+NO}^{net}$-SHD | 0.15 ± 0.15 | 0.2 ± 0.17 | 0.25 ± 0.18 | 0.26 ± 0.16 | 0.35 ± 0.23 | 0.55 ± 0.75 | 0.7 ± 0.85 | 0.9 ± 0.96 | 0.9 ± 1.0 | 0.8 ± 0.86 |
| HONO-SHD | 2.8 ± 0.8 | 3.1 ± 0.9 | 3.2 ± 0.9 | 3.7 ± 0.8 | 4.6 ± 1.2 | 3.7 ± 0.9 | 4 ± 0.8 | 4.2 ± 0.6 | 4.4 ± 0.8 | 4.6 ± 1 |
| NO-SHD | 18 ± 17 | 24 ± 20 | 30 ± 21 | 31 ± 19 | 42 ± 25 | 64 ± 84 | 81 ± 96 | 104 ± 108 | 105 ± 110 | 90 ± 97 |

**Table S3**-2 The error bars of Fig. 5. (The units of all species except $P_{OH+NO}^{net}$ are ppbv. The unit of $P_{OH+NO}^{net}$ is ppbv/h.)

| | Local Time (hh:mm) | |
|---|---|---|
| Species-period | 05:00 | 06:00 |
| $P_{OH+NO}^{net}$-CD | 0.12 ± 0.18 | 0.17 ± 0.22 |
| HONO-CD | 1.25 ± 0.21 | 1.36 ± 0.35 |
| NO-CD | 13.7 ± 20.9 | 19.5 ± 25.1 |
| $P_{OH+NO}^{net}$-HD | 0.32 ± 0.22 | 0.28 ± 0.25 |
| HONO-HD | 3.5 ± 1.16 | 3.56 ± 1.09 |
| NO-HD | 38 ± 25.2 | 33.8 ± 28.5 |
| $P_{OH+NO}^{net}$-SHD | 0.82 ± 0.87 | 0.7 ± 0.68 |
| HONO-SHD | 4.6 ± 1.2 | 4.6 ± 1.5 |
| NO-SHD | 95.6 ± 99 | 81.8 ± 77.1 |

**Table S4**-1 The error bars of Fig. 8. (The units of all species except $HONO_{correct}/NO_2$ are ppbv. The unit of $HONO_{correct}/NO_2$ is %.)

| Species-period | 19:00 | 20:00 | 21:00 | 22:00 | 23:00 | 00:00 | 01:00 | 02:00 | 03:00 | 04:00 |
|---|---|---|---|---|---|---|---|---|---|---|
| $HONO_{correct}$-CD | 1.0 ± 0.4 | 1.1 ± 0.6 | 1.4 ± 0.8 | 1.6 ± 0.7 | 1.6 ± 0.6 | 1.4 ± 1.4 | 1.2 ± 0.7 | 1.1 ± 0.5 | 1.1 ± 0.4 | 1.1 ± 0.2 |
| $NO_2$-CD | 30 ± 15 | 31 ± 15 | 30 ± 15 | 34 ± 15 | 34 ± 15 | 25 ± 9 | 24 ± 8 | 22 ± 8 | 20 ± 8 | 20 ± 8 |
| $HONO_{correct}/NO_2$-CD | 3.7 ± 2.2 | 3.9 ± 2.2 | 4.9 ± 2.6 | 5.5 ± 2.7 | 5.7 ± 2.9 | 11 ± 18.2 | 8.9 ± 12 | 8.6 ± 10.8 | 8.5 ± 9.7 | 7.7 ± 7.4 |
| $HONO_{correct}$-HD | 1.4 ± 0.3 | 1.4 ± 0.7 | 1.5 ± 0.7 | 1.6 ± 0.8 | 1.7 ± 0.8 | 2.7 ± 1.3 | 2.5 ± 1 | 2.5 ± 0.8 | 2.9 ± 0.9 | 3.1 ± 1.1 |
| NO2-HD | 36 ± 9 | 37 ± 10 | 39 ± 14 | 38 ± 13 | 37 ± 13 | 41 ± 10 | 41 ± 11 | 40 ± 11 | 38 ± 7 | 37 ± 5 |
| $HONO_{correct}/NO_2$-HD | 4.2 ± 1.5 | 3.8 ± 2 | 3.8 ± 1.2 | 4 ± 0.8 | 4.4 ± 0.7 | 6.7 ± 3.1 | 6.5 ± 2.8 | 6.7 ± 2.8 | 7.8 ± 3.1 | 8.7 ± 3.8 |
| $HONO_{correct}$-SHD | 2.4 ± 0.6 | 2.6 ± 0.7 | 2.7 ± 0.7 | 3.2 ± 0.7 | 4.1 ± 1.3 | 3.1 ± 0.8 | 3.3 ± 0.6 | 3.4 ± 0.7 | 3.6 ± 1 | 3.9 ± 1.1 |
| $NO_2$-SHD | 47 ± 11 | 44 ± 11 | 43 ± 11 | 44 ± 11 | 42 ± 13 | 45 ± 9 | 43 ± 9 | 43 ± 9 | 42 ± 8 | 42 ± 8 |
| $HONO_{correct}/NO_2$-SHD | 5.4 ± 1.4 | 6.1 ± 1.4 | 6.5 ± 1.4 | 7.8 ± 2.2 | 14.4 ± 16.7 | 7 ± 1.9 | 7.8 ± 1.6 | 8.1 ± 2.2 | 8.8 ± 2.8 | 9.3 ± 2.9 |

**Table S4**-2 The error bars of Fig. 8. (The units of all species except $HONO_{correct}/NO_2$ are ppbv. The unit of $HONO_{correct}/NO_2$ is %.)

|  | Local Time (hh:mm) | |
| --- | --- | --- |
| Species-period | 05:00 | 06:00 |
| $HONO_{correct}$-CD | $1.0 \pm 0.4$ | $1.1 \pm 0.6$ |
| $NO_2$-CD | $30 \pm 15$ | $31 \pm 15$ |
| $HONO_{correct}/NO_2$-CD | $3.7 \pm 2.2$ | $3.9 \pm 2.2$ |
| $HONO_{correct}$-HD | $1.4 \pm 0.3$ | $1.4 \pm 0.7$ |
| $NO_2$-HD | $36 \pm 9$ | $37 \pm 10$ |
| $HONO_{correct}/NO_2$-HD | $4.2 \pm 1.5$ | $3.8 \pm 2$ |
| $HONO_{correct}$-SHD | $2.4 \pm 0.6$ | $2.6 \pm 0.7$ |
| $NO_2$-SHD | $47 \pm 11$ | $44 \pm 11$ |
| $HONO_{correct}/NO_2$-SHD | $5.4 \pm 1.4$ | $6.1 \pm 1.4$ |

**Reference**

[revised manuscript text omitted]

---

## Author Comment (AC2) · 29 Jan 2020

**Itemized Response to Anonymous Referee #2's Comments**

**Ms. Ref. No.**: acp-2019-916

**Title:** Characteristics, sources, and reactions of nitrous acid during winter at an urban site in the Central Plains Economic Region in China

**Response to Anonymous Referee #2:**

We have carefully addressed your comments on our manuscript and made necessary revisions of the previous manuscript. We sincerely thank you for valuable and constructive inputs. We believe that we have adequately addressed all of your comments and thus the current version has been greatly improved with those valuable comments and further English editing. The revised phrases/sentences/paragraphs are shown in the line number of the revised text.

The followings are our itemized replies to your comments.

**Main comments**:

1) The description of HONO measurements is too brief and insufficient while all the study rely on it. A detailed and self-sufficient description of the measurement technique for HONO is therefore needed even if it has been described in another study. Estimation of instrumental uncertainties are also lacking.

Furthermore, description of the measurement techniques used for ancillary species should also be given (at least the measurement principle and not only the model and brand of the analyzers).

On the contrary Fig. S1 and S2 does not bring valuable information and should be completed to describe more precisely the measurement principle or should be removed.

Response: Thank you for your comment. A detailed description of this inlet design and the performance characteristics of the AIM system can be found in Markovic et al. (2012). HONO was hygroscopically grown in the parallel plate denuder and collected as an aqueous solution in a cyclone assembly. The aqueous sample

aliquots from both channels were transported to the ion chromatographic systems housed inside a ground container for hourly semicontinuous online analysis of HONO. The ion chromatographic system was calibrated for $NO_2^-$ using mixed anion standard solutions of $NO_2^-$, which was concentrated and analyzed as described by Markovic et al. (2012).

So, we have modified the sentence in the revised text.

L 176-179: This measurement method and its details have been successfully evaluated in many field studies (Markovic et al., 2012; Tian et al., 2018; Wang et al., 2019), and shown in the supplement.

In the supplement, we have added, this part in the supplement.

**1. This AIM method and its details.**

HONO was hygroscopically grown in the parallel plate denuder and collected as an aqueous solution in a cyclone assembly. The aqueous sample aliquots from both channels were transported to the ion chromatographic systems housed inside a ground container for hourly semicontinuous online analysis of HONO. The ion chromatographic system was calibrated for $NO_2^-$ using mixed anion standard solutions of $NO_2^-$.

The description of the measurement techniques and instrumental uncertainties was shown in **Table S1.**

**Table S1.** Measured species and performance of the instruments.

| Species | Measurement technique | Detection limit | Accuracy |
|---------|----------------------|-----------------|----------|
| $PM_{2.5}$ | Tapered Element Oscillating Microbalance | 1.5 μg m$^{-3}$ | ± 5% |
| HONO | Ion Chromatography | 4 pptv | ± 20% |
| CO | Absorbs Infrared Radiation | 40 ppbv | ± 5% |
| NO | Chemiluminescence | 60 pptv | ± 20% |
| $NO_2$ | Chemiluminescence | 300 pptv | ± 20% |
| $O_3$ | UV photometry | 0.5 ppbv | ± 5% |

The results came from instrument manufacturers.

At last, Fig. S1 and S2 have been removed.

2) P10, line 253: $1.0 \times 10^6$ molecules $cm_{-3}$ is very high for nighttime concentrations of OH especially in January. Lelieveld et al. (2016) report nocturnal concentrations of OH between $1.5 \times 10^4$ and $3 \times 10^4$ molecules $cm_{-3}$ for January in the region concerned by the present study and not $1.0 \times 10^6$ molecules $cm_{-3}$ as stated by the authors. Tan et al. (2018) also found nighttime OH concentrations below $1 \times 10^5$ in Beijing during winter (February).

Response: Thank you. Your comment is critical and important. We revisit and determine the OH concentration. You are right. $2.5 \times 10^5$ cm$^3$ molecule$^{-1}$ is very high for nighttime concentrations of OH, especially in January.

And, nighttime OH concentration increased as the latitude decreases ranged 3 to $6 \times 10^5$ cm$^3$ molecule$^{-1}$ (Lelieveld et al., 2016) (On the first figure) by the general circulation model EMAC (ECHAM/MESSy Atmospheric Chemistry).

Tan et al. (2018) found that by the field measurement, the average concentration of ·OH in Beijing at nighttime was about $2.5 \times 10^5$ cm$^3$ molecule$^{-1}$ (On the second figure). There is no specific concentration of ·OH at nighttime in winter in the study (Tan et al., 2018).

Moreover, the same ·OH concentration ($2.5 \times 10^5$ cm$^3$ molecule$^{-1}$) was also used to calculate the homogeneous reaction of HONO in the recent research (Zhang et al., 2019). And, nighttime OH concentration increased as the latitude decreases ranged 3 to $6 \times 10^5$ molecule cm$^{-3}$ (Lelieveld et al., 2016). Zhengzhou has a lower latitude than Beijing, so the concentration of OH used in this study is $2.5 \times 10^5$ cm$^3$ molecule$^{-1}$.

[Figure]

**Figure 2.** Nighttime OH in the boundary layer in January (top) and July (bottom). Color coding is the same as Fig. 1, but concentrations are scaled by a factor of 20 ($\times 0.05 \times 10^5$ molecules cm$^{-3}$).

[Figure]

**Figure 7.** Mean diurnal profiles of observed **(a)** and modeled **(b)** OH, HO$_2$, RO$_2$, and $k_{OH}$ for three different chemical and meteorological conditions. The categories for background, clean, and polluted episodes are the same as in Table 2, and similar to those applied to Figs. 9 and 12. The grey areas denote nighttime.

So, we have modified the sentences in the revised text.

L 281-288: Therefore, Tan et al. (2018) found that by the field measurement, the average concentration of ·OH in Beijing at nighttime was about $2.5 \times 10^5$ molecule cm$^{-3}$. Moreover, the same ·OH was also used to calculate the homogeneous reaction of HONO in the recent research (Zhang et al., 2019). And, nighttime OH concentration increased as the latitude decreases ranged 3 to $6 \times 10^5$ molecule cm$^{-3}$ (Lelieveld et al., 2016). Zhengzhou has a lower latitude than Beijing, so the concentration of OH used in this study is $2.5 \times 10^5$ molecule cm$^{-3}$.

The calculation of $P_{netOH+NO}$ should therefore be corrected using a more realistic OH concentrations. This may change the quantitative and relative contribution of homogeneous reaction to accumulated HONO formation at night. In this case, discussion and conclusion of the article on this point should also be revised consequently.

The calculation of $P_{OH+NO}^{net}$ has therefore been corrected using the OH concentration ($2.5 \times 10^5$ cm$^3$ molecule$^{-1}$). We have modified the sentence in the revised text.

L 293-295: The mean value of $P_{OH+NO}^{net}$ was 0.33 ppbv h$^{-1}$, and the specific values in CD, PD, and SPD periods were 0.13, 0.26, and 0.56 ppbv h$^{-1}$, respectively.

Finally, the discussion and conclusion of the article on this point were also revised consequently.

L 316-318: …Second, the hourly rate of HONO abatement pathways, except OH + HONO, should be at least 0.22 ppbv h$^{-1}$ (i.e., 3.36 – 1.59 ppbv)/8 h)…

L 549-550: The mean value of $P_{OH+NO}^{net}$ in the CD, PD, and SPD periods were 0.13, 0.26, and 0.56 ppbv h$^{-1}$, respectively.

3) A restructuration of section 3.3 is needed. Indeed all the paragraphs between the beginning of this section and the introductive paragraph for equations 4 to 6 (i.e. from P15, line 402 to P16, line 432) should be moved after these equations (i.e. eq. 4 to 6). Indeed, these paragraphs described the different terms used in the equation 4, 5 and 6 while they do not have been presented yet and this make the reading of this section very confusing.

Response: Sorry for my confusion. The equations 4 to 6 have been moved before all the paragraphs. We have modified the sentences in the revised text.

L 454-465: The expression of d HONO / d t represents the observed variations of hourly HONO concentrations, for which we can use Δ HONO/Δ d t instead:

d HONO / d t = sources − sinks

$$= (P_{unknown} + P_{OH+NO} + P_{emi} + P_{het}) - (L_{OH+HONO} + L_{photo}) \qquad (4),$$

$$P_{OH+NO} = k_{OH+NO} [OH] [NO] \qquad (5),$$

$$L_{OH+HONO} = k_{OH+HONO} [OH] [HONO] \qquad (6).$$

The d HONO / d t calculated from the measurements was small and evenly distributed around zero (Li et al., 2012). $P_{unknown}$ is the production rate by an unknown daytime HONO source. $P_{OH+NO}$ is the rate of reaction of NO and OH. $P_{emi}$ represents the direct emission rate of HONO from combustion processes. By studying the source and reduction, the daytime HONO budget was analyzed with Eq. (4) (Su et al., 2008).

4) P15, lines 403-404: "$P_{unknown}$ is the production rate by an unknown daytime HONO source". Please explain how $P_{unknown}$ is calculated. Do you assume that dHONO/dt is

equal to zero to do so? If it is the case, it should be indicated somewhere.

P17, lines 459-460: "However, further research is needed to analyze the unknown sources of daytime HONO". Why didn't you do it in this study? A deeper analysis of the processes that may be responsible for the observed unknown HONO production would have been valuable in this study. This further analysis is missing to strengthen the interest of this study for publication.

Response: Sorry for my careless. $P_{unknown}$ is calculated by:

$d\,HONO\,/\,d\,t = (P_{unknown} + P_{OH+NO} + P_{emi} + P_{het}) - (L_{OH+HONO} + L_{photo})$;

$P_{unknown} = L_{OH+HONO} + L_{photo} - P_{OH+NO} - P_{emi} - P_{het}$.

The sentence has been added in the revised text.

L 460-461: The $d\,HONO\,/\,d\,t$ calculated from the measurements was small and evenly distributed around zero (Li et al., 2012).

We have studied the correlation between the unknown source of HONO and the $PM_{2.5}$ mass concentrations was lower. So, we can not probably use the $P_{unknown}$ calculated to perform this correlation for explaining the unknown source. The unknown sources of HONO may include the $NO_2$ photolysis of sooty surface and adsorbed nitric acid and nitrate at UV wavelengths (Kleffmann et al., 1999). The homogeneous nucleation of $NO_2$, $H_2O$, and $NH_3$ is the HONO formation pathway (Zhang and Tao, 2010). In the meanwhile, HONO can deposit and react with amines in forming nitrosamines (Li et al., 2012) for sinking.

This further analysis and method are not found yet.

[Figure]

**Figure** The correlation between $PM_{2.5}$ and $P_{unknown}$.

**Minor comments**:

1. -P1, line 22: Change "(i.e., the concentration of NO…" for "(i.e., when the concentration of NO…".

    Response: OK. We have added the word, "When", in the revised text.

    L 21-23: …under high-$NO_X$ conditions (i.e., when the concentration of NO was higher than…

2. -P2, line 32: Change "The hourly abatement level of HONO abatement" for "The hourly level of HONO abatement".

    Response: Thank you. We have removed the word, "abatement", in the revised text.

    L 32-33: The hourly level of HONO abatement pathways, except for…

3. -P2, line 46: Change "OH radical is also an important oxidant" for "OH radical is an important oxidant".

Response: OK. We have removed the word, "also", in the revised text.

L 48-49:·OH is an important oxidant in the atmosphere, and it can react with organic substances…

4. -P2, lines 49-50: "Therefore, reaction changes during pollution can be observed by studying the formation mechanism of HONO". This sentence is not clear to me. Please clarify it or remove it.

Response: Sorry for the confusion. We explored the sources and characteristics of HONO at different pollution levels, as well as the reaction mechanism. We have not explained the reaction mechanism and pathways, so we have changed "reaction" for "the changes in the contribution of the homogeneous reaction, heterogeneous conversion, and direct emission". This sentence has been changed in the revised text.

L 51-54: Therefore, the changes in the contribution of the homogeneous reaction, heterogeneous conversion, and direct emission during pollution can be observed by studying the formation mechanism of HONO.

5. -P2, lines 53-54: "Nitro-Mac" is the name of the instrument but it does not described the technique of measurement. Please replace it by "wet chemical derivatization technique-HPLC/UV-VIS detection".

Response: Thank you for your comment. We have changed "Nitro-Mac" for "wet chemical derivatization technique-HPLC/UV-VIS detection". We have modified the sentence in the revised text.

L 58-69: …wet chemical derivatization technique-HPLC/UV-VIS detection…

6. -P3, line 55: The description of instruments existing for HONO measurements is not exhaustive. Important techniques such as IBBCEAS (e.g. Min et al., 2016; Duan et al., 2018) or CIMS (e.g. Hirokawa et al., 2009 ; Roberts et al., 2010) are missing. Please add them to your list.

Response: Sorry for my carelessness. We have analyzed and explored these techniques, and the important techniques have been added in the same sentence in the revised text.

L 55-62: Several instruments have been used to determine ambient HONO concentrations, and these include differential optical absorption spectrophotometer (DOAS) (Elshorbany et al., 2012; Winer and Biermann, 1994), long path absorption photometer (LOPAP) (Heland et al., 2001), wet chemical derivatization technique-HPLC/UV-VIS detection (Michoud et al., 2014), stripping coil-UV/Vis absorption photometer (SC-AP) (Pinto et al., 2014), IBBCEAS (Duan et al., 2018; Min et al., 2016), CIMS (Hirokawa et al., 2009; Roberts et al., 2010), and ambient ion monitor (AIM) (VandenBoer et al., 2014).

7. -P3, line 72: Change "be absorbed by" for "react with".

Response: OK. We have modified the sentence in the revised text.

L 76-77: …HONO can react with the ·OH…

8. -P5, lines 137-138: "The site is close to the West Fourth Ring Road". How far is it? Please be more precise.

Response: Sorry for my carelessness. We will be more precise in the full text and

examine the logic problems. The sentence has been changed in the revised text.

L 160-162: The site is about 500 m from the western Fourth-Ring Expressway of Zhengzhou City and about 2 km from Lian Huo Expressway to the north.

9. -P6, line 142: "High-Time-resolution instrument". A temporal resolution of 1h is not what is usually called high time resolution. Please change the title of this section.

Response: OK,we have changed "High-Time-resolution instrument" for "Instruments". And, the title has been modified, "Characteristics, sources, and reactions of nitrous acid during winter at an urban site in the Central Plains Economic Region in China", in the revised text.

10. -P6, line 153: Change "(e.g., O and N)" for "(e.g., O2 and N2)"

Response: Sorry for my carelessness. The sentence has been modified in the revised text.

L 74: …several gases (e.g., $O_2$ and $N_2$) were expelled…

11. -P7, lines 166-168: "The instrument parts and consumables should be changed regularly during the observation process, and the sampling flow should be calibrated to reduce the negative effect of accessories on sampling".
Could you be more specific? How often these maintenances have been made during the measurement period? What consumables exactly have been changed?
How is it compatible with the frequency of replacement given here and the frequency of calibration? Please clarify.

Response: OK. This is my omission. During the measurement, we have replaced the filter once a week and ensured enough hydrogen peroxide for absorbing

HONO by the denuder. The instrument parts and consumables should be changed before the observation process, and the sampling flow should be calibrated to reduce the negative effect of accessories. The sentence has been modified in the revised text.

L 189-191: The instrument parts and consumables should be changed before the observation process, and the sampling flow should be calibrated to reduce the negative effect of accessories.

And we have added the sentence in the revised text.

L 185-187: Before this measurement period, the membrane of the denuder has been replaced and standard anion and cation solutions have been prepared on Jan. 3rd.

The standard curve has been drawn to ensure the appropriateness of the correlation coefficient ($\geq 0.999$) and the accuracy of the sample retention time and response value. There is no need to stop the instrument during the replacement of the parts, and the calibration has been completed before the measurement period. The calibration can be used for one to two months at a time.

12. -P7, line 192: Wind direction is not presented in table 2. Please remove it from the list of parameters presented in table 2.

Response: Sorry. We have modified the table heading in the revised text.

**Table 1** Data statistics of HONO, $PM_{2.5}$, $NO_2$, NO, $NO_X$, $HONO/NO_2$, $HONO/NO_X$, $O_3$, CO, T, RH, and WS during the measurement period, mean value $\pm$ standard deviation.

13. -P8, line 217: Change "Fig. S3" for "Fig. 3".

The comparison of diurnal variation of HONO during the three period is given in Fig. 3 and not in Fig. S3. Fig. S3 concerns the whole measurement period. Once the

modification will bemade, there will be no reference in the article to Fig. S3. So please comment this figure in the text or remove it from the supplement.

Response: OK. We have put the diurnal variation of HONO during the entire period in **Fig. 3** and analyzed the diurnal variation of HONO in the three periods in **Fig. 4**(a) in the revised text. And, we have modified the sentence in the revised text.

L 232-234: The diurnal variations of HONO during the measurement were similar in the three periods, as shown in **Fig. 3** and **Fig. 4**.

[Figure]

**Fig. 3**. Diurnal variations of HONO during the measurement.

14. -P8, lines 217-218: "The NO and $NO_2$ concentration increased in the morning rush hours, decreased rapidly afterward, and remained low in the afternoon." This statement is not true for $NO_2$ and only right for NO during the CD period but not for the PD and SPD period. Please modify this statement consequently.

Response: Thank you for your comment. The sentence has been modified in the revised text.

L 242-243: The NO concentration decreased rapidly in the forenoon, and remained low in the afternoon.

15. -P10, line 251: Change "that cannot be obtained in the measurement" for "that was not measured during the campaign".

Response: OK. We have modified the sentence in the revised text.

L 275-276: [OH] is the concentration of ·OH that was not measured during the campaign.

16. -P10, line 253: Wrong unit: please change "cm$^3$ molecule$_{-1}$" for "molecule cm$^{-3}$".

Response: Sorry for my carelessness. The units have been modified in the full text.

L 282: …2.5×10$^5$ molecule cm$^{-3}$.

L 286:…3 to 6×10$^5$ molecule cm$^{-3}$…

L 493-498: The mean values of $J_{HONO}$ and ·OH concentration in the CD, PD, and SPD periods were 5.93×10$^{-4}$, 3.79×10$^{-4}$, and 3.79×10$^{-4}$ molecule cm$^{-3}$ and 4.10×10$^6$, 2.93×10$^6$, and 3.76×10$^6$ molecule cm$^{-3}$, respectively. The results of the calculated OH radicals ranged from (0.58−11.49) ×10$^6$ molecule cm$^{-3}$, and the mean value was 3.57 ×10$^6$ molecule cm$^{-3}$ at noon in Zhengzhou.

17. -P11, line 279: Change "the hourly abatement level of HONO abatement" for "the hourly level of HONO abatement".

Response: Thank you. We have removed the word, "abatement", in the revised

text.

L 316-317: Second, the hourly level of HONO abatement pathways, except OH + HONO, should be at least 0.22 ppbv h$^{-1}$ (i.e., 3.36 – 1.59 ppbv)/8 h).

18. -P11, lines 278-282: "Second, the hourly abatement level of HONO abatement pathways, except OH + HONO, should be at least 1.47 ppbv h$_{-1}$ (i.e., 13.41 – 1.59 ppbv) / 8 h). The contributions of other HONO abatement pathways in the current work even exceeded the formation of heterogeneous reactions, similar to a previous study (Spataro et al., 2013)."If this statement is maintained after the recalculation of $P_{netOH+NO}$ using a more realistic nocturnal OH concentrations, authors should comment on which other losses of HONO can be significant at night (e.g. deposition, heterogeneous losses…). At least, a raw estimation of loss by deposition could be performed to estimate whether it can explain the lacking abatement processes.

Response: Thank you for your comment. At night, in addition to reaction with HONO to OH, there were two HONO removal pathways: heterogeneous loss on aerosols and deposition (Li et al., 2012). The heterogeneous loss of aerosols can not be calculated directly. And, the main factor of the dry deposition on ground surfaces is the deposition velocity of HONO. The reported value of deposition velocity ranged from 0.092 or 2 cm s$^{-1}$ (Harrison et al., 1996; Stutz, 2002). Sorry, we can not give a raw estimation of loss by deposition, but what we can be sure of is that the phenomenon may arise because the dry deposition on ground surfaces can be the main HONO removal pathway at night.

So this is my confusion. This statement is maintained after the recalculation of $P_{OH+NO}^{net}$ using a more realistic nocturnal OH concentrations, the dry deposition on ground surfaces can be the HONO removal pathway at night. We have changed "The contributions of other HONO abatement pathways in the current work even exceeded the formation of heterogeneous reactions, similar to a previous study

(Spataro et al., 2013)." for "This phenomenon may arise because the dry deposition on ground surfaces can be the main HONO removal pathway at night, similar to a previous study (Li et al., 2012)." in the revised text (L 318-320).

19. -P13, lines 342-344: "The increased HONO in ambient air during the pollution period could have been caused by the comparatively high loading and large particle surface". The fair correlation between HONO concentrations and PM$_{2.5}$ mass concentrations may also just pinpoint the mainly anthropogenic origins of these two pollutants with high direct or indirect contribution of combustion sources for both of them and not the importance of HONO heterogeneous formation pathways on aerosol surfaces.

Response: Thank you. This is my carelessness. The fair correlation between HONO concentrations and PM$_{2.5}$ mass concentrations did not explain the importance of HONO heterogeneous formation pathways on aerosol surfaces. What we want to explain is whether there is a change in the intensity of NO$_2$ heterogeneous reactions during the increase in heavy pollution levels, so we found a relevant explanation (Cui et al., 2018). Cui et al. (2018) studied the more intense heterogeneous conversion of NO$_2$ to HONO on particle surfaces during the pollution episodes at a single particle scale. We have modified the sentences in the revised text.

L 382-386: The fair correlation between HONO and PM$_{2.5}$ may pinpoint the mainly anthropogenic origins of these two pollutants with the high direct or indirect contribution of combustion sources. The reason for the increased HONO during the heavy pollution period could be by the comparatively high loading and large particle surface (Cui et al., 2018).

A correlation between the calculated unknown source of HONO and the PM$_{2.5}$ mass

concentrations (as a proxy for aerosol surface even if it is not perfect) would have been more convincing. Authors can probably use the $P_{unknown}$ calculated in section 3.3 to perform this correlation.

We have studied the correlation between the unknown source of HONO and the $PM_{2.5}$ mass concentrations was lower. So, we can not probably use the $P_{unknown}$ calculated in section 3.3 to perform this correlation for explaining the unknown source.

[Figure]

20. -P14, line 383: Change "in then current study" for "in the current study".

Response: Sorry for my carelessness. The sentence has been modified in the revised text.

L 432: …HONO was calculated in the current study…

21. -P15, line 393: Change "the conversion rates" for "the averaged conversion rates".

Response: OK. The sentence has been modified in the revised text.

L 442-443: The averaged conversion rates…

22. -P15, lines 395-396: Change "The improvement" for "the increase".

Response: OK. The sentence has been modified in the revised text.

L 445: The increase in the conversion rate…

23. -P15, lines 398-399: "the high utilization efficiency of the aerosol surface due to good particle surface properties". I do not understand this statement. Please clarify and rephrase.

Response: Sorry for my confusion. The exact uptake coefficients of $NO_2$ on ground and aerosol surfaces are variable and should be different (Harrison and Collins, 1998). The present analysis simplified this process by treating the ground and aerosol surfaces the same. The uptake coefficient is mainly dependent on the surface characteristics, e.g. surface area, surface type (Lu et al., 2018). We have added the sentences in the revised text.

L 448-452: The exact uptake coefficients of $NO_2$ on ground and aerosol surfaces are variable and should be different (Harrison and Collins, 1998). The present analysis simplified this process by treating the ground and aerosol surfaces the same. The uptake coefficient is mainly dependent on the surface characteristics, e.g. surface type and moisture (Lu et al., 2018).

24. -P15-16, lines 415-418: "the tropospheric ultraviolet and visible (TUV) transfer model of the National Center for Atmospheric Research (http://cprm.acom.ucar.edu/Models/TUV/Interactive_TUV/) (Hou et al.,2016) was used to calculate the $J_{HONO}$ value". It should be addressed that the $J_{HONO}$ values obtained this way are only suitable for clear sky days without clouds, unless the presence of clouds have been taken into account. If so, the method used should be described. Furthermore, the values for $O_3$ column as well as for the surface albedo used in TUV

model should be indicated and justification about the choice of these values should be given.

Response: OK. Sorry for my carelessness. The problem you pointed out is correct. TUV is an interactive model for calculation of photodissociation coefficients (J values) over the visible and ultraviolet spectral range in the atmosphere under clear sky conditions. The $J_{HONO}$ values obtained this way were assumed in clear sky days without clouds. We would add a description of $O_3$ column and the surface albedo. $O_3$ column density measured by the Ozone Monitoring Instrument (OMI, data available at https://ozonewatch.gsfc.nasa.gov/data/omi/Y2019/). The $O_3$ column density ranges from 292 to 306 DU during the entire period. The experimental site being situated in an urban region, the surface albedo is considered as 0.13 (Sailor, 1995). The ground elevation and the measurement altitude are 168 and 188 m respectively.

So we have added the sentences in the revised text.

L 478-484: The $J_{HONO}$ values obtained this way were assumed in clear sky days without clouds. $O_3$ column and the surface albedo. $O_3$ column density measured by the Ozone Monitoring Instrument (OMI, data available at https://ozonewatch.gsfc.nasa.gov/data/omi/Y2019/). The $O_3$ column density ranges from 292 to 306 DU during the entire period. The experimental site being situated in an urban region, the surface albedo is considered as 0.13 (Sailor, 1995). The ground elevation and the measurement altitude are 168 and 188 m respectively.

25. -P16, lines 418-419: "The concentration of OH radicals was calculated with the formulas of $NO_2$, $O_3$, and $JO_1D$".Please specify the equation used for OH calculation.

Response: Thank you for your comment. This part of the formulas of $NO_2$, $O_3$, and $J_O{}^1{}_D$ has been described a lot in the paper (Rohrer and Berresheim, 2006). Sorry

for my carelessness. We have placed this part in the revised supplement.

**2. The concentration of OH radicals was calculated with the formulas of NO₂, O₃, and J$_O^1{}_D$.**

$$[\text{OH}] = \frac{k_{HO_2+NO}\tau_{HC}[NO_2]F_J}{k_{NO+O_3}} \times \sqrt{\frac{\alpha}{k_{HO_2+HO_2}[O_3]}} \times J(O^1D),$$

where [OH] represents the concentration of OH radicals, $k_{HO_2+NO} = 8.56 \times 10^{-12}$ cm³ s⁻¹, $\tau_{HC} = 0.3$ s, [NO₂] represents the NO₂ concentration, $F_J = 2$ s⁻⁰·⁵, $k_{NO+O_3} = 1.82 \times 10^{-14}$ cm³ s⁻¹, $\alpha = 0.075$, $k_{HO_2+HO_2} = 8.56 \times 10^{-12}$ cm³ s⁻¹, [O₃] represents the O₃ concentration, and $J(O^1D)$ represents the $O^1D$ efficiency of photolysis.

We have modified the sentence in the revised text.

L 484-485: The concentration of OH radicals was calculated with the formulas of NO₂, O₃, and J$_O^1{}_D$ in the supplement.

26. -P16, line 427: "The mean values of JHONO and OH radical concentration". Is it daily mean or mean values at noon? Please specify this.

Response: OK. TUV can only calculate the photolysis efficiency under daylight conditions. So, J$_{HONO}$ and ·OH concentration are the mean values at noon. To prevent this confusion, we have modified the sentence in the revised text.

L 495-500: The mean values of J$_{HONO}$ and ·OH concentration at noon in the CD, PD, and SPD periods were 5.93×10⁻⁴, 3.79×10⁻⁴, and 3.79×10⁻⁴ molecule cm⁻³ and 4.10×10⁶, 2.93×10⁶, and 3.76×10⁶ molecule cm⁻³, respectively."

27. -P17, lines 454-455: "Although the values of P$_{OH+NO}$ had high uncertainty because of the NO concentrations".How NO concentrations can affect largely the uncertainties of P$_{OH+NO}$ calculations? Does NO measurements suffer from high uncertainties? Why? If this is the case this point should be also addressed in the section 2.2. Please clarify

this statement.

Response: Sorry. This sentence is my expression problem. What I mean is that the concentration of NO has a great influence on it, but the homogeneous reaction is still an important pathway. The uncertainty of NO measurements was shown in **Table S1.**

So we have changed "Although the values of $P_{OH+NO}$ had high uncertainty because of the NO concentrations, $P_{OH+NO}$ contributed the most to HONO production during daytime." for "The concentration of NO has a great influence on $P_{OH+NO}$, so the homogeneous reaction is still an important pathway of HONO production during the daytime." in the revised text (L 516-518).

28. -Fig. 8: Please modify the legend of the figure to be consistent with the title and the manuscript (use PD and SPD instead of HD and SHD). Furthermore, $J_{HONO}$ and $J_{O1D}$ are shown only for two periods and not for all three. Why? Please include the values for the third period (SPD) or explain why it is not shown.

Response: OK. We have modified the problem in **Fig. 9**.

[Figure]

**Fig. 9**. The average profiles of $J_{HONO}$ and $J_{O^1D}$ concentrations during the daytime, and production and loss rate of the daytime HONO in CD, PD and SPD periods.

We treated PD and SPD the same. The reason is that the main input parameters of TUV cannot be obtained directly, so we quoted the input parameters in the literature. However, the input parameters of PD and SPD are not distinguished in the papers. We wanted to study that under the same output conditions from the TUV model, the impact of different pollution levels changed on the daytime budget. We have added the sentence in the revised text.

L 491-493: We wanted to study that under the same output conditions from the TUV model in the PD and SPD periods, the impact of different pollution levels changed on the daytime budget.

29. -Table 2: Please remove WD from the title of the table since no data of wind direction is shown in it.

Response: Sorry. We have removed the word, "WD", in the revised text.

Table 1. Data statistics of HONO, $PM_{2.5}$, $NO_2$, NO, $NO_X$, HONO/$NO_2$, HONO/$NO_X$, $O_3$, CO, T, RH, and WS during the measurement period, mean value ± standard deviation.

**Characteristics, sources, and reactions of nitrous acid during winter at an urban site in the Central Plains Economic Region in China**

Qi Hao, Nan Jiang*, Ruiqin Zhang, Liuming Yang, and Shengli Li

Key Laboratory of Environmental Chemistry and Low Carbon Technologies of Henan Province, Research Institute of Environmental Science, College of Chemistry, School of Ecology and Environment, Zhengzhou University, Zhengzhou 450001, China

**Supplement:**

**1. This AIM method and its details.**

HONO was hygroscopically grown in the parallel plate denuder and collected as an aqueous solution in a cyclone assembly. The aqueous sample aliquots from both channels were transported to the ion chromatographic systems housed inside a ground container for hourly semicontinuous online analysis of HONO. The ion chromatographic system was calibrated for $NO_2^-$ using mixed anion standard solutions of $NO_2^-$.

**2. The concentration of OH radicals was calculated with the formulas of $NO_2$, $O_3$, and $J_{O^1D}$.**

$$[OH] = \frac{k_{HO_2+NO}\tau_{HC}[NO_2]F_J}{k_{NO+O_3}} \times \sqrt{\frac{\alpha}{k_{HO_2+HO_2}[O_3]}} \times J(O^1D),$$

where [OH] represents the concentration of OH radicals, $k_{HO_2+NO} = 8.56 \times 10^{-12}$ cm$^3$ s$^{-1}$, $\tau_{HC} = 0.3$ s, $[NO_2]$ represents the $NO_2$ concentration, $F_J = 2$ s$^{-0.5}$, $k_{NO+O_3} = 1.82 \times 10^{-14}$ cm$^3$ s$^{-1}$, $\alpha = 0.075$, $k_{HO_2+HO_2} = 8.56 \times 10^{-12}$ cm$^3$ s$^{-1}$, $[O_3]$ represents the $O_3$ concentration, and $J(O^1D)$ represents the $O^1D$ efficiency of photolysis.

**Figure Captions:**

Fig. S1. The correlation study between $HONO_{correct}$ and $NO_2$ in the nighttime.

[Figure]

**Fig. S1**. The correlation study between HONO_correct and NO₂ in the nighttime.

**Table Captions:**

Table S1. Measured species and performance of the instruments.

Table S2 The error bars of Fig. 4. (The units of all species except $HONO/NO_2$ and $HONO/NO_x$ are ppbv. The units of $HONO/NO_2$ and $HONO/NO_x$ are %.)

Table S3 The error bars of Fig. 5. (The units of all species except $P_{OH+NO}^{net}$ are ppbv. The unit of $P_{OH+NO}^{net}$ is ppbv/h.)

Table S4 The error bars of Fig. 8. (The units of all species except $HONO_{correct}/NO_2$ are ppbv. The unit of $HONO_{correct}/NO_2$ is %.)

**Table S1.** Measured species and performance of the instruments.

| Species | Measurement technique | Detection limit | Accuracy |
|---------|----------------------|-----------------|----------|
| $PM_{2.5}$ | Tapered Element Oscillating Microbalance | 1.5 μg m$^{-3}$ | ± 5% |
| HONO | Ion Chromatography | 4 pptv | ± 20% |
| CO | Absorbs Infrared Radiation | 40 ppbv | ± 5% |
| NO | Chemiluminescence | 60 pptv | ± 20% |
| $NO_2$ | Chemiluminescence | 300 pptv | ± 20% |
| $O_3$ | UV Photometry | 0.5 ppbv | ± 5% |

The results came from instrument manufacturers.

**Table S2**-1 The error bars of Fig. 4. (The units of all species except $HONO/NO_2$ and $HONO/NO_x$ are ppbv. The units of $HONO/NO_2$ and $HONO/NO_x$ are %.)

| Species-period | Local Time (hh:mm) | | | | | | | | | |
|---|---|---|---|---|---|---|---|---|---|---|
| | 00:00 | 01:00 | 02:00 | 03:00 | 04:00 | 05:00 | 06:00 | 07:00 | 08:00 | 09:00 |
| HONO-CD | 1.7 ± 1.3 | 1.4 ± 0.6 | 1.3 ± 0.4 | 1.2 ± 0.3 | 1.2 ± 0.2 | 1.2 ± 0.2 | 1.4 ± 0.3 | 1.5 ± 0.6 | 1.7 ± 0.9 | 1.6 ± 0.9 |
| HONO-PD | 3.2 ± 1.5 | 3.1 ± 1.3 | 3 ± 1.1 | 3.3 ± 1.2 | 3.5 ± 1.3 | 3.5 ± 1.2 | 3.6 ± 1.1 | 3.3 ± 0.9 | 3.7 ± 1.6 | 4.1 ± 2.8 |
| HONO-SPD | 3.7 ± 0.9 | 4 ± 0.8 | 4.2 ± 0.6 | 4.4 ± 0.8 | 4.6 ± 1 | 4.6 ± 1.2 | 4.6 ± 1.5 | 4.4 ± 1.3 | 4.4 ± 1.1 | 5.7 ± 3 |
| NO-CD | 14.3 ± 17 | 9 ± 9.7 | 8.5 ± 12.7 | 10.1 ± 22.4 | 10.6 ± 21.1 | 21.9 ± 29 | 27.8 ± 33 | 40.1 ± 51 | 52.6 ± 79 | 55.5 ± 84 |
| NO-PD | 57.3 ± 48 | 62.7 ± 55.9 | 49.6 ± 49 | 44 ± 47.8 | 47 ± 48.7 | 46.6 ± 30 | 41.4 ± 34 | 44.7 ± 33 | 48.9 ± 35 | 53.7 ± 44 |
| NO-SPD | 79.4 ± 103 | 100.1 ± 118 | 128.3 ±133 | 129 ± 134 | 111 ± 119 | 117 ± 95 | 100 ± 94 | 88.4 ± 85 | 82.3 ± 70 | 85.4 ± 71 |
| $NO_2$-CD | 25.4 ± 8.2 | 25.6 ± 9.9 | 24.7 ± 10.5 | 22.9 ± 10.4 | 24 ± 11.4 | 20.7 ± 11 | 20.2 ± 9 | 23.6 ± 11 | 28.6 ± 18 | 28.6 ± 18 |
| $NO_2$-PD | 41.1 ± 10 | 40.8 ± 11.2 | 39.7 ± 10.7 | 37.9 ± 7.1 | 36.6 ± 5.4 | 35.9 ± 5 | 33.8 ± 6 | 34.4 ± 6 | 33.2 ± 5 | 30.7 ± 6 |
| $NO_2$-SPD | 45.3 ± 9.5 | 43.5 ± 9.2 | 42.8 ± 8.8 | 42.1 ± 8.2 | 42.2 ± 8.1 | 41 ± 7.1 | 40.6 ± 6.9 | 40.7 ± 6 | 40.1 ± 6 | 39.2 ± 7 |
| $O_3$-CD | 14.2 ± 10 | 13.6 ± 10.4 | 14.2 ± 10.1 | 14.9 ± 9.4 | 13.6 ± 9.1 | 11.7 ± 10 | 13.8 ± 10 | 12.9 ± 9 | 11.6 ± 8 | 12.1 ± 7 |
| $O_3$-PD | 6.6 ± 6.1 | 6.4 ± 5.2 | 7.1 ± 5.2 | 6.3 ± 3.3 | 4.7 ± 2.2 | 5.3 ± 3 | 7.7 ± 6.9 | 5.3 ± 2.8 | 5.5 ± 3 | 7.1 ± 4 |
| $O_3$-SPD | 7.8 ± 6.4 | 7.7 ± 6.2 | 7.3 ± 5 | 6 ± 2.9 | 5.3 ± 2.3 | 5 ± 2.1 | 5.6 ± 2.5 | 5.2 ± 2.2 | 5.6 ± 2.6 | 6 ± 2.6 |
| $HONO/NO_2$-CD | 3.8 ± 1.5 | 4.4 ± 1 | 4.4 ± 1.1 | 4.9 ± 1 | 5.1 ± 0.8 | 8.3 ± 6 | 6.9 ± 2.1 | 6.2 ± 1.4 | 5.1 ± 0.8 | 4.3 ± 1.1 |
| $HONO/NO_2$-PD | 8 ± 3.6 | 7.8 ± 3.4 | 8 ± 3.3 | 9 ± 3.7 | 10 ± 4.5 | 10.1 ± 4 | 11.2 ± 4.6 | 10.3 ± 4 | 12.1 ± 7 | 14.3 ± 11 |
| $HONO/NO_2$-SPD | 8.3 ± 1.9 | 9.3 ± 1.4 | 10 ± 1.5 | 10.7 ± 1.9 | 11 ± 2.2 | 11.3 ± 3 | 11.5 ± 3.9 | 10.9 ± 3 | 11.1 ± 2 | 15 ± 8.3 |
| $HONO/NO_x$-CD | 2.7 ± 1.4 | 3.7 ± 1.5 | 4.2 ± 1.4 | 4.9 ± 1.1 | 4.9 ± 1 | 5.3 ± 2.5 | 5.1 ± 2.9 | 4.5 ± 2.4 | 3.6 ± 1.5 | 2.8 ± 1.4 |
| $HONO/NO_x$-PD | 4.4 ± 1.4 | 4.3 ± 1.7 | 4.6 ± 1.5 | 5.3 ± 1.3 | 5.3 ± 1 | 5.3 ± 1.1 | 6.6 ± 2.7 | 5.9 ± 2.3 | 6.5 ± 3.8 | 6.6 ± 4.3 |
| $HONO/NO_x$-SPD | 5.1 ± 2 | 5.3 ± 2.4 | 5.4 ± 3.4 | 5.8 ± 3.9 | 6.1 ± 3.9 | 5.7 ± 3.7 | 5.9 ± 3.6 | 5.7 ± 3 | 5.8 ± 2.9 | 6.7 ± 3.1 |

**Table S2**-2 The error bars of Fig. 4. (The units of all species except HONO/NO$_2$ and HONO/NO$_x$ are ppbv. The units of HONO/NO$_2$ and HONO/NO$_x$ are %.)

| Species-period | \multicolumn{10}{c}{Local Time (hh:mm)} | | | | | | | | | |
| --- | --- | --- | --- | --- | --- | --- | --- | --- | --- | --- |
| | 10:00 | 11:00 | 12:00 | 13:00 | 14:00 | 15:00 | 16:00 | 17:00 | 18:00 | 19:00 |
| HONO-CD | 1.1 ± 0.6 | 0.6 ± 0.3 | 0.5 ± 0.3 | 0.6 ± 0.4 | 0.6 ± 0.5 | 0.7 ± 0.5 | 0.6 ± 0.5 | 0.7 ± 0.4 | 1 ± 0.5 | 1.2 ± 0.5 |
| HONO-PD | 2.9 ± 1.9 | 1.9 ± 1.3 | 1.3 ± 0.7 | 1 ± 0.3 | 0.9 ± 0.3 | 0.9 ± 0.3 | 0.9 ± 0.3 | 1.1 ± 0.4 | 1.4 ± 0.3 | 1.7 ± 0.3 |
| HONO-SPD | 6.9 ± 4.3 | 5.2 ± 3.8 | 3 ± 1.3 | 2.1 ± 0.7 | 1.8 ± 0.7 | 1.7 ± 0.6 | 1.8 ± 0.7 | 2 ± 0.5 | 2.7 ± 0.7 | 2.8 ± 0.8 |
| NO-CD | 43.9 ± 69.8 | 27.9 ± 40.8 | 14.9 ± 17.1 | 10.3 ± 7.8 | 7.3 ± 3 | 6 ± 4.5 | 6.4 ± 5.6 | 3.6 ± 3.4 | 2.6 ± 3.2 | 5.9 ± 7.7 |
| NO-PD | 49.3 ± 45.2 | 30 ± 26.2 | 21 ± 20.7 | 12.7 ± 14.7 | 9.4 ± 12.3 | 8.4 ± 9.5 | 5.7 ± 4.7 | 6.3 ± 6.8 | 9 ± 9 | 10 ± 10.3 |
| NO-SPD | 90.8 ± 73.4 | 79.3 ± 69.3 | 57.1 ± 52.3 | 34.8 ± 36.4 | 24.5 ± 28.7 | 19 ± 24.7 | 15 ± 18.8 | 11.8 ± 11 | 11.8 ± 7.9 | 22.4 ± 21 |
| NO$_2$-CD | 26.8 ± 15.7 | 22.7 ± 9.2 | 17.6 ± 7.1 | 17.1 ± 9 | 19.6 ± 9.6 | 21 ± 10.7 | 20.5 ± 9 | 21.4 ± 9 | 26 ± 12.5 | 30 ± 13.7 |
| NO$_2$-PD | 30 ± 6.9 | 28.8 ± 7.7 | 27.4 ± 9.6 | 24.8 ± 9.4 | 22.5 ± 10.6 | 25 ± 9.9 | 25.7 ± 9.3 | 27.1 ± 9 | 35 ± 8.7 | 36.2 ± 9.2 |
| NO$_2$-SPD | 39.8 ± 7.8 | 41.5 ± 8.3 | 42.3 ± 10.1 | 39.5 ± 12.6 | 38.5 ± 14.3 | 38 ± 14.7 | 38 ± 13.9 | 42 ± 15.4 | 45 ± 11.5 | 47 ± 10.8 |
| O$_3$-CD | 15.9 ± 8.8 | 19.5 ± 9.7 | 22.6 ± 8.3 | 25.5 ± 8.5 | 28.1 ± 9.1 | 29 ± 10.8 | 28 ± 10.8 | 29 ± 10.2 | 23.6 ± 10 | 17 ± 8.9 |
| O$_3$-PD | 9.6 ± 6.1 | 12.8 ± 6.2 | 18.7 ± 8.3 | 24.1 ± 8.4 | 28.2 ± 9.7 | 27 ± 10.8 | 28 ± 10.4 | 26 ± 10.5 | 17.4 ± 8.6 | 15 ± 11.6 |
| O$_3$-SPD | 6.3 ± 2.4 | 8.7 ± 4.5 | 12.8 ± 8.5 | 19.4 ± 12.9 | 24.1 ± 14.7 | 28 ± 16.6 | 29 ± 17.6 | 25 ± 16.1 | 17 ± 11.1 | 10.6 ± 9.7 |
| HONO/NO$_2$-CD | 4.1 ± 2.3 | 3.1 ± 1.9 | 3.3 ± 1.9 | 3.3 ± 1.3 | 3.1 ± 1.3 | 3.1 ± 1.3 | 2.9 ± 1.4 | 3.1 ± 1.4 | 3.9 ± 1.4 | 4.5 ± 2.2 |
| HONO/NO$_2$-PD | 9.4 ± 5.6 | 6.2 ± 3 | 4.7 ± 1.5 | 4.2 ± 1.2 | 4.7 ± 2.2 | 3.9 ± 0.7 | 3.7 ± 0.4 | 4.1 ± 1.2 | 4.3 ± 0.9 | 5 ± 1.5 |
| HONO/NO$_2$-SPD | 18.9 ± 13.7 | 13.7 ± 12 | 7.3 ± 3.5 | 5.6 ± 2.6 | 4.9 ± 2.1 | 4.8 ± 2.4 | 4.9 ± 1.6 | 5 ± 1 | 6.3 ± 1.8 | 6.2 ± 1.5 |
| HONO/NO$_x$-CD | 2.9 ± 2.1 | 2.2 ± 1.5 | 2.4 ± 1.5 | 2.5 ± 1.1 | 2.5 ± 1 | 2.6 ± 0.9 | 2.5 ± 0.9 | 2.8 ± 1 | 3.7 ± 1.1 | 4.1 ± 1.9 |
| HONO/NO$_x$-PD | 4.8 ± 2.4 | 3.8 ± 1.3 | 3.5 ± 1.2 | 3.5 ± 1.5 | 4 ± 2.1 | 3.4 ± 0.9 | 3.3 ± 0.5 | 3.7 ± 1.2 | 3.8 ± 0.7 | 4.3 ± 1.5 |
| HONO/NO$_x$-SPD | 8.2 ± 5.8 | 6.9 ± 5.7 | 4.3 ± 2 | 4 ± 2 | 3.8 ± 1.6 | 3.9 ± 1.9 | 4.3 ± 1.6 | 4.5 ± 1.2 | 5.5 ± 1.5 | 4.9 ± 1.3 |

**Table S2**-3 The error bars of Fig. 4. (The units of all species except $HONO/NO_2$ and $HONO/NO_x$ are ppbv. The units of $HONO/NO_2$ and $HONO/NO_x$ are %.)

| Species-period | Local Time (hh:mm) | | | |
| --- | --- | --- | --- | --- |
| | 20:00 | 21:00 | 22:00 | 23:00 |
| HONO-CD | $1.3 \pm 0.6$ | $1.6 \pm 0.9$ | $2 \pm 0.9$ | $2.1 \pm 0.9$ |
| HONO-PD | $1.7 \pm 0.7$ | $1.8 \pm 0.8$ | $2 \pm 0.9$ | $2.1 \pm 0.9$ |
| HONO-SPD | $3.1 \pm 0.9$ | $3.2 \pm 0.9$ | $3.7 \pm 0.8$ | $4.6 \pm 1.2$ |
| NO-CD | $11.1 \pm 16.9$ | $14.5 \pm 22.5$ | $35.5 \pm 68.9$ | $50.8 \pm 99.2$ |
| NO-PD | $15 \pm 14.1$ | $15.3 \pm 14.7$ | $27.4 \pm 28.5$ | $33.9 \pm 28.9$ |
| NO-SPD | $29.4 \pm 24.2$ | $37.3 \pm 26.6$ | $38.5 \pm 23.1$ | $51.4 \pm 31.4$ |
| $NO_2$-CD | $31 \pm 13.8$ | $30.3 \pm 14.5$ | $31.6 \pm 13.6$ | $31 \pm 14.3$ |
| $NO_2$-PD | $37.3 \pm 10.5$ | $38.5 \pm 13.9$ | $38.3 \pm 13.5$ | $37.1 \pm 13.2$ |
| $NO_2$-SPD | $44.5 \pm 11$ | $43.5 \pm 11.5$ | $43.5 \pm 11.1$ | $42.1 \pm 13.1$ |
| $O_3$-CD | $13.3 \pm 10.1$ | $14 \pm 11$ | $12.2 \pm 8.7$ | $12.7 \pm 8.8$ |
| $O_3$-PD | $13.7 \pm 10.3$ | $10.9 \pm 8.5$ | $10.9 \pm 7.7$ | $12.2 \pm 10.4$ |
| $O_3$-SPD | $9.9 \pm 8.6$ | $10.8 \pm 9.2$ | $9.7 \pm 8.7$ | $9.6 \pm 9.6$ |
| $HONO/NO_2$-CD | $4.6 \pm 2.2$ | $5.7 \pm 2.6$ | $6.5 \pm 2.6$ | $6.8 \pm 2.7$ |
| $HONO/NO_2$-PD | $4.7 \pm 1.9$ | $4.6 \pm 1.2$ | $4.9 \pm 0.8$ | $5.3 \pm 0.8$ |
| $HONO/NO_2$-SPD | $7 \pm 1.5$ | $7.5 \pm 1.4$ | $8.9 \pm 2.3$ | $9.4 \pm 2.4$ |
| $HONO/NO_x$-CD | $4 \pm 1.9$ | $4.8 \pm 2.2$ | $4.9 \pm 2.8$ | $5 \pm 3$ |
| $HONO/NO_x$-PD | $3.9 \pm 2.1$ | $3.9 \pm 1.3$ | $3.8 \pm 1$ | $3.8 \pm 0.9$ |
| $HONO/NO_x$-SPD | $5.1 \pm 1.5$ | $5.2 \pm 2$ | $5.8 \pm 2$ | $5 \pm 1.4$ |

**Table S3**-1 The error bars of Fig. 5. (The units of all species except $P_{OH+NO}^{net}$ are ppbv. The unit of $P_{OH+NO}^{net}$ is ppbv/h.)

| Species-period | Local Time (hh:mm) | | | | | | | | | |
|---|---|---|---|---|---|---|---|---|---|---|
| | 19:00 | 20:00 | 21:00 | 22:00 | 23:00 | 00:00 | 01:00 | 02:00 | 03:00 | 04:00 |
| $P_{OH+NO}^{net}$-CD | $0.04 \pm 0.06$ | $0.08 \pm 0.12$ | $0.11 \pm 0.17$ | $0.33 \pm 0.54$ | $0.47 \pm 0.79$ | $0.12 \pm 0.13$ | $0.07 \pm 0.08$ | $0.03 \pm 0.03$ | $0.01 \pm 0.1$ | $0.02 \pm 0.1$ |
| HONO-CD | $1.18 \pm 0.48$ | $1.32 \pm 0.62$ | $1.62 \pm 0.9$ | $2.02 \pm 0.94$ | $2.09 \pm 0.9$ | $1.67 \pm 1.34$ | $1.43 \pm 0.63$ | $1.26 \pm 0.44$ | $1.2 \pm 0.3$ | $1.2 \pm 0.22$ |
| NO-CD | $5.4 \pm 6.5$ | $10.2 \pm 14.4$ | $13.3 \pm 19.2$ | $38.2 \pm 62.2$ | $54.9 \pm 89.7$ | $15 \pm 14.8$ | $8.8 \pm 8.6$ | $3.7 \pm 4.2$ | $1.5 \pm 2.3$ | $2.5 \pm 2.6$ |
| $P_{OH+NO}^{net}$-HD | $0.07 \pm 0.07$ | $0.1 \pm 0.1$ | $0.1 \pm 0.1$ | $0.19 \pm 0.2$ | $0.23 \pm 0.2$ | $0.4 \pm 0.34$ | $0.44 \pm 0.4$ | $0.34 \pm 0.35$ | $0.3 \pm 0.34$ | $0.3 \pm 0.34$ |
| HONO-HD | $1.7 \pm 0.27$ | $1.71 \pm 0.68$ | $1.82 \pm 0.78$ | $1.98 \pm 0.89$ | $2.06 \pm 0.93$ | $3.21 \pm 1.54$ | $3.05 \pm 1.27$ | $3.01 \pm 1.08$ | $3.3 \pm 1.17$ | $3.5 \pm 1.34$ |
| NO-HD | $8.5 \pm 8.4$ | $12.2 \pm 11.5$ | $12.5 \pm 12$ | $22.4 \pm 23.3$ | $27.7 \pm 23.6$ | $46.8 \pm 39.5$ | $51.2 \pm 45.6$ | $40.5 \pm 40$ | $35.9 \pm 39$ | $38 \pm 39.7$ |
| $P_{OH+NO}^{net}$-SHD | $0.15 \pm 0.15$ | $0.2 \pm 0.17$ | $0.25 \pm 0.18$ | $0.26 \pm 0.16$ | $0.35 \pm 0.23$ | $0.55 \pm 0.75$ | $0.7 \pm 0.85$ | $0.9 \pm 0.96$ | $0.9 \pm 1.0$ | $0.8 \pm 0.86$ |
| HONO-SHD | $2.8 \pm 0.8$ | $3.1 \pm 0.9$ | $3.2 \pm 0.9$ | $3.7 \pm 0.8$ | $4.6 \pm 1.2$ | $3.7 \pm 0.9$ | $4 \pm 0.8$ | $4.2 \pm 0.6$ | $4.4 \pm 0.8$ | $4.6 \pm 1$ |
| NO-SHD | $18 \pm 17$ | $24 \pm 20$ | $30 \pm 21$ | $31 \pm 19$ | $42 \pm 25$ | $64 \pm 84$ | $81 \pm 96$ | $104 \pm 108$ | $105 \pm 110$ | $90 \pm 97$ |

**Table S3**-2 The error bars of Fig. 5. (The units of all species except $P_{OH+NO}^{net}$ are ppbv. The unit of $P_{OH+NO}^{net}$ is ppbv/h.)

| Species-period | Local Time (hh:mm) | |
|---|---|---|
| | 05:00 | 06:00 |
| $P_{OH+NO}^{net}$-CD | 0.12 ± 0.18 | 0.17 ± 0.22 |
| HONO-CD | 1.25 ± 0.21 | 1.36 ± 0.35 |
| NO-CD | 13.7 ± 20.9 | 19.5 ± 25.1 |
| $P_{OH+NO}^{net}$-HD | 0.32 ± 0.22 | 0.28 ± 0.25 |
| HONO-HD | 3.5 ± 1.16 | 3.56 ± 1.09 |
| NO-HD | 38 ± 25.2 | 33.8 ± 28.5 |
| $P_{OH+NO}^{net}$-SHD | 0.82 ± 0.87 | 0.7 ± 0.68 |
| HONO-SHD | 4.6 ± 1.2 | 4.6 ± 1.5 |
| NO-SHD | 95.6 ± 99 | 81.8 ± 77.1 |

**Table S4**-1 The error bars of Fig. 8. (The units of all species except $HONO_{correct}/NO_2$ are ppbv. The unit of $HONO_{correct}/NO_2$ is %.)

| Species-period | \multicolumn{10}{c}{Local Time (hh:mm)} | | | | | | | | | |
|---|---|---|---|---|---|---|---|---|---|---|
| | 19:00 | 20:00 | 21:00 | 22:00 | 23:00 | 00:00 | 01:00 | 02:00 | 03:00 | 04:00 |
| $HONO_{correct}$-CD | 1.0 ± 0.4 | 1.1 ± 0.6 | 1.4 ± 0.8 | 1.6 ± 0.7 | 1.6 ± 0.6 | 1.4 ± 1.4 | 1.2 ± 0.7 | 1.1 ± 0.5 | 1.1 ± 0.4 | 1.1 ± 0.2 |
| $NO_2$-CD | 30 ± 15 | 31 ± 15 | 30 ± 15 | 34 ± 15 | 34 ± 15 | 25 ± 9 | 24 ± 8 | 22 ± 8 | 20 ± 8 | 20 ± 8 |
| $HONO_{correct}/NO_2$-CD | 3.7 ± 2.2 | 3.9 ± 2.2 | 4.9 ± 2.6 | 5.5 ± 2.7 | 5.7 ± 2.9 | 11 ± 18.2 | 8.9 ± 12 | 8.6 ± 10.8 | 8.5 ± 9.7 | 7.7 ± 7.4 |
| $HONO_{correct}$-HD | 1.4 ± 0.3 | 1.4 ± 0.7 | 1.5 ± 0.7 | 1.6 ± 0.8 | 1.7 ± 0.8 | 2.7 ± 1.3 | 2.5 ± 1 | 2.5 ± 0.8 | 2.9 ± 0.9 | 3.1 ± 1.1 |
| NO2-HD | 36 ± 9 | 37 ± 10 | 39 ± 14 | 38 ± 13 | 37 ± 13 | 41 ± 10 | 41 ± 11 | 40 ± 11 | 38 ± 7 | 37 ± 5 |
| $HONO_{correct}/NO_2$-HD | 4.2 ± 1.5 | 3.8 ± 2 | 3.8 ± 1.2 | 4 ± 0.8 | 4.4 ± 0.7 | 6.7 ± 3.1 | 6.5 ± 2.8 | 6.7 ± 2.8 | 7.8 ± 3.1 | 8.7 ± 3.8 |
| $HONO_{correct}$-SHD | 2.4 ± 0.6 | 2.6 ± 0.7 | 2.7 ± 0.7 | 3.2 ± 0.7 | 4.1 ± 1.3 | 3.1 ± 0.8 | 3.3 ± 0.6 | 3.4 ± 0.7 | 3.6 ± 1 | 3.9 ± 1.1 |
| $NO_2$-SHD | 47 ± 11 | 44 ± 11 | 43 ± 11 | 44 ± 11 | 42 ± 13 | 45 ± 9 | 43 ± 9 | 43 ± 9 | 42 ± 8 | 42 ± 8 |
| $HONO_{correct}/NO_2$-SHD | 5.4 ± 1.4 | 6.1 ± 1.4 | 6.5 ± 1.4 | 7.8 ± 2.2 | 14.4 ± 16.7 | 7 ± 1.9 | 7.8 ± 1.6 | 8.1 ± 2.2 | 8.8 ± 2.8 | 9.3 ± 2.9 |

**Table S4**-2 The error bars of Fig. 8. (The units of all species except $HONO_{correct}/NO_2$ are ppbv. The unit of $HONO_{correct}/NO_2$ is %.)

| | Local Time (hh:mm) | |
|---|---|---|
| Species-period | 05:00 | 06:00 |
| $HONO_{correct}$-CD | $1.0 \pm 0.4$ | $1.1 \pm 0.6$ |
| $NO_2$-CD | $30 \pm 15$ | $31 \pm 15$ |
| $HONO_{correct}/NO_2$-CD | $3.7 \pm 2.2$ | $3.9 \pm 2.2$ |
| $HONO_{correct}$-HD | $1.4 \pm 0.3$ | $1.4 \pm 0.7$ |
| $NO_2$-HD | $36 \pm 9$ | $37 \pm 10$ |
| $HONO_{correct}/NO_2$-HD | $4.2 \pm 1.5$ | $3.8 \pm 2$ |
| $HONO_{correct}$-SHD | $2.4 \pm 0.6$ | $2.6 \pm 0.7$ |
| $NO_2$-SHD | $47 \pm 11$ | $44 \pm 11$ |
| $HONO_{correct}/NO_2$-SHD | $5.4 \pm 1.4$ | $6.1 \pm 1.4$ |

**Reference**

[revised manuscript text omitted]

---

## Referee Report (RR1)

**Manuscript title**: Characteristics, sources, and reactions of nitrous acid during winter at an urban site in the Central Plains Economic Region in China

Authors: Hao et al.

https://doi.org/10.5194/acp-2019-916

The authors have addressed almost all of my concerns from the previous version, as well as those of the other reviewer. I am, however, still concerned with the nighttime concentrations of OH used for the net nocturnal production of HONO by homogeneous reaction. Indeed, the authors now use $2.5 \times 10^5$ molecule cm$^{-3}$ for nocturnal OH concentrations which is still very high. This value is deduced from measurements made in Beijing by Tan et al. (2018). However Tan et al. (2018) declare a limit of detection of $4 \times 10^5$ molecule cm$^{-3}$ for their OH LIF instrument, and quantification below this value during nighttime should be taken with caution especially when you look at the variability of these data at low concentrations for the whole campaign (see Fig. 5 of Tan et al., 2018). Furthermore, the authors support their choice by the nocturnal OH winter concentrations of 3 to $6 \times 10^5$ molecule cm$^{-3}$ estimated by the global model EMAC in the area of the study (Lelieveld et al., 2016), although in the work of Lelieveld et al. (2016) a factor of 0.05 is associated to these values and the nocturnal OH winter concentrations for the area is rather of 1.5 to $3 \times 10^4$ molecule cm$^{-3}$. I therefore recommend to redo all the calculations that use the erroneous wintertime nocturnal OH concentrations of $2.5 \times 10^5$ molecule cm$^{-3}$ (too high of a factor of 20), and to modify the discussion and conclusion if necessary.

---

## Referee Report (RR2)

**Referee Report:**
**Characteristics, sources, and reactions of nitrous acid during winter at an urban site in the Central Plains Economic Region in China**

Anonymous

**1 Overview**

The manuscript covers observations of nitrous acid (HONO) in an urban area of China during winter time. The observational data were divided into three categories according to the pollution levels marked by $PM_{2.5}$. Different nocturnal sources of HONO were investigated by using observational data (HONO, NO, $NO_2$, $O_3$) and estimated data (OH). Daytime HONO budget analysis reveals a dominant contribution to HONO production from $NO + OH$ reaction followed by an unknown production channel. This manuscript is within the scope of ACP. I recommend that the manuscript be published in ACP after major revision.

**2 Main comments**

(1) 3.3. Daytime HONO budget

    (a) Line 460–461: "The dHONO/dt calculated from the measurements was small and evenly distributed around zero (Li et al., 2012)": this sentence is directly copied from Li et al. (2012), please paraphrase.

    (b) According to Fig. 9, the values of dHONO/dt calculated in this study obviously do not show the same pattern as in Li et al. (2012). For example, the values of dHONO/dt are dominantly above zero during 10:00–13:00 local time of the PD period. It would be interesting to see a discussion on the difference between this study and Li et al. (2012).

    (c) In order to calculate the daytime budget of HONO using Eqn. (4), the chemical lifetime of HONO should be short enough (10–20 min) such that HONO is considered to be in quasi steady state (Kleffmann et al., 2005; Acker et al., 2006; Li et al., 2012). Please provide proof to show that such condition is satisfied in this study.

    (d) The values of dHONO/dt in Fig. 9 do not match the hourly HONO concentrations shown in Fig. 4 (assuming data in Fig. 4 is used to calculate dHONO/dt in Fig. 9). Take SPD period for example, the HONO concentrations are 4.4, 5.7, and 6.9 ppbv at 08:00, 09:00, and 10:00 local time, respectively. This will produce a dHONO/dt value of about 1.2 ppbv/h at 09:00 local time. However, in Fig. 9, the dHONO/dt value is at around $-1.5$ ppbv/h at 09:00 local time of the SPD period. Please describe in detail on how the values of dHONO/dt are calculated. This

is critical since dHONO/dt is used to calculate $P_{unknown}$, which is discussed extensively in the daytime HONO budget section.

(2) The writing of the manuscript generally flows in logical structure. However, there are many places where the description is a bit awkward or incorrect (see the technical corrections section). The authors should review the manuscript carefully or have it edited by a professional expert before final submission.

**3 Minor comments**

(1) Line 88: the text states that the $H_2O$ in R4 is isotope-labeled, please label it accordingly.

(2) Line 119: the linked webpage shows development guidelines, please replace it with reference (with data) published in scientific journal, preferably in English, to backup the statement that Zhengzhou is an ideal place to carry out such study.

(3) Line 156: Please provide a satellite map, either in the main text or in the supplement, to show the location of the sampling site and major pollution sources (e.g., the expressways) as mentioned in the text.

(4) Line 179: "supplement", please point to the specific section in the supplement.

(5) Line 193: "The standard curve" $\rightarrow$ "The standard calibration curve". Please specify how often was the standard calibration performed during the sampling period.

(6) Line 195: "minimum detection limit" $\rightarrow$ " detection limit". If "minimum" indicates that the detection limit varied significantly during the sampling period, please provide the data and discussion.

(7) Line 221: please make sure the units are consistent for comparison in "42 ppbv (46, 63, and 78 µg m$^{-3}$". This extends to the CO concentration comparison in the next line.

(8) Line 380: it is not possible to draw the conclusion that "the HONO correlation in the PD period was significantly stronger" by just comparing the correlation coefficients, other factors, such as sample size, level of significance, play important rules in determining the result. Please provide details of your statistical experiment.

(9) Line 401–402: "certain high level", this is ambiguous, please be specific by giving the value of RH level here.

**4 Technical corrections**

(1) Line 89: "have concentrated on" $\rightarrow$ "have been focused on".

(2) Line 117–118: "CPER is the important region for food production and modern agriculture published by the Chinese government" $\rightarrow$ "CPER is an important region for food production and modern agriculture according to data published by the Chinese government".

(3) Line 172: "5.5 mM" → "5.5 mol m$^{-3}$". Please use SI unit. Please provide the manufacturer of $H_2O_2$ if this information is available.

(4) Line 181: "A temporal" → "The temporal".

(5) Line 187: "were subjected to" → "were subject to".

(6) Line 215: "mean value ± standard deviation" → "mean value ± 1 standard deviation". Please update all the other such occurrences in the text.

(7) Line 279: "molecule cm$^{-3}$" → "cm$^3$ molecule$^{-1}$ s$^{-1}$".

(8) Line 280: "[OH] is the concentration of OH that was not measured during the campaign." → "The concentration of OH was not measured during this campaign."

(9) Line 281: "Therefore, " → "".

(10) Line 289: "the reaction rates of k$_{OH+NO}$ and k$_{OH+HONO}$" → "the values of k$_{OH+NO}$ and k$_{OH+HONO}$".

(11) Line 291–292: "The error bars of Fig. 5 were placed separately in the tables of the supplement (Table S3)." → "The uncertainties of P$_{OH+NO}^{net}$, NO, and HONO in Fig. 5 are shown in Table S3.". Please revise similar description in line 418–419.

(12) Line 295–297: "We assumed ±50% OH values to estimate the uncertainty of P$_{OH+NO}^{net}$. The OH values of $1.25 \times 10^5$ and $3.75 \times 10^5$ molecule cm$^{-3}$ were calculated the P$_{OH+NO}^{net}$ values of 0.16 and 0.49 ppbv h$^{-1}$" → "The uncertainty of P$_{OH+NO}^{net}$ is calculated based on an assumed uncertainty of ±50% in OH concentration".

(13) Line 357: "the 6.2% average" → "the averaged value of 6.2%".

(14) Line 375: "occupied an important position" → "played an important rule".

(15) Line 397: "influenced" → "influences".

(16) Line 424: please revise the broken sentence "relatively s In the current study, directly emitted HONO state (Stutz, 2002)".

(17) Line 455: "ΔHONO/Δ$dt$" → "ΔHONO/Δ$t$"

**References**

Acker, K., Möller, D., Wieprecht, W., Meixner, F. X., Bohn, B., Gilge, S., Plass-Dülmer, C., and Berresheim, H.: Strong daytime production of OH from HNO2 at a rural mountain site, Geophysical Research Letters, 33, https://doi.org/10.1029/2005GL024643, https://agupubs.onlinelibrary.wiley.com/doi/abs/10.1029/2005GL024643, 2006.

Kleffmann, J., Gavriloaiei, T., Hofzumahaus, A., Holland, F., Koppmann, R., Rupp, L., Schlosser, E., Siese, M., and Wahner, A.: Daytime formation of nitrous acid: A major source of OH radicals in a forest, Geophysical Research Letters, 32, https://doi.org/10.1029/2005GL022524, https://agupubs.onlinelibrary.wiley.com/doi/abs/10.1029/2005GL022524, 2005.

Li, X., Brauers, T., Häseler, R., Bohn, B., Fuchs, H., Hofzumahaus, A., Holland, F., Lou, S., Lu, K. D., Rohrer, F., Hu, M., Zeng, L. M., Zhang, Y. H., Garland, R. M., Su, H., Nowak, A., Wiedensohler, A., Takegawa, N., Shao, M., and Wahner, A.: Exploring the atmospheric chemistry of nitrous acid (HONO) at a rural site in Southern China, Atmospheric Chemistry and Physics, 12, 1497–1513, https://doi.org/10.5194/acp-12-1497-2012, https://www.atmos-chem-phys.net/12/1497/2012/, 2012.